# Efficient Learning on Large Graphs using a Densifying Regularity Lemma

**Jonathan Kouchly**[*]
Technion – Israel Institute of Technology

**Ben Finkelshtein**[*]
University of Oxford

**Michael Bronstein**
University of Oxford / AITHYRA

**Ron Levie**
Technion – Israel Institute of Technology

## Abstract

Learning on large graphs presents significant challenges, with traditional Message Passing Neural Networks suffering from computational and memory costs scaling linearly with the number of edges. We introduce the Intersecting Block Graph (IBG), a low-rank factorization of large directed graphs based on combinations of intersecting bipartite components, each consisting of a pair of communities, for source and target nodes. By giving less weight to non-edges, we show how an IBG can efficiently approximate any graph, sparse or dense. Specifically, we prove a constructive version of the weak regularity lemma: for any chosen accuracy, every graph can be approximated by a dense IBG whose rank depends only on that accuracy. This improves over prior versions of the lemma, where the rank depended on the number of nodes for sparse graphs. Our method allows for efficient approximation of large graphs that are both directed and sparse, a crucial capability for many real-world applications. We then introduce a graph neural network architecture operating on the IBG representation of the graph and demonstrating competitive performance on node classification, spatio-temporal graph analysis, and knowledge graph completion, while having memory and computational complexity linear in the number of nodes rather than edges.

## 1 Introduction

Graphs are a powerful representation for structured data, with applications spanning social networks (Hamilton et al., 2017a; Zeng et al., 2019), biological systems (Hamilton et al., 2017b), traffic modeling (Li et al., 2018), and knowledge graphs (Kok & Domingos, 2007), to name a few. As graph sizes continue to grow in application, learning on such large-scale graphs presents computational and memory challenges. Traditional Message Passing Neural Networks (MPNNs), which form the backbone of most graph signal processing architectures, scale their computational and memory requirements linearly with the *number of edges*. This edge-dependence limits their scalability in some situations, e.g., when processing social networks that can typically have $10^8 \sim 10^9$ nodes and $10^2 \sim 10^3$ as many edges (Rossi et al., 2020).

Several strategies, called *graph reduction methods*, have been proposed to alleviate these challenges. These include *graph sparsification*, where a smaller graph is randomly sampled from the large graph (Hamilton et al., 2017a; Zeng et al., 2019; Chen et al., 2018); *graph condensation*, where a new small graph is created (Jin et al., 2022; Wang et al., 2024; Zheng et al., 2024), representing structures in the large graph; and *graph coarsening*, where sets of nodes are grouped into super nodes (Ying et al., 2018; Bianchi et al., 2020; Huang et al., 2021). However, with the exception of graph sparsification, graph reduction methods typically do not address the problem of processing a graph that is too large to fit at once in memory (e.g., on the GPU). For an extended related work, see Section A.

Recently, Finkelshtein et al. (2024a) proposed using a low-rank approximation of the graph, called *Intersecting Community Graph* (ICG), instead of the graph itself, for processing the data. When training a model on the ICG representation, the computational complexity is reduced from linear in

---

[*]Equal contribution. Correspondence to: kjonathan@campus.technion.ac.il

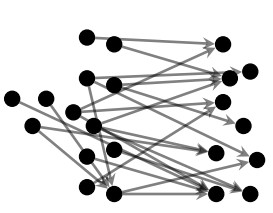 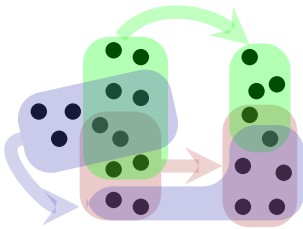 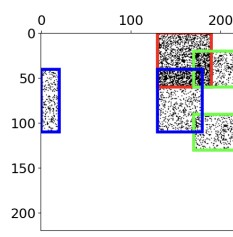

    (a) A directed graph.      (b) Approximating 3-IBG.      (c) The intersecting blocks of the 3-IBG.

Figure 1: (**a**) Directed graph sampled from a stochastic block model (SBM); (**b**) The approximating 3-IBG, where source and target community pairs are represented by the same color; (**c**) Adjacency matrix of a graph sampled from the same SBM, with the 3-IBG overlaid on the adjacency matrix.

the number of edges (as in MPNNs) to linear in the *number of nodes*. However, ICG approximation quality deteriorates as graphs become sparser – a significant limitation for domains like fraud detection, recommendation systems, and social networks where large graph size naturally results in sparse connectivity. Moreover, ICGs are restricted to undirected graphs, limiting their use in applications where edge directionality is essential, such as atmospheric flow in weather forecasting and causal reasoning in algorithms (Oskarsson et al., 2024; Smit et al., 2025). In these settings, both empirical and theoretical work has shown that directionality is key and can significantly boost GNN performance and expressive power (Rossi et al., 2024; Bechler-Speicher et al., 2025)

**Our contribution.** We introduce a new procedure for approximating general directed graphs $G$ with adjacency matrices $\boldsymbol{A} \in \{0,1\}^{N \times N}$ by low-rank matrices $\boldsymbol{C}$ that have a special interpretation: the approximating graph consists of a set of overlapping bipartite components. Namely, there is a set of $K \ll N$ pairs of node communities $(\mathcal{U}_i, \mathcal{V}_i), i = 1, \ldots, K$, and each pair defines a weighted bipartite component, in which edges connect each node of $\mathcal{U}_i$ to each node of $\mathcal{V}_i$ with some weight $r_i$ (that can be negative). The full graph $\boldsymbol{C}$ is defined as the sum of all of these components, called *blocks* or *directed communities*, where the different communities can overlap (see Figure 1 for a visualization of the approximating graph). We demonstrate how processing $\boldsymbol{C}$ instead of $\boldsymbol{A}$ leads to models that solve downstream task in linear time and space complexity with respect to the number of nodes, as opposed to standard MPNNs that are linear in the number of edges.

To fit $\boldsymbol{C}$ to $\boldsymbol{A}$, we consider a loss function $L_{\boldsymbol{A}}(\boldsymbol{C})$ defined as a *weighted norm* of $\boldsymbol{A} - \boldsymbol{C}$, namely, a standard norm weighted element-wise by a weight matrix $\boldsymbol{Q} \in (0, \infty)^{N \times N}$. The goal of using weights is to balance the contributions of edges and non-edges. The weight matrix $\boldsymbol{Q}$ is chosen adaptively, depending on the target adjacency matrix $\boldsymbol{A}$. We consider a *weighted cut norm*, denoted by $\sigma_{\square}(\boldsymbol{A}||\boldsymbol{C})$, as the approximation metric. The cut norm is a well-established graph similarity measure that quantifies the maximum discrepancy in their connectivity structure. It enables graph approximations with rank independent of the number of nodes for dense graphs. However, for sparse graphs, the standard cut-metric is dominated by non-edges, which degrades approximation quality. This motivates the use of a weighted cut-norm that balances the contributions of edges and non-edges. The cut norm also has a probabilistic interpretation that we discuss in Section 3, and Appendices B and C. Computing the cut norm is NP-hard, which prohibits explicitly optimizing it. To solve this issue, we prove that it is possible to minimize a weighted Frobenius norm $\|\boldsymbol{A} - \boldsymbol{C}\|_{\mathrm{F}}$ instead of the cut norm, and guarantee that the Frobenius minimizer $\boldsymbol{C}^*$ has a small cut error $\sigma_{\square}(\boldsymbol{A}||\boldsymbol{C}^*)$, even if the minimum $\|\boldsymbol{A} - \boldsymbol{C}^*\|_{\mathrm{F}}$ itself is large. For that, we formulate a version of the Weak Regularity Lemma (WRL) (Frieze & Kannan, 1999) that we call the *semi-constructive densifying directional soft weak regularity lemma*, or in short the *Densifying Weak Regularity Lemma*.

The WRL asserts that one can approximate any graph with $E$ edges and $N$ nodes up to error $\epsilon$ w.r.t. the cut metric by a low-rank graph consisting of $N/(\sqrt{E}\epsilon^2)$ intersecting communities. For more details on the standard WRL see Section B. Our approach, which is an extension of the ICG method (Finkelshtein et al., 2024a), is different from other forms of the WRL in a number of ways:

- While some variants of the WRL only prove existence (László Miklós Lovász, 2007), we find the approximating low-rank graph as the solution to an "easy to optimize" loss function (hence our approach is *constructive*).

- While some versions of the WRL (Frieze & Kannan, 1999) propose an algorithm that provably obtains the approximating low-rank matrix, these algorithms are exponentially slow and not applicable in practice (for example, see Section 7 of Finkelshtein et al. (2024a)). Instead, the loss function we introduce can be efficiently optimized via gradient descent. While the optimization procedure is not guaranteed to find the global minimum since the loss is non-convex, it nevertheless produces high-quality approximations in practice (hence the term *semi*-constructive). To facilitate a gradient descent-based optimization, we relax the combinatorial problem (hence the term *soft*).

- Previous versions of the WRL consider undirected graphs, while we treat general directed graphs (hence the term *directional*).

- While previous versions of the WRL required $N/(\sqrt{E}\epsilon^2)$ communities for $\epsilon$ error w.r.t. the standard cut metric, we guarantee an $\epsilon$ error in weighted cut metric with only $1/\epsilon^2$ communities. Hence, the number of communities in our method *is independent of any property of the graph, including the number of nodes and sparsity level*. This capability directly follows from balancing the importance of edges and non-edges in our optimization target. As a result, it leads to a formal approach for approximating *sparse large graphs* by *dense low rank graphs*, justifying the term *densifying*. We emphasize that this independence of the number of communities on the sparsity level is not merely an artifact of carefully renormalizing the loss to artificially facilitate the desired error bound. Rather, the loss function is deliberately designed to promote denseness when approximating graphs. The ability to efficiently densify a given graph can improve downstream tasks like node classification, as, in some sense, the densified version $\boldsymbol{C}^*$ of the graph $\boldsymbol{A}$ strengthens the connectivity patterns of the graph. We stress that as opposed to naive densification approaches, our method improves *both computational complexity and accuracy*.

In this paper, we introduce a new graph similarity measure that enables efficient approximation of any graph, including sparse and directed ones. We develop a non-trivial extension of the ICG method called the *Intersecting Blocks Graph (*IBG*)*. Our central theoretical result shows that any graph, sparse or dense, can be approximated with an error that depends only on the desired accuracy, independent of graph size or sparsity level. This advancement enables the design of IBG Neural Networks (IBG-NNs), which operate directly on the IBG representation of any graph. IBG-NNs allow solving downstream tasks such as node classification, spatio-temporal graph analysis, and knowledge graph completion in $\mathcal{O}(N)$ operations rather than $\mathcal{O}(E)$. We demonstrate that our approach achieves state-of-the art accuracy on standard benchmarks, while being very efficient. For background on the predecessor of our method, ICG (Finkelshtein et al., 2024a), see Appendix A. For a comparison of our method with ICG, see Sections J and M.2.1.

## 2 BASIC DEFINITIONS AND NOTATIONS

We denote matrices by boldface uppercase letters, e.g., $\boldsymbol{D}$, vectors by boldface lowercase $\boldsymbol{d}$, and their scalar entries by the same lowercase letter $d_i$ with subscript for the index.

**Graph signals.** We consider *directed* (unweighted) graphs $G$ with sets of $N$ nodes $\mathcal{V} = [N] = \{1, \ldots, N\}$, $E$ edges $\mathcal{E} \subseteq \mathcal{V} \times \mathcal{V}$, adjacency matrix $\boldsymbol{A} = (a_{i,j})_{i,j=1}^N \in \{0,1\}^{N \times N}$, and node feature matrix $\boldsymbol{X} = (x_{i,j})_{i,j=1}^{N,D} \in [-1,1]^{N \times D}$, called the *signal*. We follow the graph signal processing convention and represent the data as *graph-signals* $G = (\boldsymbol{A}, \boldsymbol{X})$. We emphasize that signals are always normalized to have values in $[-1,1]$, which does not limit generality as the units of measurement can always be linearly changed. All constructions also apply to signals with values in $\mathbb{R}$, but for simplicity of the analysis we limit the values to $[-1,1]$.

We denote the $j$-th column of the matrix $\boldsymbol{Q}$ by $\boldsymbol{Q}_{:,j}$, and the $i$-th row by $\boldsymbol{Q}_{i,:}$. We often also denote the $i$-th row by $\boldsymbol{q}_i^\top$ and respectively denote $\boldsymbol{Q} = (\boldsymbol{q}_i)_{i=1}^M$. We identify vectors $\boldsymbol{v} = (v_1)_{i=1}^N$ with corresponding functions $i \mapsto v_i$. Similarly, we treat $\boldsymbol{X}$ as a function $\boldsymbol{X} : [N] \to \mathbb{R}^D$, with $\boldsymbol{X}(n) = \boldsymbol{x}_n$. We denote by $\mathrm{diag}(\boldsymbol{r}) \in \mathbb{R}^{K \times K}$ the diagonal matrix with diagonal elements $\boldsymbol{r} \in \mathbb{R}^K$.

**Frobenius norm.** The *weighted Frobenius norm* of a square matrix $\boldsymbol{D} \in \mathbb{R}^{N \times N}$ with respect to the *weight* $\boldsymbol{Q} \in (0,\infty)^{N \times N}$ is defined to be $\|\boldsymbol{D}\|_{\mathrm{F};\boldsymbol{Q}} := \left(\frac{1}{\sum_{i,j=1}^N q_{i,j}} \sum_{i,j=1}^N d_{i,j}^2 q_{i,j}\right)^{1/2}$. Denote $\|\boldsymbol{D}\|_{\mathrm{F}} := \|\boldsymbol{D}\|_{\mathrm{F};\boldsymbol{1}}$, where $\boldsymbol{1}$ is the all-1 matrix. The Frobenius norm of a signal $\boldsymbol{Y} \in \mathbb{R}^{N \times D}$ is defined

by $\|\boldsymbol{Y}\|_{\mathrm{F}} := \sqrt{\frac{1}{ND}\sum_{j=1}^{D}\sum_{i=1}^{N} y_{i,j}^2}$. The *weighted Frobenius norm with weights* $\alpha, \beta > 0$ of a matrix-signal $(\boldsymbol{D}, \boldsymbol{Y})$ is defined by $\|(\boldsymbol{D}, \boldsymbol{Y})\|_{\mathrm{F};\boldsymbol{Q}} = \|(\boldsymbol{D}, \boldsymbol{Y})\|_{\mathrm{F};\boldsymbol{Q},\alpha,\beta} := \sqrt{\alpha \|\boldsymbol{D}\|_{\mathrm{F};\boldsymbol{Q}}^2 + \beta \|\boldsymbol{Y}\|_{\mathrm{F}}^2}$.

## 3 Weighted Graph Similarity Measures

**Weighted cut-metric.** The *cut-metric* is a graph similarity measure based on the *cut-norm*. Below, we define it for graphs of the same size; extensions to arbitrary graphs use graphons (see (László Miklós Lovász, 2007) for graphons, and (Levie, 2023) for graphon-signals).

**Definition 3.1.** *The weighted* matrix cut-norm *of* $\boldsymbol{D} \in \mathbb{R}^{N \times N}$ *with weights* $\boldsymbol{Q} \in (0, \infty)^{N \times N}$, *is defined to be*

$$\|\boldsymbol{D}\|_{\square;\boldsymbol{Q}} = \frac{1}{\sum_{i,j} q_{i,j}} \max_{\mathcal{U},\mathcal{V} \subset [N]} \Big| \sum_{i \in \mathcal{U}} \sum_{j \in \mathcal{V}} d_{i,j} q_{i,j} \Big|.$$

*The* signal cut-norm *of* $\boldsymbol{Y} \in \mathbb{R}^{N \times D}$ *is defined to be*

$$\|\boldsymbol{Y}\|_{\square} = \frac{1}{DN} \sum_{j=1}^{D} \max_{\mathcal{U} \subset [N]} \Big| \sum_{i \in \mathcal{U}} y_{i,j} \Big|.$$

*The* weighted matrix-signal cut-norm *of* $(\boldsymbol{D}, \boldsymbol{Y})$, *with weights* $\alpha, \beta > 0$, *is defined to be*

$$\|(\boldsymbol{D}, \boldsymbol{Y})\|_{\square;\boldsymbol{Q}} = \|(\boldsymbol{D}, \boldsymbol{Y})\|_{\square;\boldsymbol{Q},\alpha,\beta} := \alpha \|\boldsymbol{D}\|_{\square;\boldsymbol{Q}} + \beta \|\boldsymbol{Y}\|_{\square}. \tag{1}$$

Note that Finkelshtein et al. (2024a) used the weighted cut-norm $\|\boldsymbol{D}\|_{\square} := (N^2/E) \|\boldsymbol{D}\|_{\square;\mathbf{1}}$.

**Densifying cut similarity.** A key limitation of the cut-metric in sparse graphs arises from the dominance of non-edges in the graph structure. Sparse graphs, such as those common in link prediction and knowledge graph completion (Dettmers et al., 2018), often have a small number of edges relative to non-edges. This imbalance causes the cut-metric to be dominated by non-edges unless they are properly weighted. We believe that this imbalance significantly impacted the quality of the approximating ICG in Finkelshtein et al. (2024a), which uses the unweighted cut-metric, i.e. with $\boldsymbol{Q} = \mathbf{1}$, and consequently the underperformance of downstream tasks which operate on the ICG.

Motivated by this limitation, we define a similarity measure that better addresses the structural imbalance inherent in sparse graphs. We propose the *densifying cut similarity*, a modification of the cut-metric that lowers the contributions of non-edges w.r.t. the standard cut-metric. We define a weighted adjacency matrix $\boldsymbol{Q}$ that assigns a weight $e$ to non-edges and 1 to edges. The parameter $e$ is chosen based on the desired balance, controlled by a factor $\Gamma$, which determines the proportion of non-edges relative to edges. For a detailed derivation of the definition from motivating guidelines, and its relation to negative sampling in knowledge graph completion and link prediction, see Section C.

**Definition 3.2.** *Let* $\boldsymbol{A} \in \{0, 1\}^{N \times N}$ *be an unweighted adjacency matrix, and* $\Gamma > 0$. *The* densifying cut similarity *between the target* $\boldsymbol{A}$ *and any adjacency matrix* $\boldsymbol{B} \in \mathbb{R}^{N \times N}$ *is defined to be*

$$\sigma_{\square}(\boldsymbol{A}\|\boldsymbol{B}) = \sigma_{\square;\Gamma}(\boldsymbol{A}\|\boldsymbol{B}) := (1 + \Gamma) \|\boldsymbol{A} - \boldsymbol{B}\|_{\square;\boldsymbol{Q_A}},$$

*where the weight matrix* $\boldsymbol{Q_A}$ *is*

$$\boldsymbol{Q_A} = \boldsymbol{Q}_{\boldsymbol{A},\Gamma} := e_{E,\Gamma}\mathbf{1} + (1 - e_{E,\Gamma})\boldsymbol{A}, \quad \text{with } e_{E,\Gamma} = \frac{\Gamma E/N^2}{1 - (E/N^2)}. \tag{2}$$

*Given* $\alpha, \beta > 0$ *such that* $\alpha + \beta = 1$, *the* densifying cut similarity *between the target graph-signal* $(\boldsymbol{A}, \boldsymbol{X})$ *and the graph-signal* $(\boldsymbol{A}', \boldsymbol{X}')$ *is defined to be*

$$\sigma_{\square}\big((\boldsymbol{A}, \boldsymbol{X})\|(\boldsymbol{A}', \boldsymbol{X}')\big) = \sigma_{\square;\alpha,\beta,\Gamma}\big((\boldsymbol{A}, \boldsymbol{X})\|(\boldsymbol{A}', \boldsymbol{X}')\big) := \alpha\sigma_{\square;\Gamma}(\boldsymbol{A}\|\boldsymbol{B}) + \beta \|\boldsymbol{X} - \boldsymbol{X}'\|_{\square}.$$

We stress that the weighted Frobenius norm with the weight from Definition 3.2 is well normalized, i.e., $(1 + \Gamma) \|\boldsymbol{A}\|_{\mathrm{F};\boldsymbol{Q_A}} = 1$, suggesting that the norm $\sigma_{\square}\big((\boldsymbol{A}, \boldsymbol{X})\|(\boldsymbol{A}', \boldsymbol{X}')\big)$ is meaningfully standardized. Namely, we expect $\sigma_{\square}\big((\boldsymbol{A}, \boldsymbol{X})\|(\boldsymbol{A}', \boldsymbol{X}')\big)$ to have magnitude of the order of 1 when $\boldsymbol{B}$ is a "bad" approximation of $\boldsymbol{A}$, and to be $\ll 1$ when $\boldsymbol{B}$ is a "good" approximation. Also not that the standard cut metric is retrieved when $\Gamma = N^2/E - 1$.

## 4 Approximations by intersecting blocks

### 4.1 Intersecting block graphs

For any subset of nodes $\mathcal{U} \subset [N]$, the indicator function $\mathbb{1}_{\mathcal{U}}$ is defined as $\mathbb{1}_{\mathcal{U}}(i) = 1$ if $i \in \mathcal{U}$ and 0 otherwise. As explained above, we treat $\mathbb{1}_{\mathcal{U}}$ as a vector in $\mathbb{R}^N$. Denote by $\chi$ the set of all such indicator functions. We define an *Intersecting Block Graph (*IBG*)* with $K$ classes ($K$-IBG) as a low-rank graph-signal $(\boldsymbol{C}, \boldsymbol{P})$ with adjacency matrix and signals given respectively by

$$\boldsymbol{C} = \sum_{j=1}^{K} r_j \mathbb{1}_{\mathcal{U}_j} \mathbb{1}_{\mathcal{V}_j}^{\top}, \quad \boldsymbol{P} = \sum_{j=1}^{K} \mathbb{1}_{\mathcal{U}_j} \boldsymbol{f}_j^{\top} + \mathbb{1}_{\mathcal{V}_j} \boldsymbol{b}_j^{\top}$$

where $r_j \in \mathbb{R}$, $\boldsymbol{f}_j, \boldsymbol{b}_j \in \mathbb{R}^D$, and $\mathcal{U}_j, \mathcal{V}_j \subset [N]$. Next, we relax the $\{0, 1\}$-valued hard indicator functions $\mathbb{1}_{\mathcal{U}}, \mathbb{1}_{\mathcal{V}}$ to *soft affiliation functions* with values in $\mathbb{R}$, as defined next, to allow continuously optimizing IBGs. Definition 4.1 is taken from Finkelshtein et al. (2024a).

**Definition 4.1.** *A set $\mathcal{Q}$ of vectors $\boldsymbol{u} : [N] \to \mathbb{R}$ that contains $\chi$ is called a* soft affiliation model.

**Definition 4.2.** *Let $d \in \mathbb{N}$, and let $\mathcal{Q}$ be a soft affiliation model. We define $[\mathcal{Q}] \subset \mathbb{R}^{N \times N} \times \mathbb{R}^{N \times D}$ to be the set of all elements of the form $(r\boldsymbol{u}\boldsymbol{v}^{\top}, \boldsymbol{u}\boldsymbol{f}^{\top} + \boldsymbol{v}\boldsymbol{b}^{\top})$, with $\boldsymbol{u}, \boldsymbol{v} \in \mathcal{Q}$, $r \in \mathbb{R}$ and $\boldsymbol{f}, \boldsymbol{b} \in \mathbb{R}^D$. We call $[\mathcal{Q}]$ the* soft rank-1 intersecting block graph (IBG) model *corresponding to $\mathcal{Q}$. Given $K \in \mathbb{N}$, the subset $[\mathcal{Q}]_K$ of $\mathbb{R}^{N \times N} \times \mathbb{R}^{N \times D}$ of all linear combinations of $K$ elements of $[\mathcal{Q}]$ is called the* soft rank-$K$ IBG model *corresponding to $\mathcal{Q}$.*

In matrix form, an IBG $(\boldsymbol{C}, \boldsymbol{P}) \in \mathbb{R}^{N \times N} \times \mathbb{R}^{N \times D}$ in $[\mathcal{Q}]_K$ can be written as

$$\boldsymbol{C} = \boldsymbol{U} \operatorname{diag}(\boldsymbol{r}) \boldsymbol{V}^{\top} \quad \text{and} \quad \boldsymbol{P} = \boldsymbol{U}\boldsymbol{F} + \boldsymbol{V}\boldsymbol{B} \tag{3}$$

via the *target community affiliation matrix* $\boldsymbol{U} \in \mathbb{R}^{N \times K}$, the *source community affiliation matrix* $\boldsymbol{V} \in \mathbb{R}^{N \times K}$, the *community magnitude vector* $\boldsymbol{r} \in \mathbb{R}^K$, the *target community feature matrix* $\boldsymbol{F} \in \mathbb{R}^{K \times D}$ and the *source community feature matrix* $\boldsymbol{B} \in \mathbb{R}^{K \times D}$.

### 4.2 The densifying regularity lemma

Directly minimizing the densifying cut similarity (or cut metric) is both numerically unstable and computationally difficult since it involves a maximization step, making the optimization a min-max problem. To overcome this, we introduce a middle-ground solution, providing an efficient semi-constructive version of the weak regularity lemma for intersecting blocks. The approach is termed semi-constructive because it formulates the approximating graph as the solution to an "easy-to-solve" optimization problem that can be efficiently handled using standard gradient descent techniques.

The theorem generalizes the semi-constructive WRL based on intersecting communities of Finkelshtein et al. (2024a) in three main ways: (1) extending the theorem to directed graphs and the densifying graph similarity, instead of undirected graphs and the cut norm, (2) introducing a certificate for testing that the high probability event in which the cut similarity error is small occurred, and the key novelty of this theorem – (3) it addresses a major limitation of the previous work by providing a bound that is independent of the graph size for both sparse and dense graphs, whereas the bound in Finkelshtein et al. (2024a) depended on the graph size for sparse graphs. For a more extensive comparison between our approach and Finkelshtein et al. (2024a) see Appendix J.

**Theorem 4.1.** *Let $(\boldsymbol{A}, \boldsymbol{X})$ be a graph-signal, $K \in \mathbb{N}$, $\delta > 0$, and let $\mathcal{Q}$ be a soft affiliation model. Let $\alpha, \beta > 0$ such that $\alpha + \beta = 1$. Let $\Gamma > 0$ and let $\boldsymbol{Q_A}$ be the weight matrix defined in Definition 3.2. Let $R \geq 1$ such that $K/R \in \mathbb{N}$. For every $k \in \mathbb{N}$, let*

$$\eta_k = (1 + \delta) \min_{(\boldsymbol{C}, \boldsymbol{P}) \in [\mathcal{Q}]_k} \|(\boldsymbol{A}, \boldsymbol{X}) - (\boldsymbol{C}, \boldsymbol{P})\|_{\mathrm{F}; \boldsymbol{Q_A}, \alpha(1+\Gamma), \beta}^2 .$$

*Then,*

1. *For every $m \in \mathbb{N}$, any IBG $(\boldsymbol{C}^*, \boldsymbol{P}^*) \in [\mathcal{Q}]_m$ that gives a close-to-best weighted Frobenius approximation of $(\boldsymbol{A}, \boldsymbol{X})$ in the sense that*

$$\|(\boldsymbol{A}, \boldsymbol{X}) - (\boldsymbol{C}^*, \boldsymbol{P}^*)\|_{\mathrm{F}; \boldsymbol{Q_A}, \alpha(1+\Gamma), \beta}^2 \leq \eta_m, \tag{4}$$

*also satisfies*

$$\sigma_{\square;\alpha,\beta,\Gamma}\big((\boldsymbol{A},\boldsymbol{X})\|(\boldsymbol{C}^*,\boldsymbol{P}^*)\big) \le (\sqrt{\alpha(1+\Gamma)} + \sqrt{\beta})\sqrt{\eta_m - \frac{\eta_{m+1}}{1+\delta}}. \tag{5}$$

2. *If $m$ is uniformly randomly sampled from $[K]$, then in probability $1 - \frac{1}{R}$,*

$$\sqrt{\eta_m - \frac{\eta_{m+1}}{1+\delta}} \le \sqrt{\delta + \frac{R(1+\delta)}{K}}. \tag{6}$$

*Specifically, in probability $1 - \frac{1}{R}$, any $(\boldsymbol{C}^*,\boldsymbol{P}^*) \in [\mathcal{Q}]_m$ which satisfies (4), also satisfies*

$$\sigma_{\square,\alpha,\beta}\big((\boldsymbol{A},\boldsymbol{X}) - (\boldsymbol{C}^*,\boldsymbol{P}^*)\big) \le \sqrt{2+\Gamma}\sqrt{\delta + \frac{R(1+\delta)}{K}}. \tag{7}$$

The proof of Theorem 4.1 is in Appendix D.

Specifically, the three main ways in which Theorem 4.1 generalizes ICG to IBG are depicted in: (1) The use of directional IBGs, (2) the deterministic certificate for the high probability event given by Item 1, and (3) the approximation bound in Item 2 which is independent of graph sparsity.

### 4.3 Optimizing IBGs with oracle Frobenius minimizers

Suppose there exists an oracle optimization method that solves (4) in $T_K$ operations for any $m \le K$. Theorem 4.1 motivates the following algorithm for approximating a graph-signal by an IBG.

- Randomly sample $m \in [K]$. By Item 2 of Theorem 4.1, the approximation bound (7) is satisfied in high probability $(1 - 1/R)$.
- To verify that (7) really happened for the given realization of $m$, we estimate the left-hand-side of (6), using the oracle optimizer in $2T_K$ operations (computing both $\eta_m$ and $\eta_{m+1}$), checking if (6) is satisfied. If it is, it guarantees (7) by (5).
- If the bound (6) is not satisfied (in probability less than $\frac{1}{R}$), resample $m$ and repeat.

The expected number of resamplings of $m$ is $R/(R-1)$, so the algorithm's expected runtime to find an IBG satisfying (7) is $2T_K R/(R-1)$. In practice, instead of an oracle optimizer, we apply gradient descent to estimate the optimum of the left-hand side of (4), which requires $T_K = \mathcal{O}(E)$ operations by Theorem 4.2. This makes the algorithm as efficient as message passing in practice.

### 4.4 Fitting intersecting blocks using gradient descent

In this section, we propose an efficient computation for fitting IBGs to directed graphs based on Theorem 4.1 (minimizing the left-hand-side of (4) via gradient descent). As the soft affiliation model, we consider all vectors in $[0,1]^N$. In the notations of (3), we optimize the parameters $\boldsymbol{U},\boldsymbol{V} \in [0,1]^{N \times K}$, $\boldsymbol{r} \in \mathbb{R}^K$ and $\boldsymbol{F},\boldsymbol{B} \in \mathbb{R}^{K \times D}$ to minimize the weighted Frobenius norm

$$L(\boldsymbol{U},\boldsymbol{V},\boldsymbol{r},\boldsymbol{F},\boldsymbol{B}) = \alpha(1+\Gamma)\left\|\boldsymbol{A} - \boldsymbol{U}\operatorname{diag}(\boldsymbol{r})\boldsymbol{V}^\top\right\|^2_{\mathrm{F};\boldsymbol{Q_A}} + \beta\|\boldsymbol{X} - \boldsymbol{U}\boldsymbol{F} - \boldsymbol{V}\boldsymbol{B}\|^2_{\mathrm{F}}. \tag{8}$$

In practice, we implement $\boldsymbol{U},\boldsymbol{V} \in [0,1]^{N \times K}$ by applying a sigmoid activation function to learned matrices $\boldsymbol{U}',\boldsymbol{V}' \in \mathbb{R}^{N \times K}$, setting $\boldsymbol{U} = \operatorname{Sigmoid}(\boldsymbol{U}')$ and $\boldsymbol{V} = \operatorname{Sigmoid}(\boldsymbol{V}')$.

Optimizing (8) naïvely requires $\mathcal{O}(N^2)$ operations, as the matrix $\boldsymbol{A} - \boldsymbol{U}\operatorname{diag}(\boldsymbol{r})\boldsymbol{V}^\top \in \mathbb{R}^{N \times N}$ is not sparse nor low-rank. However, we can exploit the sparsity of $\boldsymbol{A}$ and the low-rank structure of $\boldsymbol{U}\operatorname{diag}(\boldsymbol{r})\boldsymbol{V}^\top$ separately to enable an efficient computation with time and space complexities of $\mathcal{O}(K^2N + KE)$ and $\mathcal{O}(KN + E)$, respectively.

**Proposition 4.2.** *Let $\boldsymbol{A} = (a_{i,j})_{i,j=1}^N$ be an adjacency matrix of an unweighted graph with $E$ edges. The graph part of the sparse Frobenius loss (8) can be written as*

$$\left\|\boldsymbol{A} - \boldsymbol{U}\operatorname{diag}(\boldsymbol{r})\boldsymbol{V}^\top\right\|^2_{\mathrm{F};\boldsymbol{Q_A}} = \|A\|^2_{\mathrm{F};\boldsymbol{Q_A}} + \frac{e_{E,\Gamma}}{(1+\Gamma)E}\operatorname{Tr}\big((\boldsymbol{V}^\top\boldsymbol{V})\operatorname{diag}(\boldsymbol{r})(\boldsymbol{U}^\top\boldsymbol{U})\operatorname{diag}(\boldsymbol{r})\big)$$

$$-\frac{2}{(1+\Gamma)E}\sum_{i=1}^N\sum_{j\in\mathcal{N}(i)}\boldsymbol{U}_{i,:}\operatorname{diag}(\boldsymbol{r})\left(\boldsymbol{V}^\top\right)_{:,j}a_{i,j} + \frac{1-e_{E,\Gamma}}{(1+\Gamma)E}\sum_{i=1}^N\sum_{j\in\mathcal{N}(i)}\left(\boldsymbol{U}_{i,:}\operatorname{diag}(\boldsymbol{r})\left(\boldsymbol{V}^\top\right)_{:,j}\right)^2$$

*where $\boldsymbol{Q_A}$ and $e_{E,\Gamma}$ are defined in (2). Computing the right-hand-side and its gradients with respect to $\boldsymbol{U},\boldsymbol{V}$ and $\boldsymbol{r}$ has a time complexity of $\mathcal{O}(K^2N + KE)$, and a space complexity of $\mathcal{O}(KN + E)$.*

We prove Proposition 4.2 in Section E. The parametres of the IBG, $U, V, r, F$, and $B$, are optimized efficiently using gradient descent on Equation (8), but restructured like Proposition 4.2.

### 4.5 THE LEARNING PIPELINE WITH IBGS

When learning on large graphs using IBGs, the first step is fitting an IBG to the given graph. This is done once with little to no hyperparameter tuning in $\mathcal{O}(E)$ time and memory complexity. The second step is solving the task, e.g., node classification. This step typically involves an extensive hyperparameter search. In our pipeline, the neural network processes the IBG representation of the data, instead of the standard graph representation. This improves time complexity from $\mathcal{O}(E)$ to $\mathcal{O}(N)$. Thus, when searching through $S \in \mathbb{N}$ hyperparameter configurations, the whole search takes $\mathcal{O}(SN)$ time, while learning directly on the graph would take $\mathcal{O}(SE)$. The efficiency of our pipeline is even more pronounced in spatio-temporal prediction, where the graph remains fixed while node features evolve over time. Here, the IBG is fitted to the graph only once, regardless of the time steps.

**Initialization of IBG.** In Appendix G we explain how to use a low-rank SVD of the graph to efficiently initialize a rank $K$ IBG, before the gradient descent minimization of Equation (8). When the graph is too large to fit in memory, we also propose an efficient randomized SVD algorithm for approximating the SVD while only loading a fraction of the graph into memory (Appendix G.2).

**SGD for fitting IBGs to large graphs.** Fitting an IBG to a graph requires $\mathcal{O}(E)$ memory complexity, which may exceed the GPU capacity in some situations. To solve this, in Appendix H we propose a sampling approach for optimizing the IBG, which reduces the memory complexity, allowing fitting IBGs to large graphs on hardware with limited memory.

## 5 PROCESSING IBGS WITH NEURAL NETWORKS

**Graph signal processing with IBG.** Finkelshtein et al. (2024a) proposed a signal processing paradigm for learning on ICGs. In this section we provide an extended paradigm for learning on IBGs, which runs in $\mathcal{O}(NK)$ operations per layer, which is often faster than the $\mathcal{O}(E)$ complexity of MPNNs. Let $U, V \in \mathbb{R}^{N \times K}$ be target and source community affiliation matrices. We call $\mathbb{R}^{N \times D}$ the *node space* and $\mathbb{R}^{K \times D}$ the *community space*. We use the following operations to process signals:

- *Target synthesis* and *source synthesis* are the respective mappings $F \mapsto UF, B \mapsto VB$ from the community space to the node space, in $O(NKD)$.
- *Target analysis* and *source analysis* are the respective mappings $X \mapsto U^\dagger X$ or $X \mapsto V^\dagger X$ from the node space to the community space, in $O(NKD)$.
- *Community processing* refers to any operation that manipulates the community feature vectors $F$ and $B$ (e.g., an MLP) in $O(K^2 D^2)$ operations (or less).
- *Node processing* is any function that operates on node features in $O(ND^2)$ operations.

**IBG Neural Networks.** We propose an IBG-based architecture (IBG-NN) defined as follows. Let $D^{(\ell)}$ denote the dimension of the node features at layer $\ell$, and set the initial node representations as $H^{(0)} = X$. Then, for layers $0 \leq \ell \leq L - 1$, the node features are defined by

$$H_s^{(\ell+1)} = \sigma\left(\Theta_1^s\left(H_s^{(\ell)}\right) + \Theta_2^s\left(VB^{(\ell)}\right)\right), \quad H_t^{(\ell+1)} = \sigma\left(\Theta_1^t\left(H_t^{(\ell)}\right) + \Theta_2^t\left(UF^{(\ell)}\right)\right),$$

and require $O(D(NK + KD + ND))$ operations. The final representation, used for node-level predictions, is taken as $H^{(L)} = H_s^{(L)} + H_t^{(L)}$, where $\Theta_1$ and $\Theta_2$ are MLPs or multiple layers of deepsets, $F^{(\ell)}, B^{(\ell)} \in \mathbb{R}^{K \times D^{(\ell)}}$ are taken as trainable parameters, and $\sigma$ is a non-linearity. See Sections J to L for further details and comparisons with MPNNs and ICG-NNs.

**IBG-NNs for spatio-temporal graphs.** Given a graph with fixed connectivity and time-varying node features, we fit the IBG to the graph once. We then train a model on the frozen IBG to predict the next-step signal from past time steps. Thus, given $T$ training signals, an IBG-NN requires $\mathcal{O}(TNKD^2)$ operations per epoch, compared to $\mathcal{O}(TED^2)$ for MPNNs, with the preprocessing time remaining independent of $T$. Thus, as the number of training signals increases, the efficiency gap between IBG-NNs and MPNNs becomes more pronounced.

## 6 EXPERIMENTS

In this section, we conduct experiments addressing the core research questions:

**Q1** Does IBG-NN 's computational efficiency match theoretical expectations in practice when compared to traditional MPNNs on dense and sparse graphs?

**Q2** Do IBG-NN's theoretical guarantees for approximating arbitrary directed and sparse graphs translate to improved empirical performance compared to traditional GNNs and ICG-NN?

**Q3** How does IBG-NN perform against other graph condensation methods, across varying condensation ratios on large-scale graph benchmarks?

In Section M, we further expand our evaluation with a series of additional experiments and ablations. We conduct an ablation study across additional domains, showcasing the versatility and applicability of IBG-NN on **Spatio-temporal tasks** (Section M.1.1), **Knowledge graph completion** (Section M.1.2), and **node classification using subgraph SGD** (Section M.1.3), achieving state-of-the-art performance. We further validate **the importance of densification** (Section M.3) with additional comparisons to ICG-NN. We analyze the computational advantages of IBG-NN with **memory complexity** experiments (Section M.4.1), and provide an **ablation on the number of communities** (Section M.2), showcasing how adding communities can lead to performance improvements and better approximations. Lastly, we **test our SVD initialization method** (Section M.4.2), demonstrating improvements in convergence time of IBG approximation.

Full hyperparameter details are provided in Section P and our codebase is publicly available at: `https://anonymous.4open.science/r/IBGNN`.

### 6.1 THE EFFICIENT RUN-TIME OF IBG-NNS

**Setup.** To evaluate the efficiency of IBG-NN (**Q1**), we measure the forward pass runtimes of IBG-NN and DirGNN (Rossi et al., 2024) – a simple and efficient method for directed graphs. We then compare it on dense *Erdős-Rényi* $ER(n, p = 0.5)$ graphs and sparse $ER(n, p = 25/n)$ graphs with up to $7,000$ nodes. We sample $128$ node features per node, independently from $U[0, 1]$. Both models use a hidden and output dimension of $128$, and $3$ layers.

**Results.** Figure 2 shows that the runtime of IBG-NN is consistently faster than DirGNN across both dense and sparse graph settings. For dense graphs, IBG-NN runtime exhibits a strong square root relationship when compared to DirGNN. This matches our theoretical expectations given that IBG-NN and MPNNs have $\mathcal{O}(N)$ and $\mathcal{O}(E)$ complexity respectively. For sparse graphs, the scaling relationship appears linear. This still aligns with our theoretical expectations, as in this setting the number of edges scales linearly with $N$. Still, compared to DirGNN, IBG-NN achieves significant speedups of $5.68\times$ and $5.26\times$ for $K = 10$ and $K = 100$ respectively. Notably, even when using $K = 100$ communities, which exceeds the average degree of $25$ used in the experiments, IBG-NN still maintains faster performance. This advantage stems from IBG-NN's simple and efficient operations compared to the complex message-passing computations required by DirGNN.

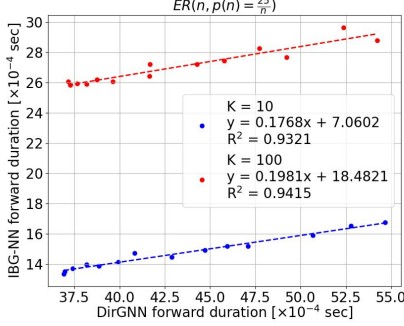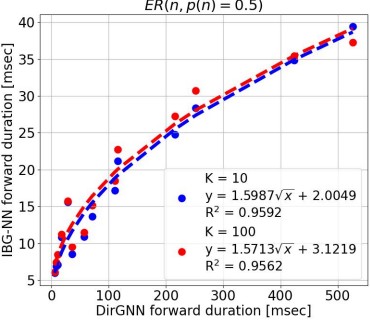

Figure 2: Runtime of K-IBG-NN as a function of DirGNN forward pass duration on sparse $ER(n, p = 25/n)$ graphs **(left)** and dense $ER(n, p = 0.5)$ graphs **(right)** for K=10, 100.

Table 1: Results on directed node classification benchmarks; top models colored First, Second, Third.

| Model | Squirrel | Chameleon | Tolokers |
|---|---|---|---|
| MLP | $28.77 \pm 1.56$ | $46.21 \pm 2.99$ | $72.95 \pm 1.06$ |
| GCN (Kipf & Welling, 2017) | $53.43 \pm 2.01$ | $64.82 \pm 2.24$ | $83.64 \pm 0.67$ |
| GAT (Veličković et al., 2018) | $40.72 \pm 1.55$ | $66.82 \pm 2.56$ | $83.70 \pm 0.47$ |
| H$_2$GCN (Zhu et al., 2020) | $61.90 \pm 1.40$ | $46.21 \pm 2.99$ | $73.35 \pm 1.01$ |
| GPR-GNN (Chien et al., 2020) | $74.80 \pm 0.50$ | $78.30 \pm 0.60$ | $72.94 \pm 0.97$ |
| FSGNN (Maurya et al., 2021) | $74.10 \pm 1.89$ | $78.27 \pm 1.28$ | – |
| GloGNN (Li et al., 2022b) | $57.88 \pm 1.76$ | $71.21 \pm 1.84$ | $73.39 \pm 1.17$ |
| DirGNN (Rossi et al., 2024) | $75.13 \pm 1.95$ | $79.74 \pm 1.40$ | – |
| FaberNet (Koke & Cremers, 2024) | $76.71 \pm 1.92$ | $80.33 \pm 1.19$ | – |
| ICG-NN (Finkelshtein et al., 2024a) | $64.02 \pm 1.67$ | $63.9 \pm 2.13$ | $83.73 \pm 0.78$ |
| IBG-NN (undirected) | $70.02 \pm 1.34$ | $75.15 \pm 1.33$ | $83.76 \pm 0.51$ |
| **IBG-NN** | $77.63 \pm 1.79$ | $80.15 \pm 1.13$ | $83.76 \pm 0.75$ |

## 6.2 THE IMPACT OF DIRECTIONALITY AND DENSIFICATION

**Setup.** To evaluate the impact of our theoretical guarantees (**Q2**), enabled by directionality and densification, we compare IBG-NN to ICG-NN (Finkelshtein et al., 2024a), and IBG-NN with an undirected optimization setting. This allows us to isolate the contribution of each addition. We evaluate IBG-NN on several directed benchmark datasets: Tolokers (Platonov et al., 2023), Squirrel, and Chameleon (Pei et al., 2020), following the 10 splits of Platonov et al. (2023); Pei et al. (2020). We report average ROC AUC and standard deviation for Tolokers, and average accuracy and standard deviation for Squirrel and Chameleon. For Tolokers, we report the baselines MLP, GCN (Kipf & Welling, 2017), GAT (Veličković et al., 2018), H$_2$GCN (Zhu et al., 2020), GPR-GNN (Chien et al., 2020), FSGNN (Maurya et al., 2021), GloGNN (Li et al., 2022b) and ICG-NN taken from Finkelshtein et al. (2024a). For Squirrel and Chameleon, we report the same baselines, as well as FSGNN (Maurya et al., 2021), DirGNN (Rossi et al., 2024) and FaberNet (Koke & Cremers, 2024), taken from (Koke & Cremers, 2024).

**Results.** Table 1 establishes IBG-NNs as state-of-the-art for directed graphs, surpassing GNNs specifically tailored for directed graphs, despite their quadratic scaling compared to IBG-NNs. The results reveal that both directionality and densification contribute significantly to the strong performance of IBG-NN. Specifically, IBG-NN without directionality surpasses ICG-NN by $6\%$ on Squirrel and $11.2\%$ on Chameleon, improvements attributed to densification. Adding directionality provides additional improvements of $7.6\%$ and $5\%$ respectively, achieving state-of-the-art performance on both datasets. For Tolokers, both IBG-NN variants achieve nearly identical performance with minimal improvement over ICG-NN. We believe this could be the result of Tolokers already being very dense, minimizing the benefits provided by densification. Similarly, the graph's directed structure may not contain meaningful directional patterns that improve node classification performance.

## 6.3 IBG-NN ON LARGE-SCALE GRAPH BENCHMARKS

**Setup.** To evaluate IBG-NN's performance on large graphs (**Q3**), we compare IBG-NNs and their predecessor ICG-NN on the large graphs Reddit (Hamilton et al., 2017a), Flickr (Zeng et al., 2019), Arxiv and Products datasets (Hu et al., 2020), following the data split provided in (Zheng et al., 2024). Accuracy and standard deviation are reported for experiments conducted with 5 different seeds over varying condensation ratios $r = \frac{M}{N^2}$, where $N$ is the total number of nodes, and $M$ is the number of sampled entries of the graph adjacency matrix. The graph coarsening baselines Random (Huang et al., 2021), Herding (Welling, 2009), K-Center (Sener & Savarese, 2017) and the graph condensation baselines GCOND (Jin et al., 2021), SFGC (Zheng et al., 2024), GC-SNTK (Wang et al., 2024) and SimGC (Xiao et al., 2024) are taken from (Zheng et al., 2024; Wang et al., 2024; Xiao et al., 2024). We note that a condensation ratio of $100\%$ corresponds to a standard GCN for the baseline methods.

**Results.** Table 2 demonstrates that subgraph SGD IBG-NN achieves state-of-the-art performance, surpassing other coarsening and condensation methods that operate on the full graph in memory,

Table 2: Results on graph densification benchmarks; top models colored First, Second, Third.

| | Flickr | | | Reddit | | | Arxiv | Products |
|---|---|---|---|---|---|---|---|---|
| Condensation (%) | 0.5% | 1% | 100% | 0.1% | 0.2% | 100% | 0.05% | 0.02% |
| Random | 44.0 ± 0.4 | 44.6 ± 0.2 | 47.2 ± 0.1 | 58.0 ± 2.2 | 66.3 ± 1.9 | 93.9 ± 0.0 | 47.1 ± 3.9 | 53.5 ± 1.3 |
| Herding | 43.9 ± 0.9 | 44.4 ± 0.6 | 47.2 ± 0.1 | 62.7 ± 1.0 | 71.0 ± 1.6 | 93.9 ± 0.0 | 52.4 ± 1.8 | 55.1 ± 0.3 |
| K-Center | 43.2 ± 0.1 | 44.1 ± 0.4 | 47.2 ± 0.1 | 53.0 ± 3.3 | 58.5 ± 2.1 | 93.9 ± 0.0 | 47.2 ± 3.0 | 48.5 ± 0.2 |
| GCOND | 47.1 ± 0.1 | 47.1 ± 0.1 | 47.2 ± 0.1 | 89.6 ± 0.7 | 90.1 ± 0.5 | 93.9 ± 0.0 | 61.3 ± 0.5 | 55.0 ± 0.8 |
| SFGC | 47.0 ± 0.1 | 47.1 ± 0.1 | 47.2 ± 0.1 | 90.0 ± 0.3 | 89.9 ± 0.4 | 93.9 ± 0.0 | 65.5 ± 0.7 | 61.7 ± 0.5 |
| GC-SNTK | 46.8 ± 0.1 | 46.5 ± 0.2 | 47.2 ± 0.1 | – | – | – | 64.4 ± 0.2 | – |
| SimGC | 45.6 ± 0.4 | 43.8 ± 1.5 | 47.2 ± 0.1 | 91.1 ± 1.0 | 92.0 ± 0.3 | 93.9 ± 0.0 | 63.6 ± 0.8 | 63.3 ± 1.1 |
| ICG-NN | 50.1 ± 0.2 | 50.8 ± 0.1 | 52.7 ± 0.1 | 89.7 ± 1.3 | 90.7 ± 1.5 | 93.6 ± 1.2 | – | – |
| **IBG-NN** | 50.5 ± 0.1 | 51.3 ± 0.2 | 53.0 ± 0.1 | 92.3 ± 1.1 | 92.6 ± 0.6 | 94.1 ± 0.5 | 64.4 ± 1.1 | 61.9 ± 0.3 |

while also improving upon the performance of its predecessor, ICG-NN. A comparison between graph coarsening methods and IBG-NNs can be found in Section A.

## 7 CONCLUSION

We proved a new semi-constructive version of the weak regularity lemma, in which the required number of communities for a given approximation error is independent of any property of the graph, including size and sparsity. This contrasts previous formulations of the lemma, where the number of communities increased with graph size for sparse graphs. Our formulation is achieved by introducing the densifying cut similarity, which, when optimized, leads the approximating IBG to effectively densify the target graph. This enables fitting IBGs of very low rank ($K = \mathcal{O}(1)$) to large sparse graphs, while previous works required the target graph to be dense for low rank approximations. We introduced IBG-NNs– a network which operates on the IBG instead of the graph, and has $\mathcal{O}(N)$ time and memory complexity. IBG-NNs demonstrate state-of-the-art performance in multiple domains: node classification on directed graphs, spatio-temporal graph analysis, and knowledge graph completion.

## ACKNOWLEDGMENTS

This research was supported by a grant from the United States-Israel Binational Science Foundation (BSF), Jerusalem, Israel, and the United States National Science Foundation (NSF), (NSF-BSF, grant No. 2024660) , and by the Israel Science Foundation (ISF grant No. 1937/23).

## REPRODUCIBILITY STATEMENT

We provide a public codebase with a complete implementation of all methods presented. All datasets used in the experiments are publicly available benchmarks. We report full experimental settings, computational resources, and hyperparameters for all conducted experiments. Theoretical results are fully supported with rigorous proofs provided in the appendices.

## ETHICS STATEMENT

This paper presents work whose goal is to advance the field of Machine Learning, and involves no human subjects or sensitive information. Our work is mostly theoretical, and poses no potentially harmful societal consequences.

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

## A  RELATED WORK

**Intersecting Community Graphs (ICG).** Our work continues the ICG setting of Finkelshtein et al. (2024a), which introduces a weak regularity lemmas for practical graph computations. The work of Finkelshtein et al. (2024a) presented a pipeline for operating on undirected, non-sparse graphs. Similarly to our work, Finkelshtein et al. (2024a) follow a two stage procedure. In the first stage, the graph is approximated by learning a factorization into undirected communities, forming what they call an Intersecting Community Graph (ICG). In the second stage, the ICG is used to enrich a neural network operating on the node features and community graph without using the original edge connectivity. This setup allows for more efficient computation, both in terms of runtime and memory, since the full edge structure of the graph, used in standard GNNs, can be replaced with a much smaller community-level graph—especially useful for graphs with a high average degree. The constructive weak regularity lemma presented in (Finkelshtein et al., 2024a) shows any graph can be approximated in *cut-norm*, regardless of its size, by minimizing the easy to compute Frobenius error.

More concretely, an ICG with $K$ communities is just like an IBG with only node features (no edge-features), but with the transmitting and receiving communities being equal $\mathbf{U} = \mathbf{V}$. Namely, an ICG can be represented by a triplet of *community affiliation matrix* $\boldsymbol{Q} \in \mathbb{R}^{N \times K}$, *community magnitude vector* $\boldsymbol{r} \in \mathbb{R}^K$, and *community feature matrix* $\mathbf{F} \in \mathbb{R}^{K \times D}$. An ICG $(\boldsymbol{C}, \boldsymbol{P})$ with adjacency matrix $\boldsymbol{C}$ and signal $\boldsymbol{P}$ is then given by

$$\boldsymbol{C} = \boldsymbol{Q} \operatorname{diag}(\boldsymbol{r}) \boldsymbol{Q}^\top \quad \text{and} \quad \boldsymbol{P} = \boldsymbol{Q} \boldsymbol{F},$$

where $\operatorname{diag}(\boldsymbol{r})$ is the diagonal matrix in $\mathbb{R}^{K \times K}$ with $\boldsymbol{r}$ as its diagonal elements. Here, $K$ is the number of communities, $N$ is the number of nodes, and $E$ is the number of edges.

When approximating a graph-signal $(\boldsymbol{A}, \boldsymbol{X})$, the measure of accuracy, or error, in (Finkelshtein et al., 2024a) is defined to be the standard (unweighted) cut metric $\|(\boldsymbol{A}, \boldsymbol{X}) - (\boldsymbol{C}, \boldsymbol{P})\|_\square$. The semi-constructive regularity lemma of Finkelshtein et al. (2024a) states that it is enough to minimize the standard Forbenius error $\|(\boldsymbol{A}, \boldsymbol{X}) - (\boldsymbol{C}, \boldsymbol{P})\|_\mathrm{F}$ in order to guarantee

$$\|(\boldsymbol{A}, \boldsymbol{X}) - (\boldsymbol{C}, \boldsymbol{P})\|_\square = \mathcal{O}(N/\sqrt{KE}), \tag{9}$$

Looking at (9) it is clear that in order to guarantee a small approximation error in cut metric, the number of communities must increase as $N/\sqrt{E}$ becomes larger. Specifically, the number of communities $K$ is independent of the size of the graph only when $E = N^2$, i.e., the graph is dense. Hence, the ICG method falls short for sparse graphs

Our IBG method solves this shortcoming and more. For example, a main contribution of our method is a densification mechanism, supported by our novel densifying cut similarity measure and our densifying regularity lemma, which is a non-trivial continuation and extension of the semi-constructive weak regularity lemma of Finkelshtein et al. (2024a). Please see Appendix J for a detailed comparison of our IBG method to ICG.

**Cluster Affiliation models (BigClam and PieClam).** A similar work is PieClam (Zilberg & Levie, 2025), extending the well known BigClam model (Yang & Leskovec, 2013), which builds a probabilistic model of graphs as intersections of overlapping communities. While BigClam only allows communities with positive coefficients, which limits the ability to approximate many graphs, like bipartite graphs, PieClam formulates a graph probabilistic autoencoder that also includes negative communities. This allows approximately encoding any dense graph with a fixed budget of parameters per node. We note that as opposed to ICG and PieClam, which can only theoretically approximate dense symmetric graphs with $\mathcal{O}(1)$ communities, our IBG method can approximate both sparse and dense (non-symmetric in general) graphs with $\mathcal{O}(1)$ communities via the densification mechanism (the densifying constructive regularity lemma with respect to the densifying cut similarity).

**GNNs for directed graphs.** The standard practice in GNN design is to assume that the graph is undirected (Kipf & Welling, 2017). However, this assumption not only alters the input data by discarding valuable directional information, but also overlooks the empirical evidence demonstrating that leveraging edge directionality can significantly enhance performance (Rossi et al., 2024). For instance, DirGNN (Rossi et al., 2024) extends message-passing neural networks (MPNNs) to directed graphs, while Geisler et al. (2023) adapts transformers for the same purpose. FaberNet (Koke & Cremers, 2024) generalizes spectral convolutions to directed graphs, all of which have led to improved performance. Co-GNN (Finkelshtein et al., 2024b) demonstrates the advantage of learning edge

directionality over using conventional undirected graph representations. Furthermore, the proper handling of directed edges has enabled Maskey et al. (2024) to extend the concept of oversmoothing to directed graphs, providing deeper theoretical insights. IBG-NNs also capitalize on edge directionality, achieving notable performance improvements over their predecessor ICG-NNs (Finkelshtein et al., 2024a), as demonstrated in Section M.2.1. When compared to existing GNNs designed for directed graphs, IBG-NNs offer a more efficient approach to signal processing. Specifically, for IBG-NNs to outperform message-passing-based GNNs in terms of efficiency, the condition $KN < E$ must hold. Such a choice of $K$ typically produces good performance for most graphs. This efficiency advantage allows IBG-NNs to make better use of the input edges while being more efficient than traditional GNNs.

**Graph Pooling GNNs.** Graph pooling GNNs generate a sequence of increasingly coarsened graphs by aggregating neighboring nodes into "super-nodes" (Ying et al., 2018; Bianchi et al., 2020), where standard message-passing is applied on the intermediate coarsened graphs. Similarly, in IBG-NNs, the signal is projected, but onto overlapping blocks rather than disjoint clusters, with several additional key distinctions: (1) The blocks in IBG-NNs are overlapping and cover large regions of the graph, allowing the method to preserve fine-grained, high-frequency signal details during projection, unlike traditional graph pooling approaches. (2) Operations on community features in IBG-NNs possess a global receptive field, enabling the capture of broader structural patterns across the graph – an extremely difficult task for local graph pooling approach. (3) IBG-NNs diverge from the conventional message-passing framework: the flattened community feature vector, which lacks symmetry, is processed by a general multilayer perceptron (MLP), whereas message-passing neural networks (MPNNs) apply the same function uniformly to all edges. (4) IBG-NNs operate exclusively on an efficient data structure, offering both theoretical guarantees and empirical evidence of significantly improved computational efficiency compared to graph pooling methods.

**Graph reduction methods.** Graph reduction aims to reduce the size of the graph while preserving key information. It can be categorized into three main approaches: graph sparsification, graph coarsening and graph condensation. Graph sparsification methods (Hamilton et al., 2017a; Zeng et al., 2019; Chen et al., 2018; Rong et al., 2020) approximate a graph by retaining only a subset of its edges and nodes, often employing random sampling techniques. Graph coarsening (Fey et al., 2020; Huang et al., 2021) clusters sets of nodes into super-nodes while aggregating inter-group edges into super-edges, aiming to preserve structural properties such as the degree distribution (Zhou et al., 2023). Graph condensation (Jin et al., 2021) generates a smaller graph with newly created nodes and edges, designed to maintain the performance of GNNs on downstream tasks.

While subgraph SGD in IBG-NNs also involves subsampling, it differs fundamentally by providing a provable approximation of the original graph. This contrasts with graph sparsification for example – where some, hopefully good heuristic-based sampling is often employed. More importantly, subgraph IBG-NNs offer a subgraph sampling approach for cases where the original graph is too large to fit in memory. This contrasts with the aforementioned coarsening and condensation methods, which lack a strategy for managing smaller data structures during the computation of the compressed graph.

Graph reduction methods generally rely on locality, applying message-passing on the reduced graph. In particular, condensation techniques require $\mathcal{O}(EM)$ operations to construct a smaller graph Jin et al. (2021); Zheng et al. (2024); Wang et al. (2024), where $E$ is the number of edges in the original graph and $M$ is the number of nodes in the condensed graph. In contrast, IBG-NNs estimate the IBG with only $\mathcal{O}(E)$ operations.

Furthermore, while conventional reduction methods process representations on either an iteratively coarsened graph or mappings between the full and reduced graphs, IBG-NNs incorporate fine-grained node information at every layer, leading to richer representations.

**Cut metric in graph machine learning.** The cut metric is a useful similarity measure, which can separate any non-isomorphic graphons Lovász (2012). This makes the cut metric particularity useful in deriving new theoretical insights for graph machine learning. For instance, (Levie, 2023) demonstrated that GNNs with normalized sum aggregation cannot separate graph-signals that are close to each other in cut metric. Using the cut distance as a theoretical tool, (Maskey et al., 2023) proves that spectral GNNs with continuous filters are transferable between graphs in sequences of that converge in homomorphism density. Finkelshtein et al. (2024a) introduced a semi constructive weak regularity lemma and used it to build new algorithms on large undirected non-sparse graphs. In

this work we introduce a new graph similarity measure – the densifying cut similarity, which gives higher importance to edges than non-edges in a graph. This allows us to approximate any graph using a set of overlapping bipartite components, where the size of the set only depends on the error tolerance. Similarly to Finkelshtein et al. (2024a), we present a semi constructive weak regularity lemma. As oppose to Finkelshtein et al. (2024a), using our novel similarity measure, our regularity lemma can be used to build new algorithms on large directed graphs which are sparse.

## B  THE WEAK REGULARITY LEMMA

Consider a graph $G$ with a node set $\mathcal{V} = [N] = \{1, \ldots, N\}$ and an edge set $\mathcal{E} \subseteq \mathcal{V} \times \mathcal{V}$. We define an equipartition $\mathcal{P} = \{\mathcal{V}_1, \ldots \mathcal{V}_k\}$ as a partition of $\mathcal{V}$ into $k$ sets where $||\mathcal{V}_i| - |\mathcal{V}_j|| \leq 1$ for every $1 \leq i, j \leq k$. For any pair of subsets $\mathcal{U}, \mathcal{S} \subset \mathcal{V}$ denote by $e_G(\mathcal{U}, \mathcal{S})$ the number of edges between $\mathcal{U}$ and $\mathcal{S}$. Now, consider two node subsets $\mathcal{U}, \mathcal{S} \subset \mathcal{V}$. If the edges between $\mathcal{V}_i$ and $\mathcal{V}_j$ were to be uniformly and independently distributed, then the expected number of edges between $\mathcal{U}$ and $\mathcal{S}$ would be

$$e_{\mathcal{P}(\mathcal{U}, \mathcal{S})} := \sum_{i=1}^{k} \sum_{j=1}^{k} \frac{e_G(\mathcal{V}_i, \mathcal{V}_j)}{|\mathcal{V}_i| \, |\mathcal{V}_j|} \, |\mathcal{V}_i \cap \mathcal{U}| \, |\mathcal{V}_j \cap \mathcal{S}| \, .$$

Using the above, we define the *irregularity*:

$$\mathrm{irreg}_G(\mathcal{P}) = \max_{\mathcal{U}, \mathcal{S} \subset \mathcal{V}} |e_G(\mathcal{U}, \mathcal{S}) - e_{\mathcal{P}}(\mathcal{U}, \mathcal{S})| / |\mathcal{V}|^2 \, . \tag{10}$$

The *irregularity* measures how non-random like the edges between $\{\mathcal{V}_j\}$ behave.

We now present the weak regularity lemma.

**Theorem B.1** (Weak Regularity Lemma Frieze & Kannan (1999)). *For every $\epsilon > 0$ and every graph $G = (\mathcal{V}, \mathcal{E})$, there is an equipartition $\mathcal{P} = \{\mathcal{V}_1, \ldots, \mathcal{V}_k\}$ of $\mathcal{V}$ into $k \leq 2^{c/\epsilon^2}$ classes such that $\mathrm{irreg}_G(\mathcal{P}) \leq \epsilon$. Here, $c$ is a universal constant that does not depend on $G$ and $\epsilon$.*

The weak regularity lemma states that any large graph $G$ can be approximated by a weighted graph $G^\epsilon$ with node set $\mathcal{V}^\epsilon = \{\mathcal{V}_1, \ldots, \mathcal{V}_k\}$. The nodes of $G^\epsilon$ represent clusters of nodes from $G$, and the edge weight between two clusters $\mathcal{V}_i$ and $\mathcal{V}_j$ is given by $\frac{e_G(\mathcal{V}_i, \mathcal{V}_j)}{|\mathcal{V}_i||\mathcal{V}_j|}$. In this context, an important property of the irregularity $\mathrm{irreg}_G(\mathcal{P})$ is that it can be seen as the cut metric between the $G$ and a SBM based on $G^\epsilon$. For each node $i$ denote by $\mathcal{V}_{q_i} \in \mathcal{P}$ the partition set that contains the node. Given $G^\epsilon$, construct a new graph $G^{\mathcal{P}}$ with $N$ nodes, whose adjacency matrix $\boldsymbol{A}^{\mathcal{P}} = (a_{i,j}^{\mathcal{P}})_{i,j=1}^{|\mathcal{V}|}$ is defined by

$$a_{i,j}^{\mathcal{P}} = \frac{e_G(\mathcal{V}_{q_i}, \mathcal{V}_{q_j})}{|\mathcal{V}_{q_i}||\mathcal{V}_{q_j}|},$$

Let $\boldsymbol{A}$ be the adjacency matrix of $G$. It can be shown that

$$\left\| \boldsymbol{A} - \boldsymbol{A}^{\mathcal{P}} \right\|_{\square} = \mathrm{irreg}_G(\mathcal{P}),$$

which shows that the weak regularity lemma can be expressed in terms of cut norm rather than irregularity.

## C  GRAPHONS AND NORMS

**Kernel.**    A kernel $Y$ is a measurable function $Y : [0, 1]^2 \to [-1, 1]$.

**Graphon.**    A graphon (Borgs et al., 2008; Lovász, 2012) is a measurable function $W : [0, 1]^2 \to [0, 1]$. A graphon can be seen as a weighted graph, where the node set is the interval $[0, 1]$, and for any pair of nodes $x, y \in [0, 1]$, the weight of the edge between $x$ and $y$ is $W(x, y)$, which can also be seen as the probability of having an edge between $x$ and $y$. We note that in the standard definition a graphon is defined to be symmetric, but we remove this restriction in our construction.

**Kernel-signal and Graphon-signal.**    A kernel-signal is a pair $(Y, y)$ where $Y$ is a kernel and $y : [0, 1] \to \mathbb{R}^D$ is a measurable function. A graphon-signal is defined similarly with a graphon in place of a kernel.

**Induced graphon-signal.** Consider an interval equipartition $\mathcal{I}_m = \{I_1, \ldots, I_m\}$, a partition of $[0, 1]$ into disjoint intervals of equal length. Given a graph $G$ with an adjacency matrix $\boldsymbol{A}$, the induces graphon $W_{\boldsymbol{A}}$ is the graphon defined by $W_{\boldsymbol{A}}(x, y) = \boldsymbol{A}_{\lceil xm \rceil \lceil ym \rceil}$, where we use the convention that $\lceil 0 \rceil = 1$. Notice that $W_{\boldsymbol{A}}$ is a piecewise constant function on $\mathcal{I}_m \times \mathcal{I}_m$. As such, a graph of $m$ nodes can be identified by its induced graphon that is piecewise constant on $\mathcal{I}_m \times \mathcal{I}_m$.

## C.1 WEIGHTED FROBENIUS AND CUT NORM

**Weighted Frobenius norm.** Let $q : [0, 1]^2 \to [c, \infty)$ be a measurable function in $\mathcal{L}^\infty([0, 1]^2)$, where $c > 0$. We call such a $q$ a *weight function*. Consider the real weighted Lebesgue space $\mathcal{L}^2([0, 1]^2; q)$ defined with the inner product

$$\langle Y, Y' \rangle_q := \frac{1}{\|1\|_{1;q}} \iint_{[0,1]^2} Y(x, y) Y'(x, y) q(x, y) dx dy,$$

where $1$ is the constant function $[0, 1]^2 \ni (x, y) \mapsto 1$ and $\|1\|_{1;q} = \iint q$. When $q = 1$, we denote $\langle X, Z \rangle := \langle X, Z \rangle_1$. Let $\alpha, \beta > 0$. Consider the real Hilbert space $L^2([0, 1]^2; q) \times (L^2[0, 1])^D$ defined with the weighted inner product

$$\langle (Y, y), (Y', y') \rangle_q = \langle (Y, y), (Y', y') \rangle_{q, \alpha, \beta}$$

$$= \alpha \frac{1}{\|1\|_{1;q}} \iint_{[0,1]^2} Y(x, y) Y'(x, y) q(x, y) dx dy + \frac{\beta}{D} \sum_{j=1}^{D} \int_{[0,1]} y_j(x) y'_j(x) dx.$$

We call the corresponding weighted norm the *weighted Frobenius norm*, denoted by

$$\|(Y, y)\|_{F;q} = \|(Y, y)\|_{F;q,\alpha,\beta} = \sqrt{\alpha \|Y\|_{F;q}^2 + \frac{\beta}{D} \sum_{j=1}^{D} \|y_j\|_F^2},$$

where $\|Y\|_{F;q}^2 = \langle Y, Y \rangle_q$ and $\|y_j\|_F^2 = \langle y_j, y_j \rangle_1$.

Similarly, for a matrix-signal, we consider a *weight matrix* $\boldsymbol{Q} \in [c, \infty)^{N \times N}$, where $c > 0$. For $\boldsymbol{D}, \boldsymbol{D}' \in \mathbb{R}^{N \times N}$, define the weighted inner product by

$$\langle \boldsymbol{D}, \boldsymbol{D}' \rangle_{\boldsymbol{Q}} := \frac{1}{\|\mathbf{1}\|_{1;\boldsymbol{Q}}} \sum_{i,j \in [N]^2} d_{i,j} d'_{i,j} q_{i,j},$$

where $\mathbf{1} \in \mathbb{R}^{N \times N}$ is the matrix with all entries equal to 1, and $\|\mathbf{1}\|_{1;\boldsymbol{Q}} = \sum_{i,j \in [N]^2} q_{i,j}$. Define the *weighted matrix-signal Frobenius norm* by

$$\|(\boldsymbol{D}, \boldsymbol{Z})\|_{F;\boldsymbol{Q}} = \|(\boldsymbol{D}, \boldsymbol{Z})\|_{F;\boldsymbol{Q},\alpha,\beta} = \sqrt{\alpha \|\boldsymbol{D}\|_{F;\boldsymbol{Q}}^2 + \frac{\beta}{D} \sum_{j=1}^{D} \|z_j\|_F^2},$$

where $\|\boldsymbol{D}\|_{F;\boldsymbol{Q}}^2 = \langle \boldsymbol{D}, \boldsymbol{D} \rangle_{\boldsymbol{Q}}$ and $\|z_j\|_F^2 = \langle z_j, z_j \rangle_1$.

**Graphon weighted cut norm and cut metric.** Define for a kernel-signal $(Y, y)$ the *weighted cut norm*

$$\|(Y, y)\|_{\square;q,\alpha,\beta} = \|(Y, y)\|_{\square;q} = \frac{\alpha}{\|1\|_{1;q}} \sup_{\mathcal{U}, \mathcal{V}} \left| \int_{\mathcal{U}} \int_{\mathcal{V}} Y(x, y) q(x, y) dx dy \right| + \beta \frac{1}{D} \sum_{j=1}^{D} \sup_{\mathcal{U}} \left| \int_{\mathcal{U}} y_j(x) dx \right|,$$

where the supremum is over the set of measurable subsets $\mathcal{U}, \mathcal{V} \subset [0, 1]$.

The weighted cut metric between two graphon-signals $(W, f)$ and $(W', f')$ is defined to be $\|(W, f) - (W', f')\|_{\square;q}$.

**Graph-signal weighted cut norm and cut metric.** Define for a matrix-signal $(\boldsymbol{D}, \boldsymbol{Z})$, where $\boldsymbol{D} \in [-1,1]^{N \times N}$ and $\boldsymbol{Z} \in \mathbb{R}^{N \times D}$, the *weighted cut norm*

$$\|(\boldsymbol{D}, \boldsymbol{Z})\|_{\square;\boldsymbol{Q},\alpha,\beta} = \|(\boldsymbol{D}, \boldsymbol{Z})\|_{\square;\boldsymbol{Q}} = \alpha \|\boldsymbol{D}\|_{\square;\boldsymbol{Q}} + \beta \|\boldsymbol{Z}\|_{\square}$$

$$= \frac{\alpha}{\|\mathbf{1}\|_{1;\boldsymbol{Q}}} \max_{\mathcal{U},\mathcal{V} \subset [N]} \left| \sum_{i \in \mathcal{U}} \sum_{j \in \mathcal{V}} d_{i,j} q_{i,j} \right| + \frac{\beta}{DN} \sum_{j=1}^{D} \max_{\mathcal{U} \subset [N]} \left| \sum_{i \in \mathcal{U}} z_{i,j} \right|.$$

The weighted cut metric between two graph-signals $(\boldsymbol{A}, \boldsymbol{s})$ and $(\boldsymbol{A}', \boldsymbol{s}')$ is defined to be $\|(\boldsymbol{A}, \boldsymbol{s}) - (\boldsymbol{A}', \boldsymbol{s}')\|_{\square;\boldsymbol{Q}}$. We note that this metric gives a meaningful notion of graph-signal similarity for graphs as long as their number of edges satisfy $\|\mathbf{1}\|_{1;\boldsymbol{Q}} = \Theta(E)$[1]. All graphs with $E \ll \|\mathbf{1}\|_{1;\boldsymbol{Q}}$ have distance close to zero from each other, so the cut metric does not have a meaningful or useful separation power for such graphs.

The *weighted cut-metric* between two graphs $\boldsymbol{A}$ and $\boldsymbol{A}'$ represents the maximum (weighted) discrepancy in edge densities of $\boldsymbol{A}$ and $\boldsymbol{A}'$ across all blocks, giving the cut-metric a probabilistic interpretation. For simple graphs $\boldsymbol{A}, \boldsymbol{A}'$, the difference $\boldsymbol{A} - \boldsymbol{A}'$ is a granular function with values jumping between $-1$, $0$, and $1$. In such a case $\ell_p$ norms of $\boldsymbol{A} - \boldsymbol{A}'$ tend to be large. In contrast, the fact that the absolute value in (1) is outside the sum, unlike the $\ell_1$ norm, results in an averaging effect, which can lead to a small distance between $\boldsymbol{A}$ and $\boldsymbol{A}'$ in the cut distance even if $\boldsymbol{A} - \boldsymbol{A}'$ is granular.

**Densifying cut similarity.** In this paper, we will focus on a special construction of a weighted cut norm, which we construct and motivate next.

In graph completion tasks, such as link prediction or knowledge graph reasoning, the objective is to complete a partially observed adjacency matrix. Namely, there is a set of known dyads[2] $\mathcal{M} \subset [N]^2$, and the given data is the restriction of $\boldsymbol{A}$ to the known dyads

$$\boldsymbol{A}|_{\mathcal{M}} : \mathcal{M} \to \{0,1\}.$$

The goal is then to find an adjacency matrix $\boldsymbol{B}$ that fits $\boldsymbol{A}$ on the known dyads, namely, $\boldsymbol{B}|_{\mathcal{M}} \approx \boldsymbol{A}|_{\mathcal{M}}$, with the hope that $\boldsymbol{B}$ also approximates $\boldsymbol{A}$ on the unknown dyads due to some inductive bias.

Recall that $\mathcal{E}$ denotes the set of edges of $\boldsymbol{A}$. We call $\mathcal{E}^c = [N]^2 \setminus \mathcal{E}$ the set of *non-edges*. The training set in graph completion consists of the edges $\mathcal{E} \cap \mathcal{M}$ and the non-edges $\mathcal{E}^c \cap \mathcal{M}$. Typical methods, such as VGAE (Kipf & Welling, 2016) and TLC-GNN (Yan et al., 2021), define a loss of the form

$$l(\boldsymbol{B}) = \sum_{(n,m) \in \mathcal{M} \cap \mathcal{E}} c_{n,m} \psi_1(b_{n,m}, a_{n,m}) + \sum_{(n,m) \in \mathcal{M} \cap \mathcal{E}^c} c_{n,m} \psi_2(b_{n,m}, a_{n,m}),$$

where $\psi_1, \psi_2 : \mathbb{R}^2 \to \mathbb{R}_+$ are dyad-wise loss functions and $c_{n,m} \in \mathbb{R}_+$ are weights. Many methods, like RotateE (Sun et al., 2019), HousE (Li et al., 2022a) and NBFNet (Zhu et al., 2021), give one weight $c_{n,m} = C$ for edges $(n,m) \in \mathcal{E} \cap \mathcal{M}$ and a smaller weight $c_{n,m} = c \ll C$ for non-edges $(n,m) \in \mathcal{E}^c \cap \mathcal{M}$. The motivation is that for sparse graphs there are many more non-edges than edges, and giving the edges and non-edges the same weight would tend to produce learned $\boldsymbol{B}$ that does not put enough emphasis on the connectivity structure of $\boldsymbol{A}$. In practice, the smaller weight for non-edges is implemented implicitly by taking random samples from $\mathcal{M}$ during training, balancing the number of samples from $\mathcal{E} \cap \mathcal{M}$ and from $\mathcal{E}^c \cap \mathcal{M}$. The samples from $\mathcal{E}^c \cap \mathcal{M}$ are called *negative samples*.

**Remark C.1.** *In this paper we interpret such an approach as learning a* densified version *of* $\boldsymbol{A}$. *Namely, by putting less emphasis on non-edges, the matrix $\boldsymbol{B}$ roughly fits the structure of $\boldsymbol{A}$, but with a higher average degree.*

Motivated by the above discussion, we also define a densifying version of cut distance. Given a target unweighted adjacency matrix $\boldsymbol{A} = (a_{i,j})_{i,j=1}^{N}$ to be approximated, we consider the weight matrix $\boldsymbol{Q} = e\mathbf{1} + (1-e)\boldsymbol{A}$ for some small $e$ and $\mathbf{1}$ being the all 1 matrix. Denote the number of edges

---

[1]The asymptotic notation $a_n = \Theta(b_n)$ means that there exist positive constants $c_1, c_2$ and $n_0$ such that $c_1 b_n \leq a_n \leq c_2 b_n$ for all $n \geq n_0$. In our analysis, we suppose that there is a sequence of graphs with $N_n$ nodes, $E_n$ edges, and weight matrices $\boldsymbol{Q}_n$.

[2]A dyad is a pair of nodes $(m,n) \in [N]^2$. For a simple graph, a dyad may be an edge or a non-edge.

by $E = |\mathcal{E}|$. Next, we would like to choose $e$ to reflect some desired balance between edges and non-edges. Since the number of non-edges is $N^2 - E$ and the number of edges is $E$, we choose $e$ in such a way that $e - (E/N^2)e = \Gamma E/N^2$ for some $\Gamma > 0$, namely,

$$e = e_{E,\Gamma} = \frac{\Gamma E/N^2}{1 - (E/N^2)}. \tag{11}$$

The interpretation of $\Gamma$ is the proposition of sampled non-edges when compared with the edges. Namely, the weight matrix $Q = e + (1 - e)A$ effectively simulates taking $\Gamma E$ negative samples and $E$ samples. Observe that

$$\|A\|^2_{\mathrm{F};e\mathbf{1}+(1-e)A} = \frac{1}{\sum_{i,j\in[N]^2} q_{i,j}} \sum_{i,j\in[N]^2} a_{i,j} q_{i,j} = \frac{E/N^2}{e + (1-e)E/N^2} = 1/(\Gamma + 1).$$

To standardize the above similarity measure, we normalize it and define the weighted Frobenius norm $(1 + \Gamma)\|B\|_{\mathrm{F};Q_A}$ and weighted cut norm $(1 + \Gamma)\|B\|_{\square;Q_A}$, where

$$Q_A = Q_{A,\Gamma} := e_{E,\Gamma}\mathbf{1} + (1 - e_{E,\Gamma})A, \tag{12}$$

and where $e_{E,\Gamma}$ is defined in (11). We now have

$$(1 + \Gamma)\|A\|_{\mathrm{F};Q_A} = 1. \tag{13}$$

This standardization assures that merely increasing $\Gamma$ in the definition of the cut metric $(1 + \Gamma)\|A - B\|_{\mathrm{F};Q_A}$ would not lead to a seemingly better approximation. The above discussion leads to the following definition.

**Definition C.1.** *Let $A \in \{0,1\}^{N\times N}$ be an unweighted adjacency matrix, and $\Gamma > 0$. The* densifying cut similarity *between the target $A$ and any adjacency matrix $B \in \mathbb{R}^{N\times N}$ is defined to be*

$$\sigma_\square(A\|B) = \sigma_{\square;\Gamma}(A\|B) := (1 + \Gamma)\|A - B\|_{\square;Q_A},$$

*where $Q_A$ is defined in (12). Given $\alpha, \beta > 0$ such that $\alpha + \beta = 1$, the* densifying cut similarity *between the target graph-signal $(A, X)$ and the graph-signal $(A', X')$ is defined to be*

$$\sigma_\square\big((A,X)\|(A',X')\big) = \sigma_{\square;\alpha,\beta,\Gamma}\big((A,X)\|(A',X')\big) := \alpha\sigma_{\square;\Gamma}(A\|B) + \beta\|X - X'\|_\square.$$

We moreover note that the similarity measure $\sigma_\square(A\|B)$ is not symmetric, and hence not a metric. The first entry $A$ in $\sigma_\square(A\|B)$ is interpreted as the thing to be approximated, and the second entry $B$ as the approximant. Here, when fitting an IBG to a graph, $A$ is a constant, and $B$ is the variable.

# D  PROOF OF THE SEMI-CONSTRUCTIVE DENSIFYING DIRECTIONAL SOFT WEAK REGULARITY LEMMA

In this section we prove a version of the constructive weak regularity lemma for asymmetric graphon signals. Prior information regarding cut-distance, the original formulation of the weak regularity lemma and it's constructive version for symmetric graphon-signals can be found in (Finkelshtein et al., 2024a, Appendix A, B).

## D.1  INTERSECTING BLOCK GRAPHONS

Below, we extend the definition of IBGs for graphons. The construction is similar to the one in Appendix B.3 of Finkelshtein et al. (2024a), where ICGs are extended to graphons. Denote by $\chi$ the set of all indicator functions of measurable subset of $[0, 1]$

$$\chi = \{\mathbb{1}_u \mid u \subset [0, 1] \text{ measurable}\}.$$

**Definition D.1.** *A set $\mathcal{Q}$ of bounded measurable functions $q : [0, 1] \to \mathbb{R}$ that contains $\chi$ is called a* soft affiliation model.

For the case of node level graphon-signals, we use the following definition:

**Definition D.2.** *Let $D \in \mathbb{N}$. Given a soft affiliation model $\mathcal{Q}$, the subset $[\mathcal{Q}]$ of $L^2[0,1]^2 \times (L^2[0,1])^D$ of all elements of the form $(au(x)v(y), bu(z) + cv(z))$, with $u, v \in \mathcal{Q}$, $a \in \mathbb{R}$ and $b, c \in \mathbb{R}^D$, is called the* soft rank-1 intersecting block graphon (IBG) model *corresponding to $\mathcal{Q}$. Given $K \in \mathbb{N}$, the subset $[\mathcal{Q}]_K$ of $L^2[0,1]^2 \times (L^2[0,1])^D$ of all linear combinations of $K$ elements of $[\mathcal{Q}]$ is called the* soft rank-$K$ IBG model *corresponding to $\mathcal{Q}$. Namely, $(C, p) \in [\mathcal{Q}]_K$ if and only if it has the form*

$$C(x,y) = \sum_{k=1}^{K} a_k u_k(x) v_k(y) \quad and \quad p(z) = \sum_{k=1}^{K} b_k u_k(z) + c_k v_k(z)$$

*where $(u_k)_{k=1}^{K} \in \mathcal{Q}^K$ are called the* target community affiliation functions, *$(v_k)_{k=1}^{K} \in \mathcal{Q}^K$ are called the* source community affiliation functions, *$(a_k)_{k=1}^{K} \in \mathbb{R}^K$ are called the* community affiliation magnitudes, *$(b_k)_{k=1}^{K} \in \mathbb{R}^{K \times D}$ are called the* target community features, *and $(c_k)_{k=1}^{K} \in \mathbb{R}^{K \times D}$ the* source community features. *Any element of $[\mathcal{Q}]_K$ is called an intersecting block graphon-signal (IBG).*

### D.2 THE SEMI-CONSTRUCTIVE WEAK REGULARITY LEMMA IN HILBERT SPACE

In this subsection we prove the constructive weak graphon-signal regularity lemma.

László Miklós Lovász (2007) extended the weak regularity lemma to graphons. They showed that the lemma follows from a more general result about approximation in Hilbert spaces – the weak regularity lemma in Hilbert spaces (László Miklós Lovász, 2007, Lemma 4). We extend this result to have a constructive form, which we later use to prove Theorem 4.1. For completeness, we begin by stating the original weak regularity lemma in Hilbert spaces from (László Miklós Lovász, 2007).

**Lemma D.1** ((László Miklós Lovász, 2007)). *Let $\mathcal{K}_1, \mathcal{K}_2, \ldots$ be arbitrary nonempty subsets (not necessarily subspaces) of a real Hilbert space $\mathcal{H}$. Then, for every $\epsilon > 0$ and $g \in \mathcal{H}$ there is $m \leq \lceil 1/\epsilon^2 \rceil$ and $(f_i \in \mathcal{K}_i)_{i=1}^{m}$ and $(\gamma_i \in \mathbb{R})_{i=1}^{m}$, such that for every $w \in \mathcal{K}_{m+1}$*

$$\left| \left\langle w, g - \left( \sum_{i=1}^{m} \gamma_i f_i \right) \right\rangle \right| \leq \epsilon \, \|w\| \, \|g\| .$$

Finkelshtein et al. (2024a) introduced a version of Lemma D.1 (Lemma B.3 therein) with a "more constructive flavor." They provide a result in which the approximating vector $\sum_{i=1}^{m} \gamma_i f_i$ is given as the solution to a "manageable" optimization problem, whereas the original lemma in (László Miklós Lovász, 2007) only proves the existence of the approximating vector. Below, we give a similar result to (Finkelshtein et al., 2024a, Lemma B.3.), where the constructive aspect is further improved. While Finkelshtein et al. (2024a) showed that the optimization problems leads to an approximate minimizer in high probability, they did not provide a way to evaluate if indeed this "good" event of high probability occurred. In contrast, we formulate this lemma in such a way that leads to a deterministic approach for checking whether the good event happened. In the discussion after Theorem D.2, we explain this in detail.

**Lemma D.2.** *Let $\{\mathcal{K}_j\}_{j \in \mathbb{N}}$ be a sequence of nonemply subsets of a real Hilbert space $\mathcal{H}$. Let $K \in \mathbb{N}$, $\delta \geq 0$, let $R \geq 1$ such that $K/R \in \mathbb{N}$, let $\delta > 0$, and let $g \in \mathcal{H}$. For every $k \in \mathbb{N}$, let*

$$\eta_k = (1 + \delta) \inf_{\boldsymbol{\kappa}, \mathbf{h}} \left\| g - \sum_{i=1}^{k} \kappa_i h_i \right\|^2$$

*where the infimum is over $\boldsymbol{\kappa} = \{\kappa_1, \ldots, \kappa_k\} \in \mathbb{R}^k$ and $\mathbf{h} = \{h_1, \ldots, h_k\} \in \mathcal{K}_1 \times \ldots \times \mathcal{K}_k$. Then,*

*1. For every $m \in \mathbb{N}$, any vector of the form*

$$g^* = \sum_{j=1}^{m} \gamma_j f_j \quad such \ that \quad \boldsymbol{\gamma} = (\gamma_j)_{j=1}^{m} \in \mathbb{R}^m \quad and \quad \mathbf{f} = (f_j)_{j=1}^{m} \in \mathcal{K}_1 \times \ldots \times \mathcal{K}_m \quad (14)$$

*that gives a close-to-best Hilbert space approximation of $g$ in the sense that*

$$\|g - g^*\| \leq \eta_m, \tag{15}$$

*also satisfies*

$$\forall w \in \mathcal{K}_{m+1}, \quad |\langle w, g - g^* \rangle| \leq \|w\| \sqrt{\eta_m - \frac{\eta_{m+1}}{1 + \delta}}. \tag{16}$$

2. *If $m$ is uniformly randomly sampled from $[K]$, then in probability $1 - \frac{1}{R}$ (with respect to the choice of $m$),*

$$\|w\| \sqrt{\eta_m - \frac{\eta_{m+1}}{1+\delta}} \le \|w\| \|g\| \sqrt{\delta + \frac{R(1+\delta)}{K}}. \tag{17}$$

*Specifically, in probability $1 - \frac{1}{R}$, any vector of the form (14) which satisfy (15), also satisfies*

$$\forall w \in \mathcal{K}_{m+1}, \quad |\langle w, g - g^* \rangle| \le \|w\| \|g\| \sqrt{\delta + \frac{R(1+\delta)}{K}}. \tag{18}$$

The lemma is used as follows. We choose $m$ at random. We know by Item 2 that in high probability the approximation is good (i.e. (18) is satisfied), but we are not certain. For certainty, we use the deterministic bound (16), which gives a certificate for a specific $m$. Namely, given a realization of $m$, we can estimate the right-hand-side of (16), which is also the left-hand-side of the probabilistic bound (17). For that, we find the $(m+1)$'th error $\eta_{m+1}$, solving another optimization problem, and verify that (17) is satisfied for $m$. If it is not, we resample $m$ and repeat. The expected number of times we need to repeat this until we get a small error is $R/(R-1)$.

Hence, under an assumption that the we can find a close to optimum $\|g - g^*\|$ for a given $m$ in $T_K$ operations, we can find in probability 1 a vector $g^*$ in the span of $\mathcal{K}_1, \dots, \mathcal{K}_K$ that solves (18) with expected number of operations $T_K R/(R-1)$.

*Proof of Lemma D.2.* Let $K > 0$. Let $R \ge 1$ such that $K/R \in \mathbb{N}$. For every $k$, let

$$\eta_k = (1 + \delta) \inf_{\boldsymbol{\kappa}, \mathbf{h}} \|g - \sum_{i=1}^{k} \kappa_i h_i\|^2$$

where the infimum is over $\boldsymbol{\kappa} = \{\kappa_1, \dots, \kappa_k\} \in \mathbb{R}^k$ and $\mathbf{h} = \{h_1, \dots, h_k\} \in \mathcal{K}_1 \times \dots \times \mathcal{K}_k$.

Note that every

$$g^* = \sum_{j=1}^{m} \gamma_j f_j \tag{19}$$

that satisfies

$$\|g - g^*\|^2 \le \eta_m$$

also satisfies: for any $w \in \mathcal{K}_{m+1}$ and every $t \in \mathbb{R}$,

$$\|g - (g^* + tw)\|^2 \ge \frac{\eta_{m+1}}{1+\delta} = \frac{\eta_m + \eta_{m+1} - \eta_m}{1+\delta} \ge \frac{\|g - g^*\|^2}{1+\delta} - \frac{\eta_m - \eta_{m+1}}{1+\delta}.$$

This can be written as

$$\forall t \in \mathbb{R}, \quad \|w\|^2 t^2 + 2 \langle w, g - g^* \rangle t + \frac{\eta_m - \eta_{m+1}}{1+\delta} + (1 - \frac{1}{1+\delta}) \|g - g^*\|^2 \ge 0. \tag{20}$$

The discriminant of this quadratic polynomial is

$$4 \langle w, g - g^* \rangle^2 - 4 \|w\|^2 \left( \frac{\eta_m - \eta_{m+1}}{1+\delta} + (1 - \frac{1}{1+\delta}) \|g - g^*\|^2 \right)$$

and it must be non-positive to satisfy the inequality (20), namely

$$4 \langle w, g - g^* \rangle^2 \le 4 \|w\|^2 \left( \frac{\eta_m - \eta_{m+1}}{1+\delta} + (1 - \frac{1}{1+\delta}) \|g - g^*\|^2 \right) \le 4 \|w\|^2 \left( \frac{\eta_m - \eta_{m+1}}{1+\delta} + (1 - \frac{1}{1+\delta}) \eta_m \right)$$

$$= 4 \|w\|^2 \left( \eta_m - \frac{\eta_{m+1}}{1+\delta} \right),$$

which proves

$$|\langle w, g - g^* \rangle| \le \|w\| \sqrt{\eta_m - \frac{\eta_{m+1}}{1+\delta}},$$

which proves Item 1.

For Item 2, note that $\|g\|^2 \geq \frac{\eta_1}{1+\delta} \geq \frac{\eta_2}{1+\delta} \geq \ldots \geq 0$. Therefore, there is a subset of at least $(1 - \frac{1}{R})K + 1$ indices $m$ in $[K]$ such that $\eta_m \leq \eta_{m+1} + \frac{R(1+\delta)}{K}\|g\|^2$. Otherwise, there are $\frac{K}{R}$ indices $m$ in $[K]$ such that $\eta_{m+1} < \eta_m - \frac{R(1+\delta)}{K}\|g\|^2$, which means that

$$\eta_K < \eta_1 - \frac{K}{R}\frac{R(1+\delta)}{K}\|g\|^2 \leq (1+\delta)\|g\|^2 - (1+\delta)\|g\|^2 = 0,$$

which is a contradiction to the fact that $\eta_K \geq 0$. Hence, there is a set $\mathcal{M} \subseteq [K]$ of $(1 - \frac{1}{R})K$ indices such that for every $m \in \mathcal{M}$,

$$\|w\|\sqrt{\eta_m - \frac{\eta_{m+1}}{1+\delta}} \leq \|w\|\sqrt{\eta_{m+1} + \frac{R(1+\delta)}{K}\|g\|^2 - \frac{\eta_{m+1}}{1+\delta}}$$

$$\leq \|w\|\sqrt{\frac{\delta}{1+\delta}\eta_{m+1} + \frac{R(1+\delta)}{K}\|g\|^2} \leq \|w\|\sqrt{\delta\|g\|^2 + \frac{R(1+\delta)}{K}\|g\|^2}$$

$$= \|w\|\|g\|\sqrt{\delta + \frac{R(1+\delta)}{K}}$$

$\square$

## D.3 THE DENSIFYING SEMI-CONSTRUCTIVE GRAPHON-SIGNAL WEAK REGULARITY LEMMA

Define for kernel-signal $(V, f)$ the *densifying cut distance*

$$\|(V, f)\|_{\square;q} = \frac{\alpha}{\iint q}\sup_{U,V}\left|\int_U\int_V V(x,y)q(x,y)dxdy\right| + \beta\frac{1}{D}\sum_{j=1}^{D}\sup_U\left|\int_U f_j(x)dx\right|.$$

Below we give a version of Theorem 4.1 for intersecting block graphons.

**Theorem D.3.** *Let $(W, s)$ be a graphon-signal, $K \in \mathbb{N}$, $\delta > 0$, and let $\mathcal{Q}$ be a soft indicators model. Let $q$ be a weight function and $\alpha, \beta > 0$. Let $R \geq 1$ such that $K/R \in \mathbb{N}$. Consider the graphon-signal Frobenius norm with weight $\|(Y, y)\|_{\mathrm{F};q} = \|(Y, y)\|_{\mathrm{F};q,\alpha,\beta}$, and cut norm with weight $\|(Y, y))\|_{\square;q} := \|(Y, y)\|_{\square;q,\alpha,\beta}$. For every $k \in \mathbb{N}$, let*

$$\eta_k = (1+\delta)\inf_{(C,p)\in[\mathcal{Q}]_k}\|(W, s) - (C, p)\|_{\mathrm{F};q}^2.$$

*Then,*

1. *For every $m \in \mathbb{N}$, any IBG $(C^*, p^*) \in [\mathcal{Q}]_m$ that gives a close-to-best weighted Frobenius approximation of $(W, s)$ in the sense that*

$$\|(W, s) - (C^*, p^*)\|_{\mathrm{F};q}^2 \leq \eta_m, \tag{21}$$

   *also satisfies*

$$\|(W, s) - (C^*, p^*)\|_{\square;q} \leq (\sqrt{\alpha} + \sqrt{\beta})\sqrt{\eta_m - \frac{\eta_{m+1}}{1+\delta}}.$$

2. *If $m$ is uniformly randomly sampled from $[K]$, then in probability $1 - \frac{1}{R}$ (with respect to the choice of $m$),*

$$\sqrt{\eta_m - \frac{\eta_{m+1}}{1+\delta}} \leq \sqrt{\alpha\|W\|_{\mathrm{F};q}^2 + \beta\|s\|_{\mathrm{F}}^2}\sqrt{\delta + \frac{R(1+\delta)}{K}}. \tag{22}$$

   *Specifically, in probability $1 - \frac{1}{R}$, any $(C^*, p^*) \in [\mathcal{Q}]_m$ which satisfy (21), also satisfies*

$$\|(W, s) - (C^*, p^*)\|_{\square;q} \leq (\sqrt{\alpha} + \sqrt{\beta})\left(\sqrt{\alpha\|W\|_{\mathrm{F},q}^2 + \beta\|s\|_{\mathrm{F}}^2}\sqrt{\delta + \frac{R(1+\delta)}{K}}\right). \tag{23}$$

Theorem D.3 is similar to the semi-constructive weak regularity lemma of Finkelshtein et al. (2024a, Theorem B.1). However, our result extends the result of Finkelshtein et al. (2024a) by providing a deterministic certificate for the approximation quality, as we explained in the discusson after Lemma D.2, extending the cut-norm to the more general weighted cut norm, and extending to general non-symmetric graphons.

*Proof of Theorem D.3.* Let us use Lemma D.2, with $\mathcal{H} = L^2([0,1]^2; q) \times (L^2[0,1])^D$ with the weighted inner product

$$\langle (V, y), (V', y') \rangle_q = \alpha \frac{1}{\|1\|_{1;q}} \iint_{[0,1]^2} V(x,y) V'(x,y) q(x,y) dx dy + \beta \sum_{j=1}^{D} \int_{[0,1]} y_j(x) y_j'(x) dx,$$

and corresponding norm denoted by $\|(Y, y)\|_{\mathrm{F};q} = \sqrt{\alpha \|Y\|_{\mathrm{F};q}^2 + \beta \sum_{j=1}^{D} \|y_j\|_{\mathrm{F}}^2}$, and $\mathcal{K}_j = [\mathcal{Q}]$. Note that the Hilbert space norm is the Frobenius norm in this case. Let $m \in \mathbb{N}$. In the setting of the lemma, we take $g = (W, s)$, and $g^* \in [\mathcal{Q}]_m$. By the lemma, any approximate Frobenius minimizer $(C^*, p^*)$, namely, that satisfies $\|(W, s) - (C^*, p^*)\|_{\mathrm{F};q} \leq \eta_m$, also satisfies

$$\langle (T, y), (W, s) - (C^*, p^*) \rangle_q \leq \|(T, y)\|_{\mathrm{F};q} \sqrt{\eta_m - \frac{\eta_{m+1}}{1+\delta}}$$

for every $(T, y) \in [\mathcal{Q}]$.

Hence, for every choice of measurable subsets $\mathcal{S}, \mathcal{T} \subset [0,1]$, we have

$$\frac{1}{\|1\|_{1;q}} \left| \int_{\mathcal{S}} \int_{\mathcal{T}} (W(x,y) - C^*(x,y)) q(x,y) dx dy \right|$$

$$= \left| \frac{1}{\alpha} \langle (\mathbb{1}_{\mathcal{S}} \otimes \mathbb{1}_{\mathcal{T}}, 0), (W, s) - (C^*, p^*) \rangle_q \right|$$

$$\leq \frac{1}{\alpha} \|(\mathbb{1}_{\mathcal{S}} \otimes \mathbb{1}_{\mathcal{T}}, 0)\|_{\mathrm{F};q} \sqrt{\eta_m - \frac{\eta_{m+1}}{1+\delta}}$$

$$\leq \frac{1}{\alpha} \sqrt{\alpha} \sqrt{\eta_m - \frac{\eta_{m+1}}{1+\delta}}$$

Hence, taking the supremum over $\mathcal{S}, \mathcal{T} \subset [0,1]$, we also have

$$\alpha \|W - C^*\|_{\square;q} \leq \sqrt{\alpha} \sqrt{\eta_m - \frac{\eta_{m+1}}{1+\delta}}.$$

Now, for $n$ randomly uniformly sampled from $[K]$, consider the event $\mathcal{M}$ (regarding the uniform choice of $n$) of probability $(1 - 1/R)$ in which

$$\sqrt{\eta_n - \frac{\eta_{n+1}}{1+\delta}} \leq \sqrt{\alpha \|W\|_{\mathrm{F};q}^2 + \beta \|s\|_{\mathrm{F}}^2} \sqrt{\delta + \frac{R(1+\delta)}{K}}.$$

Hence, in the event $\mathcal{M}$, we also have

$$\alpha \|W - C^*\|_{\square;q} \leq \sqrt{\alpha^2 \|W\|_{\mathrm{F};q}^2 + \alpha\beta \|s\|_{\mathrm{F}}} \sqrt{\delta + \frac{R(1+\delta)}{K}}.$$

Similarly, for every measurable $\mathcal{T} \subset [0,1]$ and every standard basis element $\boldsymbol{b} = (\delta_{j,i})_{i=1}^{D}$ for any $j \in [D]$,

$$\left| \int_{\mathcal{T}} (s_j(x) - p_j^*(x)) dx \right|$$

$$= \left| \frac{1}{\beta} \langle (0, \boldsymbol{b} \mathbb{1}_{\mathcal{T}}), (W, s) - (C^*, p^*) \rangle_q \right|$$

$$\leq \frac{1}{\beta} \|(0, \boldsymbol{b} \mathbb{1}_{\mathcal{T}})\|_{\mathrm{F};q} \sqrt{\eta_m - \frac{\eta_{m+1}}{1+\delta}}$$

$$\leq \frac{\sqrt{\beta}}{\beta} \sqrt{\eta_m - \frac{\eta_{m+1}}{1+\delta}},$$

so, taking the supremum over $\mathcal{T} \subset [0,1]$ independently for every $j \in [D]$, and averaging over $j \in [D]$, we get

$$\beta \|s - p^*\|_\square \leq \sqrt{\beta}\sqrt{\eta_m - \frac{\eta_{m+1}}{1+\delta}}.$$

Now, for the same event $\mathcal{M}$ as above regarding the choice of $n \in [K]$,

$$\beta \|s - p^*\|_\square \leq \sqrt{\alpha\beta\|W\|_{\mathrm{F};q}^2 + \beta^2 \|s\|_\mathrm{F}^2}\sqrt{\delta + \frac{R(1+\delta)}{K}}.$$

Overall, we get for every $m$ and corresponding approximately optimum $(C^*, p^*)$,

$$\|(W, s) - (C^*, p^*)\|_{\square;q} \leq (\sqrt{\alpha} + \sqrt{\beta})\sqrt{\eta_m - \frac{\eta_{m+1}}{1+\delta}}.$$

Moreover, for uniformly sampled $n \in [K]$, in probability more than $1 - 1/R$,

$$\|(W, s) - (C^*, p^*)\|_{\square;q} \leq (\sqrt{\alpha} + \sqrt{\beta})\Big(\sqrt{\alpha\|W\|_{\mathrm{F},q}^2 + \beta \|s\|_\mathrm{F}^2}\sqrt{\delta + \frac{R(1+\delta)}{K}}.$$

$\square$

## D.4 PROOF OF THE SEMI-CONSTRUCTIVE DENSIFYING WEAK REGULARITY LEMMA

Next, we show that Theorem D.3 reduces to Theorem 4.1 in the case of graphon-signals induced by graph-signals.

**Theorem 4.1.** *Let $(A, X)$ be a graph-signal, $K \in \mathbb{N}$, $\delta > 0$, and let $\mathcal{Q}$ be a soft indicators model. Let $\alpha, \beta > 0$ such that $\alpha + \beta = 1$. Let $\Gamma > 0$ and let $Q_A$ be the weight matrix defined in Definition 3.2. Let $R \geq 1$ such that $K/R \in \mathbb{N}$. For every $k \in \mathbb{N}$, let*

$$\eta_k = (1 + \delta)\min_{(C,P)\in[\mathcal{Q}]_k} \|(A, X) - (C, P)\|_{\mathrm{F};Q_A,\alpha(1+\Gamma),\beta}^2$$

*Then,*

1. *For every $m \in \mathbb{N}$, any IBG $(C^*, P^*) \in [\mathcal{Q}]_m$ that gives a close-to-best weighted Frobenius approximation of $(A, X)$ in the sense that*

$$\|(A, X) - (C^*, P^*)\|_{\mathrm{F};Q_A,\alpha(1+\Gamma),\beta}^2 \leq \eta_m, \tag{24}$$

   *also satisfies*

$$\sigma_{\square;\alpha,\beta,\Gamma}\big((A, X)\|(C^*, P^*)\big) \leq (\sqrt{\alpha(1+\Gamma)} + \sqrt{\beta})\sqrt{\eta_m - \frac{\eta_{m+1}}{1+\delta}}. \tag{25}$$

2. *If $m$ is uniformly randomly sampled from $[K]$, then in probability $1 - \frac{1}{R}$ (with respect to the choice of $m$),*

$$\sqrt{\eta_m - \frac{\eta_{m+1}}{1+\delta}} \leq \sqrt{\delta + \frac{R(1+\delta)}{K}} \tag{26}$$

   *Specifically, in probability $1 - \frac{1}{R}$, any $(C^*, P^*) \in [\mathcal{Q}]_m$ which satisfy (24), also satisfies*

$$\sigma_{\square;\alpha,\beta,\Gamma}\big((A, X) - (C^*, P^*)\big) \leq \big(\sqrt{2+\Gamma}\big)\sqrt{\delta + \frac{R(1+\delta)}{K}}. \tag{27}$$

In practice, Theorem 4.1 is used to motivate the following computational approach for approximating graph-signals by IBGs. We suppose that there is an oracle optimization method that can solve (24) in $T_K$ operations whenever $m \leq K$. In practice, we use gradient descent on the left-hand side of (24), which takes $O(E)$ operations as shown in Proposition 4.2. The oracle is used as follows. We choose $m \in [K]$ at random. We know by Item 2 of Theorem 4.1 that in high probability the good approximation bound (27) is satisfied, but we are not certain. For certainty, we use Item 1 of Theorem 4.1. Given our specific realization of $m$, we can estimate the right-hand-side of (25) by our oracle optimization method in $2T_K$ operations, and verify that the right-hand-side of (25) is less than the right-hand-side of (27). If it is not (in probability $1/R$), we resample $m$ and repeat. The expected number of times we need to repeat this process until we get a small error is $R/(R-1)$, so the expected time it takes the algorithm to find an IBG with error bound (27) is $2T_K R/(R-1)$.

*Proof of Theorem 4.1.* Let $\boldsymbol{Q_A}$ be the weight matrix defined in (12). Consider the following identities between the weighted Frobenius norm of induced graphon-signals and the Frobenius norm of the graph-signal, and, a similar identity for the densifying cut similarity.

$$
\begin{aligned}
\|(W_{\boldsymbol{A}}, s_{\boldsymbol{X}}) - (W_{\boldsymbol{C}}, W_{\boldsymbol{P}})\|^2_{\mathrm{F}; W_{\boldsymbol{Q_A}}, \alpha(1+\Gamma), \beta} &= \|(\boldsymbol{A}, \boldsymbol{X}) - (\boldsymbol{C}, \boldsymbol{P})\|^2_{\mathrm{F}; \boldsymbol{Q_A}, \alpha(1+\Gamma), \beta}, \\
\sigma_{\square, \alpha, \beta}\big((W_{\boldsymbol{A}}, s_{\boldsymbol{X}}) \| (W_{\boldsymbol{C}}, s_{\boldsymbol{P}})\big) &= \sigma_{\square, \alpha, \beta}\big((\boldsymbol{A}, \boldsymbol{X}) \| (\boldsymbol{C}, \boldsymbol{P})\big).
\end{aligned}
\tag{28}
$$

We apply Theorem D.3 on the weighted Frobenius and cut norms with weight $Q = W_{\boldsymbol{Q_A}}$. We immediately obtain (25) from (28). For (26), by (22) of Theorem D.3, and by (13) and by the fact that signals have values in $[-1, 1]$,

$$
\sqrt{\eta_m - \frac{\eta_{m+1}}{1 + \delta}} \le \sqrt{\alpha(1+\Gamma)\|W_{\boldsymbol{A}}\|^2_{\mathrm{F}; W_{\boldsymbol{Q_A}}} + \beta \|s_{\boldsymbol{A}}\|^2_{\mathrm{F}}} \sqrt{\delta + \frac{R(1+\delta)}{K}}
$$

$$
\le \sqrt{\delta + \frac{R(1+\delta)}{K}}.
$$

Lastly, (27) follows the fact that for $\alpha, \beta > 0$ such that $\alpha + \beta = 1$, we must have $(\sqrt{\alpha(1+\Gamma)} + \sqrt{\beta}) \le \sqrt{2 + \Gamma}$. $\square$

## E  FITTING IBGS TO GRAPHS EFFICIENTLY

Below we present the proof of Theorem 4.2. The proof follows the lines of the proof of Proposition 4.1 in (Finkelshtein et al., 2024a). We restate the proposition below for the benefit of the reader.

**Proposition 4.2.** *Let $\boldsymbol{A} = (a_{i,j})_{i,j=1}^N$ be an adjacency matrix of an unweighted graph with $E$ edges. The graph part of the sparse Frobenius loss can be written as*

$$
\begin{aligned}
\left\|\boldsymbol{A} - \boldsymbol{U} \operatorname{diag}(\boldsymbol{r}) \boldsymbol{V}^\top\right\|^2_{\mathrm{F}; \boldsymbol{Q_A}} &= \|A\|^2_{\mathrm{F}; \boldsymbol{Q_A}} + \frac{e}{(1+\Gamma)E} \operatorname{Tr}\big((\boldsymbol{V}^\top \boldsymbol{V}) \operatorname{diag}(\boldsymbol{r})(\boldsymbol{U}^\top \boldsymbol{U}) \operatorname{diag}(\boldsymbol{r})\big) \\
&\quad - \frac{2}{(1+\Gamma)E} \sum_{i=1}^N \sum_{j \in \mathcal{N}(i)} \boldsymbol{U}_{i,:} \operatorname{diag}(\boldsymbol{r}) \left(\boldsymbol{V}^\top\right)_{:,j} a_{i,j} \\
&\quad + \frac{1 - e}{(1+\Gamma)E} \sum_{i=1}^N \sum_{j \in \mathcal{N}(i)} (\boldsymbol{U}_{i,:} \operatorname{diag}(\boldsymbol{r}) \left(\boldsymbol{V}^\top\right)_{:,j})^2
\end{aligned}
$$

*where $\boldsymbol{Q_A}$ is defined in Equation* (2). *Computing the right-hand-side and its gradients with respect to $\boldsymbol{U}$, $\boldsymbol{V}$ and $\boldsymbol{r}$ has a time complexity of $\mathcal{O}(K^2 N + KE)$, and a space complexity of $\mathcal{O}(KN + E)$.*

The proof is similar to that of Proposition 4.1 in (Finkelshtein et al., 2024a), while applying the necessary changes under the new weighted Frobenius norm and the structure of IBGs.

*Proof.* The loss can be expressed as

$$
\begin{aligned}
\left\|\boldsymbol{A} - \boldsymbol{U} \operatorname{diag}(\boldsymbol{r}) \boldsymbol{V}^\top\right\|^2_{\mathrm{F}; \boldsymbol{Q_A}} &= \frac{1}{(1+\Gamma)E} \sum_{i=1}^N \Bigg( \sum_{j \in \mathcal{N}(i)} \Big(a_{i,j} - \boldsymbol{U}_{i,:} \operatorname{diag}(\boldsymbol{r}) \left(\boldsymbol{V}^\top\right)_{:,j}\Big)^2 + \\
&\qquad\qquad + \sum_{j \notin \mathcal{N}(i)} e \left(-\boldsymbol{U}_{i,:} \operatorname{diag}(\boldsymbol{r}) \left(\boldsymbol{V}^\top\right)_{:,j}\right)^2 \Bigg) \\
&= \frac{1}{(1+\Gamma)E} \sum_{i=1}^N \Bigg( \sum_{j \in \mathcal{N}(i)} \Big(a_{i,j} - \boldsymbol{U}_{i,:} \operatorname{diag}(\boldsymbol{r}) \left(\boldsymbol{V}^\top\right)_{:,j}\Big)^2 + \sum_{j=1}^N e \left(\boldsymbol{U}_{i,:} \operatorname{diag}(\boldsymbol{r}) \left(\boldsymbol{V}^\top\right)_{:,j}\right)^2 \\
&\qquad - \sum_{j \in \mathcal{N}(i)} e \left(\boldsymbol{U}_{i,:} \operatorname{diag}(\boldsymbol{r}) \left(\boldsymbol{V}^\top\right)_{:,j}\right)^2 \Bigg)
\end{aligned}
$$

We expand the quadratic term $\left(a_{i,j} - \boldsymbol{U}_{i,:} \operatorname{diag}(\boldsymbol{r}) \left(\boldsymbol{V}^\top\right)_{:,j}\right)^2$, and get

$$\left\|\boldsymbol{A} - \boldsymbol{U} \operatorname{diag}(\boldsymbol{r})\boldsymbol{V}^\top\right\|_{\mathrm{F};\boldsymbol{Q_A}}^2 = \frac{e}{(1+\Gamma)\,E} \sum_{i,j=1}^{N} \left(\boldsymbol{U}_{i,:} \operatorname{diag}(\boldsymbol{r}) \left(\boldsymbol{V}^\top\right)_{:,j}\right)^2 +$$

$$+ \frac{1}{(1+\Gamma)\,E} \sum_{i=1}^{N} \sum_{j \in \mathcal{N}(i)} \left(a_{i,j}^2 - 2\boldsymbol{U}_{i,:} \operatorname{diag}(\boldsymbol{r}) \left(\boldsymbol{V}^\top\right)_{:,j} a_{i,j}\right)$$

$$+ \frac{1-e}{(1+\Gamma)\,E} \sum_{i=1}^{N} \sum_{j \in \mathcal{N}(i)} \left(\boldsymbol{U}_{i,:} \operatorname{diag}(\boldsymbol{r}) \left(\boldsymbol{V}^\top\right)_{:,j}\right)^2$$

$$= \frac{eN^2}{(1+\Gamma)\,E} \left\|\boldsymbol{U} \operatorname{diag}(\boldsymbol{r})\boldsymbol{V}^\top\right\|_{\mathrm{F}}^2 +$$

$$+ \frac{1}{(1+\Gamma)\,E} \sum_{i,j=1}^{N} a_{i,j}^2 - \frac{2}{(1+\Gamma)\,E} \sum_{i=1}^{N} \sum_{j \in \mathcal{N}(i)} \boldsymbol{U}_{i,:} \operatorname{diag}(\boldsymbol{r}) \left(\boldsymbol{V}^\top\right)_{:,j} a_{i,j}$$

$$+ \frac{1-e}{(1+\Gamma)\,E} \sum_{i=1}^{N} \sum_{j \in \mathcal{N}(i)} \left(\boldsymbol{U}_{i,:} \operatorname{diag}(\boldsymbol{r}) \left(\boldsymbol{V}^\top\right)_{:,j}\right)^2$$

$$= \frac{e}{(1+\Gamma)\,E} \operatorname{Tr}\left(\boldsymbol{V}^\top\boldsymbol{V} \operatorname{diag}(\boldsymbol{r})\boldsymbol{U}^\top\boldsymbol{U} \operatorname{diag}(\boldsymbol{r})\right) +$$

$$+ \frac{N^2}{(1+\Gamma)\,E} \left\|\boldsymbol{A}\right\|_{\mathrm{F}}^2$$

$$- \frac{2}{(1+\Gamma)\,E} \sum_{i=1}^{N} \sum_{j \in \mathcal{N}(i)} \boldsymbol{U}_{i,:} \operatorname{diag}(\boldsymbol{r}) \left(\boldsymbol{V}^\top\right)_{:,j} a_{i,j} +$$

$$\frac{1-e}{(1+\Gamma)\,E} \sum_{i=1}^{N} \sum_{j \in \mathcal{N}(i)} \left(\boldsymbol{U}_{i,:} \operatorname{diag}(\boldsymbol{r}) \left(\boldsymbol{V}^\top\right)_{:,j}\right)^2$$

Here, the last equality uses the trace cyclicity, i.e., $\forall \boldsymbol{I}, \boldsymbol{J} \in \mathbb{R}^{N \times K} : \operatorname{Tr}(\boldsymbol{I}\boldsymbol{J}^\top) = \operatorname{Tr}(\boldsymbol{J}^\top\boldsymbol{I})$, with $\boldsymbol{I} = \boldsymbol{V} \operatorname{diag}(\boldsymbol{r})\boldsymbol{U}^\top\boldsymbol{U} \operatorname{diag}(\boldsymbol{r})$ and $\boldsymbol{J}^\top = \boldsymbol{V}^\top$.

To calculate the first term efficiently, we can either perform matrix multiplication from right to left or compute $\boldsymbol{U}^\top\boldsymbol{U}$ and $\boldsymbol{V}^\top\boldsymbol{V}$, followed by the rest of the product. This calculation has a time complexity of $\mathcal{O}(K^2 N)$ and a memory complexity $\mathcal{O}(KN)$. The second term in the equality is constant and, therefore, can be left out during optimization. The third and fourth terms in the expression are calculated using message-passing, and thus have a time complexity of $\mathcal{O}(KE)$. Overall, we end up with a complexity of $\mathcal{O}(K^2 N + KE)$ and a space complexity of $\mathcal{O}(KN + E)$ for the full computation of the loss and its gradients with respect to $\boldsymbol{U}, \boldsymbol{V}$ and $\boldsymbol{r}$. $\qquad\square$

## F  EXTENDING THE DENSIFYING WEAK REGULARITY LEMMA FOR GRAPHON-EDGE-SIGNALS

In this section we prove a version of Theorem D.3 for the case where the graph has an edge signal. The proof is very similar to the previous case, and the new theorem can be used for the analysis of IBG-NN when used for knowledge graphs (see Section I).

### F.1  WEIGHTED FROBENIUS AND CUT NORMS FOR GRAPHON-EDGE-SIGNALS

**Graphon-edge-signal**  A graphon-edge-signal is a pair $(V, Y)$ where V is a graphon and $Y : [0,1]^2 \to \mathbb{R}^D$ is a measurable function.

**Weighted edge-signal Frobenius norm** Consider the real Hilbert space $L^2([0,1]^2; q) \times (L^2[0,1]^2)^D$ defined with the weighted inner product

$$\langle (V,Y), (V',Y') \rangle_q = \langle (V,Y), (V',Y') \rangle_{q,\alpha,\beta} =$$

$$= \alpha \frac{1}{\|1\|_{1;q}} \iint_{[0,1]^2} V(x,y)V'(x,y)q(x,y)dxdy + \frac{\beta}{D} \sum_{j=1}^{D} \iint_{[0,1]^2} Y_j(x,y)Y'_j(x,y)dxdy.$$

We call the corresponding weighted norm the *weighted edge-signal Frobenius norm*, denoted by

$$\|(V,Y)\|_{\mathrm{F};q} = \|(V,Y)\|_{\mathrm{F};q,\alpha,\beta} = \sqrt{\alpha \|V\|_{\mathrm{F};q}^2 + \frac{\beta}{D} \sum_{j=1}^{D} \|Y_j\|_{\mathrm{F}}^2},$$

We similarly extend the definition of a graphon weighted cut norm and cut metric.

**Graphon weighted cut norm and cut metric.** A kernel-edge-signal $(V,Y)$ is a pair where $V : [0,1]^2 \to [-1,1]$ and $Y : [0,1]^2 \to \mathbb{R}^D$ are measurable. Define for a kernel-edge-signal $(V,Y)$ the *weighted edge signal cut norm*

$$\|(V,Y)\|_{\square;q,\alpha,\beta} = \|(V,Y)\|_{\square;q} = \frac{\alpha}{\|1\|_{1;q}} \sup_{\mathcal{U},\mathcal{V}} \left| \int_{\mathcal{U}} \int_{\mathcal{V}} V(x,y)q(x,y)dxdy \right| +$$

$$+ \beta \frac{1}{D} \sum_{j=1}^{D} \sup_{\mathcal{U},\mathcal{V}} \left| \int_{\mathcal{U}} \int_{\mathcal{V}} Y_j(x,y)dxdy \right|,$$

where the supremum is over the set of measurable subsets $\mathcal{U}, \mathcal{V} \subset [0,1]$.

The weighted edge signal cut metric between two graphon-edge-signals $(W,f)$ and $(W',f')$ is defined to be $\|(W,f) - (W',f')\|_{\square;q}$.

For simplicity's sake, and for this section only, we refer to the weighted edge-signal Frobenius and cut norms simply as the weighted Frobenius and cut norms.

### F.2 IBGs with edge signals

Here, we define IBGs for graphon-edge-signals. We use the same terminology of *soft rank-$K$ IBG model* introduced in Definition D.2, slightly changing the signal part of the graphon.

**Definition F.1.** *Let $D \in \mathbb{N}$. Given a soft affiliation model $\mathcal{Q}$, the subset $[\mathcal{Q}]$ of $L^2[0,1]^2 \times (L^2[0,1]^2)^D$ of all elements of the form $(au(x)v(y), bu(z)v(w))$, with $u,v \in \mathcal{Q}$, $a \in \mathbb{R}$ and $b \in \mathbb{R}^D$, is called the* soft rank-1 intersecting block graphon (IBG) model *corresponding to $\mathcal{Q}$. Given $K \in \mathbb{N}$, the subset $[\mathcal{Q}]_K$ of $L^2[0,1]^2 \times (L^2[0,1]^2)^D$ of all linear combinations of $K$ elements of $[\mathcal{Q}]$ is called the* soft rank-$K$ IBG model *corresponding to $\mathcal{Q}$. Namely, $(C,p) \in [\mathcal{Q}]_K$ if and only if it has the form*

$$C(x,y) = \sum_{k=1}^{K} a_k u_k(x) v_k(y) \quad and \quad p(x,y) = \sum_{k=1}^{K} b_k u_k(x) v_k(y)$$

*where $(u_k)_{k=1}^{K} \in \mathcal{Q}^K$ are called the* target community affiliation functions*, $(v_k)_{k=1}^{K} \in \mathcal{Q}^K$ are called the* source community affiliation functions*, $(a_k)_{k=1}^{K} \in \mathbb{R}^K$ are called the* community affiliation magnitudes*, $(b_k)_{k=1}^{K} \in \mathbb{R}^{K \times D}$ are called the* edge features*. Any element of $[\mathcal{Q}]_K$ is called an* intersecting block graphon-signal (*IBG*).

We emphasize that for the rest of this section, when referencing weighted Frobenius and cut norms, as well as the *soft rank-$K$ IBG model*, we refer to the new definitions as formulated in Section F.

**Corollary F.1.** *Let $(W,s)$ be a graphon-edge-signal, $K \in \mathbb{N}$, $\delta > 0$, and let $\mathcal{Q}$ be a soft indicators model. Let $q$ be a weight function and $\alpha, \beta > 0$. Let $R \geq 1$ such that $K/R \in \mathbb{N}$. Consider the graphon-signal Frobenius norm with weight $\|(Y,y)\|_{\mathrm{F};q} = \|(Y,y)\|_{\mathrm{F};q,\alpha,\beta}$, and cut norm with weight $\|(Y,y)\|_{\square;q} := \|(Y,y)\|_{\square;q,\alpha,\beta}$. For every $k \in \mathbb{N}$, let*

$$\eta_k = (1+\delta) \inf_{(C,p) \in [\mathcal{Q}]_k} \|(W,s) - (C,p)\|_{\mathrm{F};q}^2.$$

*Then,*

1. *For every $m \in \mathbb{N}$, any* IBG $(C^*, p^*) \in [\mathcal{Q}]_m$ *that gives a close-to-best weighted Frobenius approximation of $(W, s)$ in the sense that*

$$\|(W, s) - (C^*, p^*)\|_{\mathrm{F};q}^2 \leq \eta_m, \tag{29}$$

*also satisfies*

$$\|(W, s) - (C^*, p^*)\|_{\square;q} \leq (\sqrt{\alpha} + \sqrt{\beta})\sqrt{\eta_m - \frac{\eta_{m+1}}{1 + \delta}}.$$

2. *If $m$ is uniformly randomly sampled from $[K]$, then in probability $1 - \frac{1}{R}$ (with respect to the choice of $m$),*

$$\sqrt{\eta_m - \frac{\eta_{m+1}}{1 + \delta}} \leq \sqrt{\alpha\|W\|_{\mathrm{F};q}^2 + \beta\|s\|_{\mathrm{F}}^2}\sqrt{\delta + \frac{R(1 + \delta)}{K}}. \tag{30}$$

*Specifically, in probability $1 - \frac{1}{R}$, any $(C^*, p^*) \in [\mathcal{Q}]_m$ which satisfy (29), also satisfies*

$$\|(W, s) - (C^*, p^*)\|_{\square;q} \leq (\sqrt{\alpha} + \sqrt{\beta})\left(\sqrt{\alpha\|W\|_{\mathrm{F},q}^2 + \beta\|s\|_{\mathrm{F}}^2}\sqrt{\delta + \frac{R(1 + \delta)}{K}}\right).$$

The proof is very similar to the original proof, with a slight adjustment for the analysis of the signal part of the graphon-signal. For completeness of the analysis, we provide the full proof.

*Proof.* Let us use Lemma D.2, with $\mathcal{H} = L^2([0, 1]^2; q) \times (L^2[0, 1]^2)^D$ with the weighted inner product

$$\langle (V, Y), (V', Y') \rangle_q = \alpha \frac{1}{\|1\|_{1;q}} \iint_{[0,1]^2} V(x, y)V'(x, y)q(x, y)dxdy +$$

$$+ \beta \sum_{j=1}^{D} \frac{1}{\|1\|_{1;q}} \iint_{[0,1]^2} Y(x, y)Y'(x, y)q(x, y)dxdy,$$

and corresponding norm denoted by $\|(V, Y)\|_{\mathrm{F};q}\sqrt{\alpha\|V\|_{\mathrm{F};q}^2 + \beta\sum_{j=1}^{D}\|Y_j\|_{\mathrm{F};q}^2}$, and $\mathcal{K}_j = [\mathcal{Q}]$. Note that the Hilbert space norm is a weighted Frobenius norm. Let $m \in \mathbb{N}$. In the setting of the lemma, we take $g = (W, s)$, and $g^* \in [\mathcal{Q}]_m$. By the lemma, any approximate Frobenius minimizer $(C^*, p^*)$, namely, that satisfies $\|(W, s) - (C^*, p^*)\|_{\mathrm{F};q} \leq \eta_m$, also satisfies

$$\langle (T, y), (W, s) - (C^*, p^*) \rangle_q \leq \|(T, y)\|_{\mathrm{F};q}\sqrt{\eta_m - \frac{\eta_{m+1}}{1 + \delta}}$$

for every $(T, y) \in [\mathcal{Q}]$. Hence, for every choice of measurable subsets $\mathcal{S}, \mathcal{T} \subset [0, 1]$, we have

$$\frac{1}{\|1\|_{1;q}}\left|\int_{\mathcal{S}}\int_{\mathcal{T}}(W(x, y) - C^*(x, y))q(x, y)dxdy\right|$$

$$= \left|\frac{1}{\alpha}\langle (\mathbb{1}_{\mathcal{S}} \otimes \mathbb{1}_{\mathcal{T}}, 0), (W, s) - (C^*, p^*) \rangle_q\right|$$

$$\leq \frac{1}{\alpha}\|(\mathbb{1}_{\mathcal{S}} \otimes \mathbb{1}_{\mathcal{T}}, 0)\|_{\mathrm{F};q}\sqrt{\eta_m - \frac{\eta_{m+1}}{1 + \delta}}$$

$$\leq \frac{1}{\alpha}\sqrt{\alpha}\sqrt{\eta_m - \frac{\eta_{m+1}}{1 + \delta}}$$

Hence, taking the supremum over $\mathcal{S}, \mathcal{T} \subset [0, 1]$, we also have

$$\alpha\|W - C^*\|_{\square;q} \leq \sqrt{\alpha}\sqrt{\eta_m - \frac{\eta_{m+1}}{1 + \delta}}.$$

Now, for $n$ randomly uniformly from $[K]$, consider the event $\mathcal{M}$ (regarding the uniform choice of $n$) of probability $(1 - 1/R)$ in which

$$\sqrt{\eta_n - \frac{\eta_{n+1}}{1 + \delta}} \leq \sqrt{\alpha\|W\|_{\mathrm{F};q}^2 + \beta\|s\|_{\mathrm{F}}^2}\sqrt{\delta + \frac{R(1 + \delta)}{K}}.$$

Hence, in the event $\mathcal{M}$, we also have

$$\alpha \left\| W - C^* \right\|_{\square;q} \leq \sqrt{\alpha^2 \|W\|_{\mathrm{F};q}^2 + \alpha\beta \left\| s \right\|_{\mathrm{F}}} \sqrt{\delta + \frac{R(1+\delta)}{K}}.$$

Similarly, for every measurable $\mathcal{S}, \mathcal{T} \subset [0,1]$ and every standard basis element $\boldsymbol{b} = (\delta_{j,i})_{i=1}^{D}$ for any $j \in [D]$,

$$\frac{1}{\|1\|_{1;q}} \left| \int_{\mathcal{S}} \int_{\mathcal{T}} (s(x,y) - p^*(x,y)) q(x,y) dx dy \right|$$

$$= \left| \frac{1}{\beta} \left\langle 0, \boldsymbol{b}(\mathbb{1}_{\mathcal{S}} \otimes \mathbb{1}_{\mathcal{T}}), (W,s) - (C^*, p^*) \right\rangle_q \right|$$

$$\leq \frac{1}{\beta} \|(0, \boldsymbol{b}(\mathbb{1}_{\mathcal{S}} \otimes \mathbb{1}_{\mathcal{T}})\|_{\mathrm{F};q} \sqrt{\eta_m - \frac{\eta_{m+1}}{1+\delta}}$$

$$\leq \frac{1}{\beta} \sqrt{\beta} \sqrt{\eta_m - \frac{\eta_{m+1}}{1+\delta}}$$

so, taking the supremum over $\mathcal{S}, \mathcal{T} \subset [0,1]$, independently for every $j \in [D]$, and averaging over $j \in [D]$, we get

$$\beta \left\| s - p^* \right\|_{\square;q} \leq \sqrt{\beta} \sqrt{\eta_m - \frac{\eta_{m+1}}{1+\delta}}.$$

Now, for the same event $\mathcal{M}$ as above regarding the choice of $n \in [K]$,

$$\beta \left\| s - p^* \right\|_{\square} \leq \sqrt{\alpha\beta \|W\|_{\mathrm{F};q}^2 + \beta^2 \left\| s \right\|_{\mathrm{F}}^2} \sqrt{\delta + \frac{R(1+\delta)}{K}}.$$

Overall, we get for every $m$ and corresponding approximately optimum $(C^*, p^*)$,

$$\|(W, s) - (C^*, p^*)\|_{\square;q} \leq (\sqrt{\alpha} + \sqrt{\beta}) \sqrt{\eta_m - \frac{\eta_{m+1}}{1+\delta}}.$$

Moreover, for uniformly sampled $n \in [K]$, in probability more than $1 - 1/R$,

$$\|(W, s) - (C^*, p^*)\|_{\square;q} \leq (\sqrt{\alpha} + \sqrt{\beta}) \left( \sqrt{\alpha \|W\|_{\mathrm{F},q}^2 + \beta \left\| s \right\|_{\mathrm{F}}^2} \sqrt{\delta + \frac{R(1+\delta)}{K}}. \right.$$

$\square$

## G    INITIALIZING THE OPTIMIZATION WITH SINGULAR VECTORS

Here, we propose a good initialization for the GD minimization of 8. We explain how to use the SVD of the graph to initialize the parameters of a rank $K$-IBG, before the gradient descent minimization of Equation (8). This is inspired by the eigendecomposition initialization described of ICGs. The full method is presented in Section G, and summarized here.

We begin by calculating the $K/4$ SVD decomposition of the graph adjacency matrix. Denote by $\boldsymbol{\sigma}_{K/4} = (\sigma_k)_{k=1}^{K/4}$ the sequence of the $K/4$ largest singular values of $\boldsymbol{A}$, and by $\boldsymbol{\Phi}_{K/4} = (\boldsymbol{\phi}_k)_{k=1}^{K/4}$, $\boldsymbol{\Psi}_{K/4} = (\boldsymbol{\psi}_k)_{k=1}^{K/4}$ their corresponding left and right singular vectors.

For each singular value $\sigma$ and corresponding singular vectors $\boldsymbol{\phi}, \boldsymbol{\psi}$, we designate $\{\frac{\boldsymbol{\phi}_+}{\|\boldsymbol{\phi}_+\|}, \frac{\boldsymbol{\phi}_+}{\|\boldsymbol{\phi}_+\|}, \frac{\boldsymbol{\phi}_-}{\|\boldsymbol{\phi}_-\|}, \frac{\boldsymbol{\phi}_-}{\|\boldsymbol{\phi}_-\|}\}$ as target communities, and $\{\frac{\boldsymbol{\psi}_+}{\|\boldsymbol{\psi}_+\|}, \frac{\boldsymbol{\psi}_-}{\|\boldsymbol{\psi}_-\|}, \frac{\boldsymbol{\psi}_+}{\|\boldsymbol{\psi}_+\|}, \frac{\boldsymbol{\psi}_-}{\|\boldsymbol{\psi}_-\|}\}$ as the corresponding source communities, where $\boldsymbol{\xi}_{\pm} \in [0, \infty)^N$ denotes the positive or negative parts of the vector $\boldsymbol{\xi}$, i.e., $\boldsymbol{\xi} = \boldsymbol{\xi}_+ - \boldsymbol{\xi}_-$. The corresponding affiliation magnitudes are then taken to be

$$r_1 = \sigma \left\| \boldsymbol{\phi}_+ \right\|_{\infty} \left\| \boldsymbol{\psi}_+ \right\|_{\infty}, \ r_2 = -\sigma \left\| \boldsymbol{\phi}_+ \right\|_{\infty} \left\| \boldsymbol{\psi}_- \right\|_{\infty},$$
$$r_3 = -\sigma \left\| \boldsymbol{\phi}_- \right\|_{\infty} \left\| \boldsymbol{\psi}_+ \right\|_{\infty}, \ r_4 = \sigma \left\| \boldsymbol{\phi}_- \right\|_{\infty} \left\| \boldsymbol{\psi}_- \right\|_{\infty}.$$

If $K$ is not divisible by 4, we discard the excess components with the smallest community affiliation magnitudes in absolute value.

To efficiently calculate the leading left and right singular vectors, we may use power method variants such as the Lanczos algorithm (Saad, 2011) or simultaneous iteration (Trefethen & Bau, 2022) in $\mathcal{O}(E)$ operations per iteration. For very large graphs, we propose in Section G.2 a more efficient randomized SVD algorithm that does not require reading the whole graph into memory at once.

## G.1 THEORETICAL ANALYSIS OF SVD INITIALIZATION

Next, we analyze this initialization and show that it attains a relatively high initial accuracy. The following is a corollary of the densifying weak regularity lemma with constant $q(x, y) = \frac{N^2}{E}$, the signal weight in the cut norm set to $\beta = 0$, using all measurable functions from $[0, 1]^2$ to $[0, 1]$ as the soft affiliation model, and taking relative Frobenius error with $\delta = 0$ on theorem D.3. In this case, according to the best rank-$K$ approximation theorem (Eckart–Young–Mirsky Theorem (Trefethen & Bau, 2022, Theorem 5.9)), the minimizer of the Frobenius error is the rank-K singular value decomposition (SVD). This leads to the following corollary.

**Corollary G.1.** *Let $A$ be a graph, $K \in \mathbb{N}$, let $m$ be sampled uniformly from $[K]$, and let $R \geq 1$ such that $K/R \in \mathbb{N}$. Let $u_1, \ldots, u_m$ and $v_1, \ldots, v_m$ be the leading left and right singular vectors of $A$ respectively, with singular values $\sigma_1, \ldots, \sigma_m$ of highest magnitudes $|\sigma_1| \geq |\sigma_2| \geq \ldots \geq |\sigma_m|$, and let $C^* = \sum_{k=1}^{m} \sigma_k u_k v_k^\top$. Then, in probability $1 - \frac{1}{R}$ (with respect to the choice of $m$),*

$$\|A - C^*\|_\square < \|A\|_\mathrm{F} \sqrt{\frac{R}{K}}.$$

*Proof.* Consider Theorem D.3, with $\delta = 0, \beta = 0, \Gamma = 0$, and taking the constant weight

$$q(x, y) = \frac{N^2}{E}.$$

Under this setting, the weighted Frobenius norm becomes the standard Frobenius norm, and the weighted similarity measure becomes the cut norm. Consider the induced graphon signals $W_A$ and $W_{C^*}$. Note that under the standard Frobenius norm, $W_{C^*}$ satisfies Equation (21), as $W_{C^*}$ is the SVD of $W_A$, and is also the best Frobenius norm approximator. Hence, the bound in Equation (23) is satisfied in probability $1 - 1/R$, and becomes

$$\|W_A - W_{C^*}\|_\square \leq \|W_A\|_\mathrm{F} \sqrt{\frac{R}{K}}.$$

We immediately get in probability $1 - 1/R$

$$\|A - C^*\|_\square < \|A\|_\mathrm{F} \sqrt{\frac{R}{K}},$$

as required. $\qquad\square$

The initialization is based on Corollary G.1 restricted to induced graphon-signals with a densifying cut similarity with a constant weight $q_{i,j} = N^2/E$. Consider $C = U_{K/4}\Sigma_{K/4}V_{K/4}^\top$, the rank $K/4$ singular value decomposition of $A$. We get

$$\|A - C\|_{\square;N,E} < \frac{N}{\sqrt{E}}\sqrt{\frac{4R}{K}}. \tag{31}$$

We note that while the lemma guarantees a good initialization for the graph, the goodness is not measured in terms of the graph-signal densifying similarity, but rather with respect to cut metric. Still, we find that in practice this initialization improves performance significantly.

It is important to note that the method described in the previous section relies on computing the singular value decomposition (SVD) of the graph's adjacency matrix. Traditional algorithms for SVD require the entire adjacency matrix to be loaded into memory, which can be computationally expensive for large graphs. Similar to the approach proposed for computing the IBG in Section H, we aim to allow SVD computation without loading the entire edge set into memory.

In the following subsection, we introduce a Monte Carlo algorithm for computing the SVD of a matrix by processing only a random small fraction of its rows, significantly reducing memory requirements.

## G.2 MONTE CARLO SVD ALGORITHM

For very large graphs, it is impossible to load the whole graph to GPU memory and estimate the SVD with standard algorithms. Instead, we propose here a randomized SVD algorithm which loads the matrix to GPU memory in small enough chunks.

The following theorem relates eigendecomposition of symmetric matrices to SVD (Trefethen & Bau, 2022, Lecture 31).

**Theorem G.2.** *Let* $A \in \mathbb{R}^{N \times N}$ *with singular values* $\sigma_1 \geq \ldots \geq \sigma_N$, *left singular vectors* $u_1, \ldots, u_n$, *and right singular vectors* $v_1, \ldots, v_N$. *Consider the* augmented matrix

$$B = \left( \begin{array}{cc} \mathbf{0} & A^\top \\ A & \mathbf{0} \end{array} \right).$$

*Then, the eigenvalues of* $B$ *are exactly* $\sigma_1, \ldots, \sigma_N, -\sigma_N, \ldots, -\sigma_1$, *and the eigenvectors are all vectors of the form*

$$\frac{1}{\sqrt{2}} \left( \begin{array}{c} v_i \\ \pm u_i \end{array} \right), \quad i = 1, \ldots, N. \tag{32}$$

Theorem G.2 is the standard way to convert eigendecomposition algorithms of symmetric matrices to SVD of general matrices. Namely, one applies the eigendecomposition algorithm to the symmetric augmented matrix $B$, and reads the singular values and vectors from $\sigma_1, \ldots, \sigma_N, -\sigma_N, \ldots, -\sigma_1$ and from (32).

We hence start with a basic randomized eigendecomposition algorithm for symmetric matrices, based on the simultaneous iteration (also called the block power method) (Trefethen & Bau, 2022, Part V).

Given a symmetric matrix $C$, the standard *simultaneous iteration* (Algorithm 1) is an algorithm for finding the leading (largest in their absolute values) $M$ eigenvector-eigenvalue pairs of $C$. We next replace the full matrix product $CQ$ by a Monte Carlo method, initially proposed in (Drineas et al., 2006). Consider a matrix $C \in \mathbb{R}^{N \times N}$ with columns $c_1, \ldots, c_N \in \mathbb{R}^N$ and a column vector $v = (v_1, \ldots, v_N) \in \mathbb{R}^N$. Let $m_1, \ldots, m_J$ be chosen independently uniformly at random from $[N]$. Let us denote $m = (m_1, \ldots, m_J)$ and

$$[Cv]_m = \sum_{j=1}^{J} v_{m_j} c_{m_j}.$$

One can show that in high probability

$$[Cv]_m \approx Cv,$$

where the expected square error satisfies

$$\mathbb{E}\|[Cv]_m - Cv\|_2^2 = O(1/J).$$

The advantage in the computation $[Cv]_m$ is that it reduces the computational complexity of matrix-vector product from $O(N^2)$ to $O(JN)$.

We hence consider a *Monte Carlo simultaneous iteration algorithm*, which is identical to Algorithm 1 with the exception that the matrix-vector product $Z = CQ$ is approximated by $[CQ]_m$.

---

**Algorithm 1** Simultaneous Iteration for Eigendecomposition of Symmetric Matrix

---

**Input:** Matrix $C \in \mathbb{R}^{N \times N}$, number of leading eigenvalues to be computed $M$, number of iterations $J$.
Initialize $Q \in \mathbb{R}^{N \times M}$ randomly
**for** $i = 1$ **to** $J$ **do**
    $Z = CQ$
    $(Q, R) = $ Reduced-QR-Factorization$(Z)$
**end for**
Compute $\lambda = (\lambda_1, \ldots, \lambda_M)$ with $\lambda_j = Q[:, j]^\top CQ[:, j]$
**Output:** Approximate eigenvalues $\lambda_1, \ldots, \lambda_M$ and corresponding columns $v_1, \ldots, v_M$ of $Q$ as the approximate eigenvectors.

---

Lastly, we would like to use the Monte Carlo simultaneous iteration algorithm for estimating the SVD of a matrix $\boldsymbol{A}$ via Theorem G.2. For that, we require an additional consideration. Note that the simultaneous iteration finds the leading eigenvalues and eigenvectors (up to sign) in case they are distinct in their absolute value. In case $\lambda_i \neq \lambda_j$ but $|\lambda_i| = |\lambda_j|$, the simultaneous iteration would find a vector in the span of the eigenspaces corresponding to $\lambda_i$ and $\lambda_j$. Now, note that Theorem G.2 builds the SVD of $\boldsymbol{A}$ via the eigendecomposition of $\boldsymbol{B}$, and every value $\lambda_i$ of $\boldsymbol{B}$ is repeated twice, with positive and negative sign (in case the singular values are not repeated), and Algorithm 2 finds vectors only in the span of the two eigenspaces corresponding to $\pm\lambda_i$. This is not an issue in the exact algorithm, as the singular vectors can be read off (32) regardless of the mixing between eigenspaces. However, in the Monte Carlo simultaneous iteration algorithm, the algorithm separates the two dimensional spaces of $\boldsymbol{B}$ to one-dimensional spaces arbitrarily due to the inexact Monte Carlo matrix product. Moreover, the split of the space to two one dimensional spaces arbitrarily changes between iterations. Hence, when naively implemented, the SVD algorithm based on Theorem G.2 fails to converge.

Instead, we apply the Monte Carlo eigendecomposition algorithm on $\boldsymbol{B} + \lambda_1\boldsymbol{I}$, where $\lambda_1$ is the largest eigenvalue of $\boldsymbol{B}$, computed by a Monte Carlo power iteration on $\boldsymbol{B}$. Since the addition of $\lambda_1\boldsymbol{I}$ shifts the spectrum of $\boldsymbol{B}$ by $\lambda_1$, all eigenvalues of $\boldsymbol{B}$ become non-negative and distinct (under the assumption that the singular values of $\boldsymbol{A}$ are distinct). We summarize in Algorithm 2 the resulting method (in the next page).

## H  LEARNING IBG WITH SUBGRAPH SGD

For message-passing neural networks, processing large graphs becomes challenging when the number of edges $E$ exceeds the capacity of GPU memory. For this reason, processing IBGs with neural networks, which take $\mathcal{O}(N)$ operations, is advantageous. However, one still has to fit the IBG to the graph as a preprocessing step, which takes $\mathcal{O}(E)$ operations and memory complexity. To address this, we propose two sampling-based optimization methods for the IBG. The first method performs node sampling, extending the SGD approach of Finkelshtein et al. (2024a). The second samples individual entries of the adjacency matrix, which we refer to as diodes. We term these methods node-sampling SGD and diode-sampling SGD, accordingly.

### H.1  CONSTRUCTION OF NODE SAMPLING SGD

In a standard GD procedure, all edges of the graph are loaded into memory. Instead, in our node-sampling SGD procedure, a set of $M \ll N$ nodes is sampled uniformly with replacement from $[N]$. Denote these nodes by $\boldsymbol{n} := (n_m)_{m=1}^M$. We then perform the gradient step using the gradients calculated solely using these sampled nodes. More specifically, by denoting $\boldsymbol{A}^{(\boldsymbol{n})} \in \mathbb{R}^{M \times M}$ the sampled subgraph with entries $a_{i,j}^{(\boldsymbol{n})} = a_{n_i,n_j}$, $\boldsymbol{X}^{(\boldsymbol{n})} \in \mathbb{R}^{M \times K}$ the sampled sub-signal with entries $\boldsymbol{x}_i^{(\boldsymbol{n})} = \boldsymbol{x}_{n_i}$, and $\boldsymbol{U}^{(\boldsymbol{n})}, \boldsymbol{V}^{(\boldsymbol{n})} \in [0,1]^{M \times K}$ as the sampled community target and source affiliation matrices with entries $u_{i,j}^{(\boldsymbol{n})} = u_{n_i,j}$ and $v_{i,j}^{(\boldsymbol{n})} = v_{n_i,j}$. The loss over the sampled nodes becomes:

$$L^{(M)}(\boldsymbol{U}^{(\boldsymbol{n})}, \boldsymbol{V}^{(\boldsymbol{n})}, \boldsymbol{r}, \boldsymbol{F}, \boldsymbol{B}) = \frac{\alpha(1+\Gamma)\mu}{M^2} \sum_{i,j=1}^M \sum_{k=1}^K (u_{n_i,k} r_k v_{n_j,k} - a_{n_i,n_j})^2 q_{n_i,n_j}$$

$$+ \frac{\beta}{MD} \sum_{i=1}^M \sum_{d=1}^D \sum_{k=1}^K (u_{n_i,k} f_{k,d} + v_{n_i,k} b_{k,d} - x_{n_i,d})^2,$$

### H.2  CONSTRUCTION OF DIODE SAMPLING SGD

We sample a set of $M \ll N^2$ diodes $\mathcal{D} = (\boldsymbol{s}, \boldsymbol{t}) = (s_m, t_m)_{m=1}^M$ independently from a distribution we define over all diodes of the graph. The probability of sampling diode $(i,j)$ is $\frac{q_{ij}}{(1+\Gamma)E}$, and the

---

**Algorithm 2** Monte Carlo Simultaneous Iteration for SVD decomposition of Non-Symmetric Matrix

---

**Input:** Matrix $C \in \mathbb{R}^{N \times N}$, number of leading eigenvalues to be computed $M$, number of iterations $J$, sample ratio $0 < r \leq 1$.

---

{*Augment the non-symmetric matrix to be symmetric*}

Define $B = \begin{pmatrix} \mathbf{0} & C^\top \\ C & \mathbf{0} \end{pmatrix}$.

---

{*Leading eigenvalue estimation*}

Initialize $Q_1 \in \mathbb{R}^{2N \times 1}$ randomly
**for** $i = 1$ **to** $J$ **do**
    Generate $2N \cdot r$ samples $n$ of indices from $[2N]$ with repetitions.
    $\tilde{B} = B[:, n]$
    $\tilde{Q}_1 = Q_1[n, 1]$
    $Z = \tilde{B}\tilde{Q}_1$
    $(Q_1, R) = $ Reduced-QR-Factorization$(Z)$
**end for**
$\lambda_1 = Q_1^\top B Q_1$

---

{*Shift the spectrum of $B$*}

$\hat{B} = B + |\lambda_1| I_{2N}$

---

{*Randomized simultaneous iteration on $\hat{B}$*}

Initialize $Q \in \mathbb{R}^{2N \times M}$ randomly
**for** $i = 1$ **to** $J$ **do**
    Sample vector $n$ of $2N \cdot r$ indices from $[2N]$ with repetitions.
    $\tilde{B} = \hat{B}[:, n]$
    $\tilde{Q} = Q[n, :]$
    $Z = \tilde{B}\tilde{Q}$
    $(Q, R) = $ Reduced-QR-Factorization$(Z)$
**end for**

---

{*Generate the output*}

Sample a vector $n$ of $2N \cdot r$ indices from $[2N]$ with repetitions.
$\tilde{B} = B[:, n]$
$\tilde{Q} = Q[n, :]$
Compute $\sigma = (\sigma_1, \ldots, \sigma_M)$ with $\sigma_j = Q[:, j]^\top \tilde{B}\tilde{Q}[:, j]$
$Q_v = Q[1 : N, :]$
$Q_u = Q[N + 1 : 2N, :]$

---

**Output:** Approximate singular values $\sigma_1, \ldots, \sigma_m$ and corresponding columns $u_1, \ldots, u_m$ and $v_1, \ldots, v_m$ of $Q_u$ and $Q_v$ as the approximate left and right singular vectors.

---

loss over the sampled nodes is

$$L^{(M)}(U, V, r) = \frac{\alpha(1 + \Gamma)}{M} \sum_{m=1}^{M} \sum_{k=1}^{K} (u_{s_m,k} r_k v_{t_m,k} - a_{s_m,t_m})^2.$$

We then perform the gradient step using the gradients calculated solely using these sampled diodes.

We note that for diode-sampling SGD, we define the loss only over the graph part. To perform SGD over the signal part of the loss, the method is identical to node-sampling SGD.

### H.3 THEORETICAL ANALYSIS OF DIODE-SAMPLING SGD

In the following section, we prove that the gradients calculated using diode-sampling SGD with respect to the sampled diodes approximate those calculated over the full graph using the standard loss in (8). Throughout the section we denote $\mu = 1/\sum q_{i,j}$ and refer only to the sample loss of diode-sampling SGD.

**Proposition H.1.** *Let* $0 < p < 1$. *Consider the Frobenius loss weighted by* $\boldsymbol{Q}$, $\alpha$ *and* $\beta$. *If we restrict all entries of* $\boldsymbol{C}$, $\boldsymbol{U}$, $\boldsymbol{V}$, *and* $\boldsymbol{r}$ *to be in* $[-1, 1]$, *then in probability at least* $1 - p$, *for every* $k \in [K]$ *and* $m \in [M]$

$$\left| \nabla_{u_{n_m,k}} L - \nabla_{u_{n_m,k}} L^{(M)} \right| \leq 2\alpha(1+\Gamma)\mu\sqrt{\frac{2\log(1/p) + 2\log(K) + 2\log(6)}{M}},$$

$$\left| \nabla_{v_{n_t,k}} L - \nabla_{v_{n_m,k}} L^{(M)} \right| \leq 2\alpha(1+\Gamma)\mu\sqrt{\frac{2\log(1/p) + 2\log(K) + 2\log(6)}{M}},$$

$$\left| \nabla_{r_k} L - \nabla_{r_k} L^{(M)} \right| \leq 2\alpha(1+\Gamma)\mu\sqrt{\frac{2\log(1/p) + 2\log(K) + 2\log(6)}{M}}.$$

Theorem H.1 provide a probabilistic bound on the difference between the gradients computed on the full graph and those calculated using subgraph SGD. Notably, it shows that the gradients of the IBG parameters calculated with SGD closely approximate those calculated with standard GD.

We prove Theorem H.1 by comparing the gradients of the full loss $L$ with those of the sampled loss $L^{(M)}$.

Consider the graph part of the IBG loss:

$$L(\boldsymbol{U}, \boldsymbol{V}, \boldsymbol{r}) = \alpha(1+\Gamma)\mu \sum_{i,j=1}^{N} \sum_{k=1}^{K} (u_{i,k} r_k v_{j,k} - a_{i,j})^2 q_{i,j}$$

Next, we calculate the gradients of $\boldsymbol{C} = \boldsymbol{U} \operatorname{diag}(\boldsymbol{r}) \boldsymbol{V}^\top$ with respect to $\boldsymbol{U}$, $\boldsymbol{V}$, and $\boldsymbol{r}$ in coordinates. We have

$$\nabla_{r_k} c_{i,j} = u_{i,k} v_{j,k},$$
$$\nabla_{u_{t,k}} c_{i,j} = r_k \delta_{i-t} v_{j,k},$$

and

$$\nabla_{v_{t,k}} c_{i,j} = r_k \delta_{j-t} u_{i,k},$$

where $\delta_i$ is 1 if $i = 0$ and zero otherwise. Hence, the gradients of the loss are

$$\nabla_{r_k} L = 2\alpha(1+\Gamma)\mu \sum_{i,j=1}^{N} (c_{i,j} - a_{i,j}) q_{i,j} u_{i,k} v_{j,k},$$

$$\nabla_{u_{n,k}} L = 2\alpha(1+\Gamma)\mu \sum_{j=1}^{N} (c_{n,j} - a_{n,j}) q_{n,j} r_k v_{j,k},$$

and

$$\nabla_{v_{n,k}} L = 2\alpha(1+\Gamma)\mu \sum_{j=1}^{N} (c_{j,n} - a_{j,n}) q_{j,n} r_k u_{j,k}.$$

Similarly

$$\nabla_{r_k} L^{(M)} = \frac{2\alpha(1+\Gamma)}{M} \sum_{j=1}^{M} (c_{s_j,t_j} - a_{s_j,t_j}) u_{s_j,k} v_{t_j,k},$$

$$\nabla_{u_{n_m,k}} L^{(M)} = \frac{2\alpha(1+\Gamma)\mu}{M} \sum_{j=1}^{M} (c_{n_m,t_j} - a_{n_m,t_j}) r_k v_{t_j,k},$$

and

$$\nabla_{v_{n_m,k}} L^{(M)} = \frac{2\alpha(1+\Gamma)\mu}{M} \sum_{j=1}^{M} (c_{s_j,n_m} - a_{s_j,n_m}) r_k u_{s_j,k},$$

The following convergence analysis is based on Hoeffding's inequality, and a supporting Monte Carlo approximation lemma.

**Theorem H.2** (Hoeffding's Inequality)**.** *Let $Y_1, \dots, Y_M$ be independent random variables such that $a \le Y_m \le b$ almost surely. Then, for every $k > 0$,*

$$\mathbb{P}\Big(\Big|\frac{1}{M}\sum_{m=1}^{M}(Y_m - \mathbb{E}[Y_m])\Big| \ge k\Big) \le 2\exp\Big(-\frac{2k^2 M}{(b-a)^2}\Big).$$

We now use Hoeffding's inequality to derive a standard Monte Carlo approximation error bound.

**Lemma H.3.** *Let $\{i_m\}_{m=1}^{M}$ be uniform i.i.d in $[N]$. Let $\boldsymbol{v} \in \mathbb{R}^N$ be a vector with entries $v_n$ in the set $[-1, 1]$. Then, for every $0 < p < 1$, in probability at least $1 - p$*

$$\left|\frac{1}{M}\sum_{m=1}^{M} v_{i_m} - \frac{1}{N}\sum_{n=1}^{N} v_n\right| \le \sqrt{\frac{2\log(1/p) + 2\log(2)}{M}}.$$

*Proof.* This is a direct result of Hoeffding's Inequality on the i.i.d. variables $\{v_{i_m}\}_{m=1}^{M}$. $\qquad\square$

We use Theorem H.3 on the i.i.d samples $\boldsymbol{m}$. We first show:

$$\mathbb{E}[\nabla_{r_k} L^{(M)}] = \nabla_{r_k} L,$$

$$\mathbb{E}[\nabla_{u_{n_m,k}} L^{(M)}] = \nabla_{u_{n_m,k}} L,$$

and

$$\mathbb{E}[\nabla_{v_{n_m,k}} L^{(M)}] = \nabla_{v_{n_m,k}} L.$$

Indeed we have

$$\mathbb{E}[\nabla_{r_k} L^{(M)}] = \frac{2\alpha(1+\Gamma)}{M} \sum_{j=1}^{M} \mathbb{E}[(c_{s_j,t_j} - a_{s_j,t_j}) u_{s_j,k} v_{t_j,k}] =$$

$$= \frac{2\alpha(1+\Gamma)}{M} \sum_{j=1}^{M} \mathbb{E}[(c_{s_1,t_1} - a_{s_1,t_1}) u_{s_1,k} v_{t_1,k}] =$$

$$= 2\alpha(1+\Gamma) \sum_{ij=1}^{N} P(s=i, t=j)(c_{i,j} - a_{i,j}) u_{i,k} v_{j,k} =$$

$$= 2\alpha(1+\Gamma)\mu \sum_{ij=1}^{N} (c_{i,j} - a_{i,j}) q_{ij} u_{i,k} v_{j,k},$$

$$\mathbb{E}[\nabla_{u_{n_m,k}} L^{(M)}] = \frac{2\alpha(1+\Gamma)}{M} \sum_{j=1}^{M} \mathbb{E}[(c_{n_m,t_j} - a_{n_m,t_j}) r_k v_{t_j,k}] =$$

$$= \frac{2\alpha(1+\Gamma)}{M} \sum_{j=1}^{M} \mathbb{E}[(c_{n_m,t_1} - a_{n_m,t_1}) r_k v_{t_1,k}] =$$

$$= 2\alpha(1+\Gamma)\mu \sum_{j=1}^{N} (c_{n_m,j} - a_{n_m,j}) q_{n_m j} r_k v_{j,k},$$

and

$$\mathbb{E}[\nabla_{v_{n_m,k}} L^{(M)}] = \frac{2\alpha(1+\Gamma)}{M} \sum_{j=1}^{M} \mathbb{E}[(c_{n_m,t_j} - a_{n_m,t_j})r_k v_{t_j,k}] =$$

$$= \frac{2\alpha(1+\Gamma)}{M} \sum_{j=1}^{M} \mathbb{E}[(c_{n_m,t_1} - a_{n_m,t_1})r_k v_{t_1,k}] =$$

$$= 2\alpha(1+\Gamma)\mu \sum_{j=1}^{N} (c_{n_m,j} - a_{n_m,j})q_{n_m j}r_k v_{j,k}.$$

This shows that the expected value of the sampled loss gradients is equal to the standard loss's gradients, which meets the conditions of Theorem H.3. We therefore use the lemma to obtain a probabilistic bound on the difference between the approximated gradients and the full gradients.

Specifically, for any $0 < p_1 < 1$, for every $k \in [K]$ there is an event $\mathcal{A}_k$ of probability at least $1 - p_1$ such that

$$\left| \nabla_{r_k} L - \nabla_{r_k} L^{(M)} \right| \le 2\alpha(a+\Gamma)\mu\sqrt{\frac{2\log(1/p_1) + 2\log(2)}{M}}.$$

For any $0 < p_2 < 1$, for every $k \in [K]$ there is an event $\mathcal{U}_k$ of probability at least $1 - p_2$ such that for every $n \in [N]$

$$\left| \nabla_{u_{n,k}} L - \nabla_{u_{n,k}} L^{(M)} \right| \le 2\alpha(a+\Gamma)\mu\sqrt{\frac{2\log(1/p_2) + 2\log(2)}{M}}.$$

For any $0 < p_3 < 1$, for every $k \in [K]$ there is an event $\mathcal{V}_k$ of probability at least $1 - p_3$ such that for every $n \in [N]$

$$\left| \nabla_{v_{n,k}} L - \nabla_{v_{n,k}} L^{(M)} \right| \le 2\alpha(a+\Gamma)\mu\sqrt{\frac{2\log(1/p_3) + 2\log(2)}{M}}.$$

Lastly, given $0 < p < 1$, choosing $p_1 = p_2 = p_3 = p/3K$ and intersecting all events for all coordinates gives in the event $\mathcal{E}$ of probability at least $1 - p$

$$\left| \nabla_{r_k} L - \nabla_{r_k} L^{(M)} \right| \le 2\alpha(a+\Gamma)\mu\sqrt{\frac{2\log(1/p) + 2\log(K) + 2\log(6)}{M}},$$

$$\left| \nabla_{u_{n,k}} L - \nabla_{u_{n,k}} L^{(M)} \right| \le 2\alpha(a+\Gamma)\mu\sqrt{\frac{2\log(1/p) + 2\log(K) + 2\log(6)}{M}},$$

and

$$\left| \nabla_{v_{n,k}} L - \nabla_{v_{n,k}} L^{(M)} \right| \le 2\alpha(a+\Gamma)\mu\sqrt{\frac{2\log(1/p) + 2\log(K) + 2\log(6)}{M}}.$$

proving Theorem H.1

### H.4 THEORETICAL ANALYSIS OF NODE SAMPLING SGD

In this section, we provide a similar result to the one in Section H.3, proving that the gradients calculated using node-sampling SGD with respect to the sampled nodes approximate those calculated over the full graph using the standard loss in (8). Throughout the section we use the loss of node-sampling SGD.

**Proposition H.4.** *Let $0 < p < 1$. Consider the Frobenius loss weighted by $\boldsymbol{Q}, \alpha$ and $\beta$. If we restrict all entries of $\boldsymbol{C}, \boldsymbol{P}, \boldsymbol{U}, \boldsymbol{V}, \boldsymbol{r}, \boldsymbol{F}$ and $\boldsymbol{B}$ to be in $[-1,1]$, then in probability at least $1 - p$, for every*

$k \in [K]$, $d \in [D]$ *and* $m \in [M]$

$$\left| \nabla_{u_{n_m,k}} L - \frac{M}{N} \nabla_{u_{n_m,k}} L^{(M)} \right| \le 2\alpha(1+\Gamma)\mu N \sqrt{\frac{2\log(1/p) + 2\log(2N) + 2\log(K) + 2\log(10)}{M}},$$

$$\left| \nabla_{v_{n_t,k}} L - \frac{M}{N} \nabla_{v_{n_m,k}} L^{(M)} \right| \le 2\alpha(1+\Gamma)\mu N \sqrt{\frac{2\log(1/p) + 2\log(2N) + 2\log(K) + 2\log(10)}{M}},$$

$$\left| \nabla_{r_k} L - \nabla_{r_k} L^{(M)} \right| \le 4\alpha(1+\Gamma)\mu N^2 \sqrt{\frac{2\log(1/p) + 2\log(N) + 2\log(K) + 2\log(10)}{M}},$$

$$\left| \nabla_{f_{k,d}} L - \nabla_{f_{k,d}} L^{(M)} \right| \le \frac{4\beta}{D} \sqrt{\frac{2\log(1/p) + 2\log(K) + 2\log(D) + 2\log(10)}{M}},$$

$$\left| \nabla_{b_{k,d}} L - \nabla_{b_{k,d}} L^{(M)} \right| \le \frac{4\beta}{D} \sqrt{\frac{2\log(1/p) + 2\log(K) + 2\log(D) + 2\log(10)}{M}}.$$

The proof of Theorem H.4 is similar to that of Theorem H.1. Define the graph part and the signal part of the IBG loss

$$L_1(\boldsymbol{U}, \boldsymbol{V}, \boldsymbol{r}) = \frac{\alpha(1+\Gamma)\mu}{N^2} \sum_{i,j=1}^{N} \sum_{k=1}^{K} (u_{i,k} r_k v_{j,k} - a_{i,j})^2 q_{i,j}$$

$$L_2(\boldsymbol{U}, \boldsymbol{V}, \boldsymbol{F}, \boldsymbol{B}) = \frac{\beta}{ND} \sum_{i=1}^{N} \sum_{d=1}^{D} \sum_{k=1}^{K} (u_{i,k} f_{k,d} + v_{i,k} b_{k,d} - x_{i,d})^2.$$

The IBG loss in (8) is $L = N^2 L_1 + L_2$. We normalize and multiply $L_1$ by $N^2$ for reasons that will become clear later.

Similarly, we define the graph and signal parts of the subgraph SGD loss

$$L_1^{(M)}(\boldsymbol{U}^{(n)}, \boldsymbol{V}^{(n)}, \boldsymbol{r}) = \frac{\alpha(1+\Gamma)\mu}{M^2} \sum_{i,j=1}^{M} \sum_{k=1}^{K} (u_{n_i,k} r_k v_{n_j,k} - a_{n_i,n_j})^2 q_{n_i,n_j}$$

$$L_2^{(M)}(\boldsymbol{U}^{(n)}, \boldsymbol{V}^{(n)}, \boldsymbol{F}, \boldsymbol{B}) = \frac{\beta}{MD} \sum_{i=1}^{M} \sum_{d=1}^{D} \sum_{k=1}^{K} (u_{n_i,k} f_{k,d} + v_{n_i,k} b_{k,d} - x_{n_i,d})^2,$$

where $L^{(M)} = N^2 L_1^{(M)} + L_2^{(M)}$

Next, we extend the calculations of the gradients of $\boldsymbol{C} = \boldsymbol{U} \operatorname{diag}(\boldsymbol{r}) \boldsymbol{V}^\top$ and $\boldsymbol{P} = \boldsymbol{UF} + \boldsymbol{VB}$ with respect to $\boldsymbol{U}$, $\boldsymbol{V}$, $\boldsymbol{r}$, $\boldsymbol{F}$ and $\boldsymbol{B}$ in coordinates. We have

$$\nabla_{f_{k,l}} p_{i,d} = u_{i,k} \delta_{l-d},$$
$$\nabla_{b_{k,l}} p_{i,d} = v_{i,k} \delta_{l-d},$$
$$\nabla_{u_{t,k}} p_{i,d} = f_{k,d} \delta_{i-t},$$

and

$$\nabla_{v_{t,k}} p_{i,d} = b_{k,d} \delta_{i-t}.$$

Hence,

$$\nabla_{r_k} L_1 = \frac{\alpha(1+\Gamma)\mu}{N^2} \sum_{i,j=1}^{N} (c_{i,j} - a_{i,j}) q_{i,j} u_{i,k} v_{j,k},$$

$$\nabla_{u_{t,k}} L_1 = \frac{\alpha(1+\Gamma)\mu}{N^2} \sum_{j=1}^{N} (c_{t,j} - a_{t,j}) q_{t,j} r_k v_{j,k} \qquad \nabla_{u_{t,k}} L_2 = \frac{\beta}{ND} \sum_{d=1}^{D} (p_{t,d} - x_{t,d}) f_{k,d},$$

$$\nabla_{v_{t,k}} L_1 = \frac{\alpha(1+\Gamma)\mu}{N^2} \sum_{j=1}^{N} (c_{t,j} - a_{t,j}) q_{t,j} r_k u_{j,k} \qquad \nabla_{v_{t,k}} L_2 = \frac{\beta}{ND} \sum_{d=1}^{D} (p_{t,d} - x_{t,d}) b_{k,d},$$

$$\nabla_{f_{k,d}} L_2 = \frac{\beta}{ND} \sum_{i=1}^{N} (p_{i,d} - x_{i,d}) u_{i,k},$$

and

$$\nabla_{b_{k,d}} L_2 = \frac{\beta}{ND} \sum_{i=1}^{N} (p_{i,d} - x_{i,d}) v_{i,k}.$$

Similarly

$$\nabla_{r_k} L_1^{(M)} = \frac{\alpha(1+\Gamma)\mu}{M^2} \sum_{i,j=1}^{M} (c_{n_i,n_j} - a_{n_i,n_j}) q_{n_i,n_j} u_{n_i,k} v_{n_j,k},$$

$$\nabla_{u_{n_m,k}} L_1^{(M)} = \frac{\alpha(1+\Gamma)\mu}{M^2} \sum_{j=1}^{M} (c_{n_m,n_j} - a_{n_m,n_j}) q_{n_m,n_j} r_k v_{n_j,k}$$

$$\nabla_{u_{n_m,k}} L_2^{(M)} = \frac{\beta}{MD} \sum_{d=1}^{D} (p_{n_m,d} - x_{n_m,d}) f_{k,d},$$

$$\nabla_{v_{n_m,k}} L_1^{(M)} = \frac{\alpha(1+\Gamma)\mu}{M^2} \sum_{j=1}^{M} (c_{n_m,n_j} - a_{n_m,n_j}) q_{n_m,n_j} r_k u_{n_j,k}$$

$$\nabla_{v_{n_m,k}} L_2^{(M)} = \frac{\beta}{MD} \sum_{d=1}^{D} (p_{n_m,d} - x_{n_m,d}) b_{k,d},$$

$$\nabla_{f_{k,d}} L_2^{(M)} = \frac{\beta}{MD} \sum_{i=1}^{M} (p_{n_i,d} - x_{n_i,d}) u_{n_i,k},$$

and

$$\nabla_{b_{k,d}} L_2^{(M)} = \frac{\beta}{MD} \sum_{i=1}^{M} (p_{n_i,d} - x_{n_i,d}) v_{n_i,k}.$$

The following Lemma provides an error bound between the sum of a 2D array of numbers and the sum of random points from the 2D array. We study the setting where one randomly (and independently) samples only points in a 1D axis, and the 2D random samples consist of all pairs of these samples. This results in dependent 2D samples, but still, one can prove a Monte Carlo-type error bound in this situation. The Lemma is similar to Lemma E.3. in Finkelshtein et al. (2024a), generalized to non-symmetric matrices.

**Lemma H.5.** *Let $\{i_m\}_{m=1}^{M}$ be uniform i.i.d in $[N]$. Let $A \in \mathbb{R}^{N \times N}$ with $a_{i,j} \in [-1, 1]$. Then, for every $0 < p < 1$, in probability more than $1 - p$*

$$\left| \frac{1}{N^2} \sum_{j=1}^{N} \sum_{n=1}^{N} a_{j,n} - \frac{1}{M^2} \sum_{m=1}^{M} \sum_{l=1}^{M} a_{i_m,i_l} \right| \le 2\sqrt{\frac{2\log(1/p) + 2\log(2N) + 2\log(2)}{M}}.$$

*Proof.* Let $0 < p < 1$. For each fixed $n \in [N]$, consider the independent random variables $Y_m^n = a_{i_m,n}$, with

$$\mathbb{E}(Y_m^n) = \frac{1}{N} \sum_{j=1}^{N} a_{j,n}$$

and $-1 \le Y_m \le 1$.

Similarly define the independent random variables $W_m^n = a_{n,i_m}$ with

$$\mathbb{E}(W_m^n) = \frac{1}{N} \sum_{j=1}^{N} a_{n,j}.$$

By Hoeffding's Inequality, for $k = \sqrt{\frac{2\log(1/p) + 2\log(2N) + 2\log(2)}{M}}$, we have

$$\left| \frac{1}{N} \sum_{j=1}^{N} a_{j,n} - \frac{1}{M} \sum_{m=1}^{M} a_{i_m,n} \right| \leq k$$

and

$$\left| \frac{1}{N} \sum_{j=1}^{N} a_{n,j} - \frac{1}{M} \sum_{m=1}^{M} a_{n,i_m} \right| \leq k$$

in the event $\mathcal{E}_n^Y$ and $\mathcal{E}_n^W$ of probability more than $1 - p/2N$. Intersecting the events $\{\mathcal{E}_n^Y\}_{n=1}^N$ and $\{\mathcal{E}_n^W\}_{n=1}^N$, we get $\forall n \in [N]$ :

$$\left| \frac{1}{N} \sum_{j=1}^{N} a_{j,n} - \frac{1}{M} \sum_{m=1}^{M} a_{i_m,n} \right| \leq k$$

and

$$\left| \frac{1}{N} \sum_{j=1}^{N} a_{n,j} - \frac{1}{M} \sum_{m=1}^{M} a_{n,i_m} \right| \leq k$$

in the event $\mathcal{E} = \cap_n \mathcal{E}_n^Y \cap_n \mathcal{E}_n^W$ with probability at least $1 - p$. The rows and columns of $A$ are not independent, meaning the probability of their intersection is at least $1 - p$ and by the triangle inequality, we also have in the event $\mathcal{E}$

$$\left| \frac{1}{NM} \sum_{l=1}^{M} \sum_{j=1}^{N} a_{i_l,j} - \frac{1}{M^2} \sum_{l=1}^{M} \sum_{m=1}^{M} a_{i_l,i_m} \right| \leq k,$$

and

$$\left| \frac{1}{N^2} \sum_{n=1}^{N} \sum_{j=1}^{N} a_{j,n} - \frac{1}{NM} \sum_{n=1}^{N} \sum_{m=1}^{M} a_{i_m,n} \right| \leq k.$$

Hence, by the triangle inequality,

$$\left| \frac{1}{N^2} \sum_{n=1}^{N} \sum_{j=1}^{N} a_{j,n} - \frac{1}{M^2} \sum_{l=1}^{M} \sum_{m=1}^{M} a_{i_m,i_l} \right| \leq 2k.$$

$\square$

We now derive bounds on the approximation errors for the gradients of $L$. Note that the gradients of the SGD loss with respect to each element of the IBG are

$$\nabla_{u_{n_m,k}} L^{(M)} = N^2 \nabla_{u_{n_m,k}} L^{(M)} + \nabla_{u_{n_m,k}} L_2^{(M)}$$

$$\nabla_{v_{n_m,k}} L^{(M)} = N^2 \nabla_{v_{n_m,k}} L_1^{(M)} + \nabla_{v_{n_m,k}} L_2^{(M)}$$

$$\nabla_{r_k} L^{(M)} = N^2 \nabla_{r_k} L_1^{(M)}$$

$$\nabla_{b_{k,m}} L^{(M)} = \nabla_{b_{k,m}} L_2^{(M)}$$

$$\nabla_{f_{k,m}} L^{(M)} = \nabla_{f_{k,m}} L_2^{(M)}$$

We use Lemmas H.3 and H.5. These results are applicable in our setting because all entries of the relevant matrices and vectors, $A, X, C, P, U, V, r, F$, and $B$ are bounded in $[-1, 1]$.

Specifically, for any $0 < p_1 < 1$, for every $k \in [K]$ there is an event $\mathcal{A}_k$ of probability at least $1 - p_1$ such that

$$\left| \nabla_{r_k} L - \nabla_{r_k} L^{(M)} \right| \leq 4\alpha(1 + \Gamma)\mu N^2 \sqrt{\frac{2\log(1/p_1) + 2\log(2N) + 2\log(2)}{M}}.$$

Moreover, for every $k \in [K]$ and $l \in [D]$, and every $0 < p_2 < 1$ there is an event $\mathcal{C}_{k,j}$ of probability at least $1 - p_2$ such that

$$\left| \nabla_{f_{k,m}} L - \nabla_{f_{k,m}} L^{(M)} \right| \leq \frac{4\beta}{D} \sqrt{\frac{2\log(1/p_2) + 2\log(2)}{M}}.$$

Similarly, for every $0 < p_3 < 1$ there is an event $\mathcal{D}_{k,m}$ of probability at least $1 - p_3$ such that

$$\left| \nabla_{b_{k,m}} L - \nabla_{b_{k,m}} L^{(M)} \right| \leq \frac{4\beta}{D} \sqrt{\frac{2\log(1/p_3) + 2\log(2)}{M}}$$

For the approximation analysis of $\nabla_{u_{n_m,l}} L$ and $\nabla_{v_{n_m,l}} L$, note that the index $n_i$ is random, so we derive a uniform convergence analysis for all possible values of $n_i$. For that, for every $n \in [N]$ and $k \in [K]$, define the vectors

$$\widetilde{\nabla_{u_{n,k}} L_1^{(M)}} = \frac{\alpha(1 + \Gamma)\mu}{M^2} \sum_{j=1}^{M} (c_{n,n_j} - a_{n,n_j})q_{n,n_j} r_k v_{n_j,k}$$

$$\widetilde{\nabla_{u_{n,k}} L_2^{(M)}} = \frac{\beta}{MD} \sum_{d=1}^{D} (p_{n,d} - x_{n,d})f_{k,d},$$

and

$$\widetilde{\nabla_{v_{n,k}} L_1^{(M)}} = \frac{\alpha(1 + \Gamma)\mu}{M^2} \sum_{j=1}^{M} (c_{n,n_j} - a_{n,n_j})q_{n,n_j} r_k u_{n_j,k}$$

$$\widetilde{\nabla_{v_{n,k}} L_2^{(M)}} = \frac{\beta}{MD} \sum_{d=1}^{D} (p_{n,d} - x_{n,d})b_{k,d}.$$

Note that $\widetilde{\nabla_{u_{n,k}} L^{(M)}}$ and $\widetilde{\nabla_{v_{n,k}} L^{(M)}}$ are not gradients of $L^{(M)}$ (since if $n$ is not a sample from $\{n_i\}$ the gradient must be zero), but are denoted with $\nabla$ for their structural similarity to $\nabla_{u_{n_m,k}} L^{(M)}$ and $\nabla_{v_{n_m,k}} L^{(M)}$. However, we get for every $m \in [M]$

$$\nabla_{u_{n_m,k}} L_2 = \widetilde{\nabla_{u_{n_m,k}} L_2^{(M)}}$$

and

$$\nabla_{v_{n_m,k}} L_2 = \widetilde{\nabla_{v_{n_m,k}} L_2^{(M)}}$$

Hence, for every $m \in [M]$, we have

$$\nabla_{u_{n,k}} L^{(M)} = N^2 \widetilde{\nabla_{u_{n,k}} L_1^{(M)}}$$

and

$$\nabla_{v_{n,k}} L^{(M)} = N^2 \widetilde{\nabla_{v_{n,k}} L_1^{(M)}}.$$

Let $0 < p_4 < 1$. By Lemma H.3, for every $k \in [K]$ there is an event $\mathcal{U}_k$ of probability at least $1 - p_4$ such that for every $n \in [N]$

$$\left| \nabla_{u_{n,k}} L_1 - \frac{M}{N} \widetilde{\nabla_{u_{n,k}} L_1^{(M)}} \right| \leq \frac{\alpha(1 + \Gamma)\mu}{N} \sqrt{\frac{2\log(1/p_4) + 2\log(N) + 2\log(2)}{M}},$$

Similarly, for $0 < p_5 < 1$ for every $k \in [K]$ there is an event $\mathcal{V}_k$ of probability at least $1 - p_5$ such that for every $n \in [N]$

$$\left| \nabla_{v_{n,k}} L_1 - \frac{M}{N} \widetilde{\nabla_{v_{n,k}} L_1^{(M)}} \right| \leq \frac{\alpha(1 + \Gamma)\mu}{N} \sqrt{\frac{2\log(1/p_5) + 2\log(N) + 2\log(2)}{M}},$$

This means that in the event $\mathcal{U}_k$, for every $m \in [M]$ we have

$$\left| \nabla_{u_{n_m,k}} L - \frac{M}{N} \nabla_{u_{n_m,k}} L^{(M)} \right| \leq \alpha(1+\Gamma)\mu N \sqrt{\frac{2\log(1/p_4) + 2\log(N) + 2\log(2)}{M}},$$

and in the event $\mathcal{V}_k$, for every $m \in [M]$ we have

$$\left| \nabla_{v_{n_m,k}} L - \frac{M}{N} \nabla_{v_{n_m,k}} L^{(M)} \right| \leq \alpha(1+\Gamma)\mu N \sqrt{\frac{2\log(1/p_5) + 2\log(N) + 2\log(2)}{M}},$$

Lastly, given $0 < p < 1$, choosing $p_1 = p_4 = p_5 = p/5K$ and $p_2 = p_3 = p/5KD$ and intersecting all events for all coordinates gives in the event $\mathcal{E}$ of probability at least $1 - p$

$$\left| \nabla_{r_k} L - \nabla_{r_k} L^{(M)} \right| \leq 4\alpha(1+\Gamma)\mu N^2 \sqrt{\frac{2\log(1/p) + 2\log(N) + 2\log(K) + 2\log(10)}{M}}$$

$$\left| \nabla_{f_{k,m}} L - \nabla_{f_{k,m}} L^{(M)} \right| \leq \frac{2\beta}{D} \sqrt{\frac{2\log(1/p) + 2\log(K) + 2\log(D) + 2\log(10)}{M}},$$

$$\left| \nabla_{b_{k,m}} L - \nabla_{b_{k,m}} L^{(M)} \right| \leq \frac{2\beta}{D} \sqrt{\frac{2\log(1/p) + 2\log(K) + 2\log(D) + 2\log(10)}{M}},$$

$$\left| \nabla_{u_{n_m,k}} L - \frac{M}{N} \nabla_{u_{n_m,k}} L^{(M)} \right| \leq \alpha(1+\Gamma)\mu N \sqrt{\frac{2\log(1/p) + 2\log(N) + 2\log(K) + 2\log(10)}{M}},$$

and

$$\left| \nabla_{v_{n_m,k}} L - \frac{M}{N} \nabla_{v_{n_m,k}} L^{(M)} \right| \leq \alpha(1+\Gamma)\mu N \sqrt{\frac{2\log(1/p) + 2\log(N) + 2\log(K) + 2\log(10)}{M}},$$

thus proving Theorem H.4.

## H.5 COMPARISON OF NODE SAMPLING AND DIODE SAMPLING

Propositions H.1 and H.4 both provide a probabilistic bound on the difference between the gradients calculated using subgraph SGD and the standard loss (8) using the full graph. Despite this similarity, these approces differ in their sampling schemes and computational tradeoffs.

For Theorem H.4, unlike Theorem H.1, only a subset of entries from $U$ and $V$ are involved in the loss computation per step. This introduces a time–memory tradeoff: since only a subset of $U$ and $V$ is updated in each step, the number of iterations required grows by a factor of $N/M$ in expectation, and the memory consumption reduces by $M/N$, where $M$ denotes the number of sampled nodes. This mild growth in runtime is acceptable, as the number of iterations only grows by a linear scale, while each iteration becomes faster to complete due to the reduced size of the graph.

For Theorem H.1 we sample the diodes of the graph directly. Therefore potentially all nodes of the graph are involved in the calculation of the loss. Using this method, the time and memory complexity of calculating the loss becomes $\mathcal{O}(MK)$, where $M$ is the number of sampled edges.

Notably, the bounds in Theorem H.1 are independent of the graph size, while the bounds in Theorem H.4 improve as the graph becomes more dense. However, in practice, we see that both diode sampling SGD and node sampling SGD provide good IBG approximations that work well for IBG-NN.

## I KNOWLEDGE GRAPHS

### I.1 BASIC DEFINITIONS

**Knowledge graph signals.** We consider knowledge graphs $G = (\mathcal{N}, \mathcal{E}, \mathcal{R})$, where $\mathcal{N}, \mathcal{E}$ and $\mathcal{R}$ represent the set of $N$ nodes, the set of $E$ typed edges, where $\mathcal{E} \subseteq \mathcal{R} \times \mathcal{N} \times \mathcal{N}$, and the set of $R$ relations, correspondingly. Note that in this case there is no signal (i.e. feature matrix). We represent the graphs by an adjacency tensor $\boldsymbol{T} = (t_{i,j,r})_{i,j,r=1}^{N,N,R} \in \mathbb{R}^{N \times N \times R}$. **Frobenius norm.** The *weighted*

*Frobenius norm* of a tensor where its two first dimension are equal $\boldsymbol{D} \in \mathbb{R}^{N \times N \times R}$ with respect to the *weight* $\boldsymbol{Q} \in (0, \infty)^{N \times N \times R}$ is defined to be $\|\boldsymbol{D}\|_{\mathrm{F};\boldsymbol{Q}} := \sqrt{\frac{1}{\sum_{i,j,r=1}^{N,N,R} q_{i,j,r}} \sum_{i,j,r=1}^{N,N,R} d_{i,j,r}^2 q_{i,j,r}}$.

**Cut-norm.** The weighted *tensor cut norm* of $\boldsymbol{D} \in \mathbb{R}^{N \times N \times R}$ with weights $\boldsymbol{Q} \in (0, \infty)^{N \times N \times R}$, is defined to be

$$\|\boldsymbol{D}\|_{\square;\boldsymbol{Q}} = \frac{1}{\sum_{i,j,r} q_{i,j,r}} \sum_{r=1}^{R} \max_{\mathcal{U},\mathcal{V} \subset [N]} \Big| \sum_{i \in \mathcal{U}} \sum_{j \in \mathcal{V}} d_{i,j,r} q_{i,j,r} \Big|.$$

and the definition for the densifying cut similarity follows.

## I.2  APPROXIMATIONS BY INTERSECTING BLOCKS

We define an *Intersecting Block Graph Embedding (IbgE)* with $K$ classes ($K$-IbgE) as a low rank knowledge graph $\boldsymbol{C} \in \mathbb{R}^{N \times N \times R}$ with an adjacency tensor given respectively by

$$\boldsymbol{C}_{:,:,\rho} = \sum_{j=1}^{K} r_{j,\rho} \mathbb{1}_{\mathcal{U}_j} \mathbb{1}_{\mathcal{V}_j}^{\top}$$

where $\boldsymbol{r}_{j,:} \in \mathbb{R}^R$, and $\mathcal{U}_j, \mathcal{V}_j \subset [N]$. We relax the $\{0,1\}$-valued hard source/target community affiliation functions $\mathbb{1}_{\mathcal{S}}, \mathbb{1}_{\mathcal{T}}$ to soft affiliation functions in $\mathbb{R}$ to allow differentiability.

**Definition I.1.** *Let $d \in \mathbb{N}$, and let $\mathcal{Z}$ be a soft affiliation model. We define $[\mathcal{Z}] \subset \mathbb{R}^{N \times N \times R}$ to be the set of all elements of the form $\boldsymbol{u}\boldsymbol{v}^{\top} \otimes \boldsymbol{m}$, with $\boldsymbol{u}, \boldsymbol{v} \in \mathcal{Q}$ and $\boldsymbol{m} \in \mathbb{R}^R$. We call $[\mathcal{Z}]$ the* soft rank-1 intersecting block graph embedding (IbgE) model *corresponding to $\mathcal{Z}$. Given $K \in \mathbb{N}$, the subset $[\mathcal{Q}]_K$ of $\mathbb{R}^{N \times N \times R}$ of all linear combinations of $K$ elements of $[\mathcal{Z}]$ is called the* soft rank-$K$ IbgE model *corresponding to $\mathcal{Z}$.*

In matrix form, an IbgE $\boldsymbol{C} \in \mathbb{R}^{N \times N \times R}$ in $[\mathcal{Z}]_K$ can be represented by a triplet of *source community affiliation matrix* $\boldsymbol{V} \in \mathbb{R}^{N \times K}$, *target community affiliation matrix* $\boldsymbol{U} \in \mathbb{R}^{N \times K}$, and *community relations matrix* $\boldsymbol{M} \in \mathbb{R}^{K \times R}$, that satisfies:

$$\boldsymbol{C}_{:,:,\rho} = \boldsymbol{U} \operatorname{diag}(\boldsymbol{M}_{:,\rho}) \boldsymbol{V}^{\top}$$

where $\rho \in [R]$.

## I.3  FITTING IBGS TO KNOWLEDGE GRAPHS

Given a knowledge graph $G = (\mathcal{N}, \mathcal{E}, \mathcal{R})$ with $N$ nodes and $R$ relations and an adjacency tensor $\boldsymbol{T} \in \mathbb{R}^{N \times N \times R}$ – a true triple is defined as $(\eta, \rho, \tau)$, where $\eta, \tau \in \mathcal{N}$ and $\rho \in [R]$. We define the following soft rank-$K$ intersecting block score function, based on the weighted Frobenius distance:

$$d(\eta, \rho, \tau) = \big\| \delta(\eta, \rho, \tau) - \boldsymbol{u}_{\eta,:} \operatorname{diag}(\boldsymbol{r}_{:,\rho}) \boldsymbol{v}_{:,\tau}^{\top} \big\|_{\mathrm{F};\boldsymbol{Q_T}}^2$$

where the weight matrix $\boldsymbol{Q_T}$ is

$$\boldsymbol{Q_T} = \boldsymbol{Q}_{\boldsymbol{T},\Gamma} := e_{E,\Gamma} \mathbf{1} + (1 - e_{E,\Gamma}) \max_{\rho \in [R]} \boldsymbol{T}_{:,:,\rho},$$

$\delta(h, r, t)$ is 1 if $(h, r, t)$ is a true triplet, otherwise 0, $\boldsymbol{V}_{\tau,:} \in \mathbb{R}^K$ is the source community affiliation of the head entity, $\boldsymbol{U}_{\eta,:} \in \mathbb{R}^K$ is the target community affiliation of the tail entity, $\boldsymbol{R}_{:,\rho} \in \mathbb{R}^K$ is the community relation vector, and $e_{E,\Gamma} = \frac{\Gamma E / N^2}{1 - (E/N^2)}$ with $\Gamma > 0$.

We minimize a margin-based loss function with negative sampling, similar to Sun et al. (2019):

$$L = -\log \sigma \left( \gamma - d(\eta, \rho, \tau) \right) - \sum_{i=1}^{\zeta} \frac{1}{\zeta} \log \sigma \left( d(\eta_i', \rho_i', \tau_i') - \gamma \right)$$

where $\gamma$ is a fixed margin, $\sigma$ is the sigmoid function, $\zeta$ is the number of negative samples and $(\eta_i', \rho_i', \tau_i')$ is the $i$-th negative triplet. An empirical validation of IbgE is provided in Section M.1.2.

## J COMPARISON OF IBGS TO ICGS

In this section we highlight the key advancements and distinctions between our method and its predecessor (Finkelshtein et al., 2024a).

**Approximation of directed graphs.** Graph directionality is crucial for accurately modeling real-world systems where relationships between entities are inherently asymmetric. Many applications rely on directed graphs to capture the flow of information, influence, or dependencies, and ignoring directionality can lead to the loss of critical structural information. One domain which benefits from directionality is spatiotemporal graphs – graphs where the topology is constant over time but the signal varies. For example, in traffic networks Li et al. (2018), where directionality represents the movement of vehicles along roads, traffic congestion in one direction does not necessarily imply congestion in the opposite direction. Thus, ignoring directionality can lead to inaccurate predictions. Another domain is citation networks, where nodes represent academic papers, and directed edges represent citations from one paper to another. The concrete task of prediction of the publication year is highly depedant on the direction of each edge due to the causal nature of the relation between two papers (Rossi et al., 2024). Many more additional domains benefit from directionality, some of which are Email Networks, Knowledge Graphs and Social Networks (Yujie et al., 2023; Changping et al., 2020).

An ICG takes on the form

$$\boldsymbol{C} = \boldsymbol{Q} diag(\boldsymbol{r}) \boldsymbol{Q}^T, \boldsymbol{P} = \boldsymbol{Q} \boldsymbol{F},$$

Where $Q$ is the community affiliation matrix, and $F$ is the community feature matrix. Here, $Q$ is used as both the source and the target affiliation matrix, making the ICG a symmetric, undirected graph. The community affiliation matrix $Q$ also acts as a mapping from the community space to the node space, converting the community feature matrix $F$ to node features.

An IBG takes the form

$$\boldsymbol{C} = \boldsymbol{U} diag(\boldsymbol{r}) \boldsymbol{V}^T, \boldsymbol{P} = \boldsymbol{U} \boldsymbol{F} + \boldsymbol{V} \boldsymbol{B}.$$

IBG generalizes the ICG by using a different affiliation matrix for source nodes $V$, and target nodes $U$, enabling approximation of directed graphs. The use of source and target community affiliation matrices naturally leads to the use of two community feature matrices. The source community feature matrix, $B$, and the target community feature matrix, $F$. Over undirected graphs, the IBG representation converges to the ICG representation.

In our analysis of IBGs, we adopt the simplifying assumption that the approximated graph is unweighted, whereas ICG assumes a weighted approximated graph. This assumption is introduced to provide a clearer derivation of our theoretical guarantees; however, it can be relaxed using the same line of derivation with straightforward modifications to account for edge weights,

**Densifying the adjacency matrix.** The major caveat of ICG-NNs is their limited applicability to sparse graphs. Both their theoretical guarantees and approximation capabilities weaken as sparsity increases, with the approximation error of ICGs being $\mathcal{O}(N/(EK)^{1/2})$. This issue stems from the imbalance between the number of edges and non-edges in sparse graphs. As graphs grow sparser, non-edges dominate, and since standard metrics assign equal weight to edges and non-edges, the approximation shifts from capturing the relational information that we aim to capture – to capturing the absence of relations. To address this limitation, we introduce the densifying cut similarity, a novel similarity measure that explicitly accounts for the structural imbalance in sparse graphs. This similarity measure enables IBG-NNs to efficiently learn a densified representation of sparse graphs, achieving an approximation error of $\mathcal{O}(K^{-1/2})$ for both sparse and dense graphs while preserving the relational information. We emphasize that, unlike ICGs, whose approximation quality is measured with respect to the cut norm, IBGs are evaluated based on the densifying cut similarity.

**Architecture and empirical performance.** At first glance, ICG-NNs and IBG-NNs may appear conceptually similar, as both follow a two-step process: first estimating the graph approximation, followed by processing through a neural network architecture. Both also support a subgraph stochastic gradient descent (SGD) method. However, the fine-grained details of their implementations differ significantly.

In the graph approximation stage, IBG-NNs extend the capabilities of ICG-NNs in two ways. They can approximate directed graphs, and more importantly they minimize a weighted Frobenius

norm, where edges and non-edges are assigned different weights. This contrasts with ICG-NNs, which minimize the standard Frobenius norm. This critical difference enables IBG-NNs to handle sparse graphs more effectively, as demonstrated in Section M.3. In the neural network stage, IBG-NNs offer greater architectural flexibility. While ICG-NNs operate on a single community signal, IBG-NNs incorporate two community signals (source and target) when mapping back to the node space. Although we simplify the architecture by using a basic addition operation, more sophisticated manipulations could be employed. These architectural and methodological advancements result in IBG-NNs' empirical superiority across various domains. Specifically, IBG-NNs outperform ICG-NNs in node classification (see Section 6.2), spatiotemporal property prediction (see Section M.1.1), subgraph SGD on large graphs (see Sections 6.3 and M.1.3), and efficiency in the number of communities used (see Section M.2.1). This broad dominance underscores the effectiveness of IBG-NNs in addressing the limitations of ICG-NNs while delivering state-of-the-art performance. Lastly, we note that IBG-NNs uses DeepSets within each layer. We include this component because it improves performance in practice. For fairness in our experiments, we also evaluated an ICG-NN variant using DeepSets and observed no additional gains.

## K    COMPLEXITY COMPARISON OF IBG-NN AND MPNNS

Our IBG-NN architecture takes $O(D(NK + KD + ND))$ operations at each layer. For comparison, simple MPNNs such as GCN and GIN compute messages using only the features of the nodes, with computational complexity $\mathcal{O}(ED + ND^2)$. More general message-passing layers which apply an MLP to the concatenated features of the node pairs along each edge have a complexity of $\mathcal{O}(ED^2)$. Consequently, IBG neural networks are more computationally efficient than MPNNs when $K < D\mathrm{d}$, where d denotes the average node degree, and more efficient than simplified MPNNs like GCN when $K < \mathrm{d}$.

## L    ADDITIONAL IMPLEMENTATION DETAILS ON IBG-NN

Let us recall the update equation of an IBG-NN for layers $1 \leq \ell \leq L - 1$

$$\boldsymbol{H}_s^{(\ell+1)} = \sigma \left( \Theta_1^s \left( \boldsymbol{H}_s^{(\ell)} \right) + \Theta_2^s \left( \boldsymbol{V} \boldsymbol{B}^{(\ell)} \right) \right),$$

$$\boldsymbol{H}_t^{(\ell+1)} = \sigma \left( \Theta_1^t \left( \boldsymbol{H}_t^{(\ell)} \right) + \Theta_2^t \left( \boldsymbol{U} \boldsymbol{F}^{(\ell)} \right) \right),$$

And finally for layer $L$:

$$\boldsymbol{H}^{(L)} = \boldsymbol{H}_s^{(L)} + \boldsymbol{H}_t^{(L)},$$

In each layer, the learned functions $\Theta_1^s$ and $\Theta_1^t$ are applied to the previous node representations $\boldsymbol{H}_s^{(\ell)}$ and $\boldsymbol{H}_t^{(\ell)}$ seperately, while $\Theta_2^s$ and $\Theta_2^t$ are applied to the post-analysis source community features $\boldsymbol{V} \boldsymbol{B}^{(\ell)}$ and the post-analysis target community features $\boldsymbol{U} \boldsymbol{B}^{(\ell)}$, respectively. To simplify, we restrict the aforementioned architecture of the learned function to linear layers and a pooling operation (e.g. DeepSets (Manzil et al., 2017)), interleaving with a ReLU activation function. An example of an IBG-NN linear layer with mean pooling is:

$$\Theta_1 \left( \boldsymbol{H}^{(\ell)} \right) = \boldsymbol{H}^{(\ell)} \boldsymbol{W}_1^{(\ell)} + \frac{1}{N} \mathbf{1} \mathbf{1}^\top \boldsymbol{H}^{(\ell)} \boldsymbol{W}_2^{(\ell)}$$

with $\boldsymbol{W}_1^{(\ell)}, \boldsymbol{W}_2^{(\ell)} \in \mathbb{R}^{D^{(\ell)} \times D^{(\ell+1)}}$ being learnable weight matrices.

## M    ADDITIONAL EXPERIMENTS

### M.1    ANALYSIS ON VARYING DOMAINS

#### M.1.1    SPATIO-TEMPORAL GRAPHS

**Setup.**    We evaluate IBG-NN on the real world traffic-network datasets METR-LA and PEMS-BAY (Li et al., 2018). We report the baselines DCRNN (Li et al., 2018), GraphWaveNet (Wu et al., 2019),

Table 3: Results on temporal graphs. Top three models are colored by First, Second, Third.

| Model | METR-LA | PEMS-BAY |
|---|---|---|
| # nodes | 207 | 325 |
| # edges | 1515 | 2369 |
| Avg. degree | 7.32 | 7.29 |
| Metrics | MAE | MAE |
| DCRNN | $3.22 \pm 0.01$ | $1.64 \pm 0.00$ |
| GraphWaveNet | $3.05 \pm 0.03$ | $1.56 \pm 0.01$ |
| AGCRN | $3.16 \pm 0.01$ | $1.61 \pm 0.00$ |
| T&S-IMP | $3.35 \pm 0.01$ | $1.70 \pm 0.01$ |
| TTS-IMP | $3.34 \pm 0.01$ | $1.72 \pm 0.00$ |
| T&S-AMP | $3.22 \pm 0.02$ | $1.65 \pm 0.00$ |
| TTS-AMP | $3.24 \pm 0.01$ | $1.66 \pm 0.00$ |
| ICG-NN | $3.12 \pm 0.01$ | $1.56 \pm 0.00$ |
| **IBG-NN** | $3.10 \pm 0.01$ | $1.55 \pm 0.00$ |

Table 4: Comparison with KG completion methods. Top three models are colored by First, Second, Third.

| Method | Model | Kinship | | | UMLS | | |
|---|---|---|---|---|---|---|---|
| | | MRR | Hit@1 | Hit@10 | MRR | Hit@1 | Hit@10 |
| KGE | TransE | 0.31 | 0.9 | 84.1 | 0.69 | 52.3 | 89.7 |
| | DistMult | 0.35 | 18.9 | 75.5 | 0.39 | 25.6 | 66.9 |
| | ComplEx | 0.42 | 24.2 | 81.2 | 0.41 | 27.3 | 70.0 |
| | RotatE | 0.65 | 50.4 | 93.2 | 0.74 | 63.6 | 93.9 |
| | **IbgE** | 0.69 | 55.0 | 95.3 | 0.82 | 71.5 | 96.3 |
| Rule Learning | Neural-LP | 0.30 | 16.7 | 59.6 | 0.48 | 33.2 | 77.5 |
| | DRUM | 0.33 | 18.2 | 67.5 | 0.55 | 35.8 | 85.4 |
| | RNNLogic | 0.64 | 49.5 | 92.4 | 0.75 | 63.0 | 92.4 |
| | RLogic | 0.58 | 43.4 | 87.2 | 0.71 | 56.6 | 93.2 |
| | NCRL | 0.64 | 49.0 | 92.9 | 0.78 | 65.9 | 95.1 |

AGCRN (Bai et al., 2020), T&S-IMP, TTS-IMP, T&S-AMP, and TTS-AMP (Cini et al., 2024), and ICG-NN (Finkelshtein et al., 2024a) all taken from Finkelshtein et al. (2024a). We follow the methodology of Cini et al. (2024), segmenting the datasets into windows of time steps, and training the model to predict the subsequent 12 observations. Each window is divided sequentially into train, validation, and test using a $70\%/10\%/20\%$ split. We report mean average error and standard deviation over 5 different seeds. Finally, we use a GRU to embed the data before using it as input for the IBG-NN model.

**Results.** Table 3 demonstrates that IBG-NNs suppresses ICG-NNs by a small margin. This slight difference could be attributed to the small size of the graph (207 and 325 nodes), where local interactions are likely sufficient for the task, and the ability of IBG-NNs to capture global structure becomes less relevant. This raises questions about the role of directionality in traffic networks, suggesting the need for further investigation. Notably, IBG-NNs show strong performance in another domain, matching the effectiveness of methods specifically designed for spatio-temporal data, such as DCRNN, GraphWaveNet, and AGCRN, despite the small graph size and low edge density.

### M.1.2  KNOWLEDGE GRAPHS

**Setup.** We evaluate IbgE on the the Kinship and UMLS (Kok & Domingos, 2007) datasets. We report the knowledge graph embedding baselines TransE (Bordes et al., 2013), DistMult (Yang et al., 2014), ConvE (Dettmers et al., 2018), ComplEx (Trouillon et al., 2016) and RotatE (Sun et al., 2019). We also report the rule learning baselines Neural-LP (Yang et al., 2017), RNNLogic (Qu et al., 2020), RLogic (Cheng et al., 2022) and NCRL (Cheng et al., 2023) to compare with the state-of-the-art models over these datasets. Results for all baselines were taken from (Cheng et al., 2023).

Table 5: Comparison of node-sampling subgraph SGD with coarsening methods across varying condensation ratios. Top three models are colored by First, Second, Third.

| Condensation ratio | Flickr | | | Reddit | | |
|---|---|---|---|---|---|---|
| | 0.5% | 1% | 100% | 0.1% | 0.2% | 100% |
| Coarsening | $44.5 \pm 0.1$ | $44.6 \pm 0.1$ | $47.2 \pm 0.1$ | $42.8 \pm 0.8$ | $47.4 \pm 0.9$ | $93.9 \pm 0.0$ |
| Random | $44.0 \pm 0.4$ | $44.6 \pm 0.2$ | $47.2 \pm 0.1$ | $58.0 \pm 2.2$ | $66.3 \pm 1.9$ | $93.9 \pm 0.0$ |
| Herding | $43.9 \pm 0.9$ | $44.4 \pm 0.6$ | $47.2 \pm 0.1$ | $62.7 \pm 1.0$ | $71.0 \pm 1.6$ | $93.9 \pm 0.0$ |
| K-Center | $43.2 \pm 0.1$ | $44.1 \pm 0.4$ | $47.2 \pm 0.1$ | $53.0 \pm 3.3$ | $58.5 \pm 2.1$ | $93.9 \pm 0.0$ |
| GCOND | $47.1 \pm 0.1$ | $47.1 \pm 0.1$ | $47.2 \pm 0.1$ | $89.6 \pm 0.7$ | $90.1 \pm 0.5$ | $93.9 \pm 0.0$ |
| SFGC | $47.0 \pm 0.1$ | $47.1 \pm 0.1$ | $47.2 \pm 0.1$ | $90.0 \pm 0.3$ | $89.9 \pm 0.4$ | $93.9 \pm 0.0$ |
| GC-SNTK | $46.8 \pm 0.1$ | $46.5 \pm 0.2$ | $47.2 \pm 0.1$ | – | – | – |
| SimGC | $45.6 \pm 0.4$ | $43.8 \pm 1.5$ | $47.2 \pm 0.1$ | $91.1 \pm 1.0$ | $92.0 \pm 0.3$ | $93.9 \pm 0.0$ |
| ICG-NN | $50.1 \pm 0.2$ | $50.8 \pm 0.1$ | $52.7 \pm 0.1$ | $89.7 \pm 1.3$ | $90.7 \pm 1.5$ | $93.6 \pm 1.2$ |
| **IBG-NN** | $50.7 \pm 0.1$ | $51.2 \pm 0.2$ | $53.0 \pm 0.1$ | $92.3 \pm 1.1$ | $92.3 \pm 0.6$ | $94.1 \pm 0.5$ |

**Results.** Table 4 shows that IbgE achieves state of the art results across all datasets, solidifying IBGs potential as a knowledge graph embedding method. This strong performance is expected, as IbgE is both computationally efficient and expressive, capable of modeling arbitrary head-relation-tail triplets, making it particularly well-suited for modeling knowledge graphs.

### M.1.3 SUBGRAPH SGD USING NODE SAMPLING

**Setup.** In Section 6.3 we test the diode sampling SGD method proposed in Section H. In this section we test the same experiment using the node sampling SGD method (see Section H.1). We follow the setup described in Section 6.3 for Flickr and Reddit, under condensation ratios $r = M/N$, where $N$ is the total number of nodes, and $M$ is the number of sampled nodes. For a condensation ratio of $100\%$, the competing methods correspond to standard GCN.

**Results.** Once again, Table 5 shows IBG-NN using node sampling subgraph SGD outperforms all other coarsening and condensation methods, as well as its predecessor, ICG-NN. These results are consistent with our theoretical guarantees (see Theorems H.1 and H.4), which predict a low approximation error. This demonstrates that IBG-NNs are capable of good approximations even while loading a small fraction of the graph into memory. Additionally, we observe that node-sampling and diode-sampling methods yield nearly identical empirical performance, although their approximation errors decay at different asymptotic rates (see Theorems H.1 and H.4), offering flexibility in choosing the sampling strategy based on practical considerations.

## M.2 THE EFFECT OF NUMBER OF COMMUNITIES

### M.2.1 PERFORMANCE

**Setup.** We evaluate IBG-NN and ICG-NN on the non-sparse Squirrel and Chameleon graph (Pei et al., 2020). We follow the 10 data splits of Pei et al. (2020); Li et al. (2022b) for Squirrel and Chameleon reporting the accuracy and standard deviation.

**Results.** In Figure 3, IBG-NNs exhibit significantly improved performance compared to ICG-NNs. For a small number of communities (10) on Squirrel, IBG-NNs achieve 66% accuracy, whereas ICG-NNs achieve only 45%, further highlighting the superiority of IBG-NNs, allowing it to reach competitive performance while being efficient. The performance gap persists across both datasets, with IBG-NNs continuing to improve as the number of communities increases, whereas ICG-NNs appear to plateau. This trend can be attributed to IBG-NNs ' ability to capture directionality in the graph, allowing them to model increasingly fine-grained asymmetric structures that ICG-NNs cannot represent.

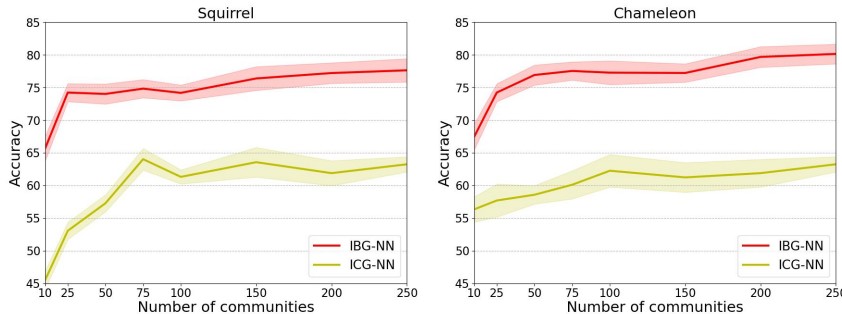

Figure 3: Accuracy of IBG-NNs and ICG-NNs on the Squirrel (**left**) and Chameleon (**right**) datasets as a function of the number of communities.

### M.2.2 APPROXIMATION QUALITY

**Setup.** We evaluate the effect of the number of communities $K$ on the densifying cut similarity of the IBG approximation on the Chameleon and Squirrel datasets. We compare the weighted Frobenius error to the densifying cut similarity for a number of communities ranging from 10 to 250.

**Results.** As shown in Figure 4, both the densifying cut similarity and the weighted Frobenius error of IBG decrease as the number of communities increases. This stands in line with our theory, and demonstrated the bound in Equation (27) also in fact holds in practice.

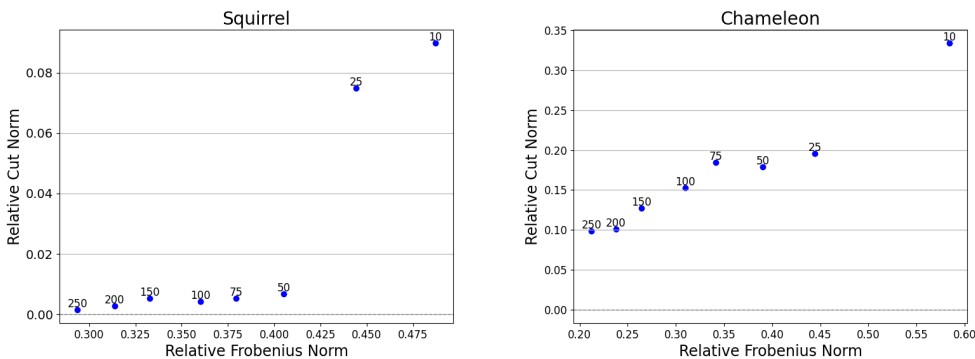

Figure 4: Densifying cut similarity as a function of the weighted Frobenius norm on the Squirrel (left) and Chameleon (right) datasets. The number of communities is specified above every point.

### M.3 THE IMPORTANCE OF DENSIFYING

### M.3.1 DENSIFICATION FOR SPARSE GRAPHS

**Setup.** We evaluate the effect of our proposed densifying lemma by assessing the performance of IBG-NN as a function of $\Gamma$, a parameter that controls the weight assigned to non-edges relative to edges in the IBG approximation. We explore $\Gamma$ values ranging from $0$, where weight is assigned only to existing edges, to $\frac{1-(E/N^2)}{E/N^2}$, where a uniform weight of $1$ is given to each entry in the adjacency matrix. Our experiments are conducted on the large node classification graph Arxiv-Year (Lim et al., 2021) due to its low density (average degree of $6.89$).

**Results.** Figure 5 presents that the perfromance of densified IBG-NNs with $\Gamma = \frac{1-(E/N^2)}{E/N^2}$ equals to that of IBG-NNs learned with standard unweighted loss, matching expectations. More importantly, Figure 5 shows that densified IBG-NNs consistently outperform IBG-NNs learned with standard unweighted loss, across all the densification scales, validating that the theoretical improvements also translate into significant practical benefits.

Table 6: Comparison of node-sampling subgraph SGD with and without densification.

| | Flickr | | | Reddit | | |
|---|---|---|---|---|---|---|
| Condensation ratio | 0.5% | 1% | 100% | 0.1% | 0.2% | 100% |
| ICG-NN | $50.1 \pm 0.2$ | $50.8 \pm 0.1$ | $52.7 \pm 0.1$ | $89.7 \pm 1.3$ | $90.7 \pm 1.5$ | $93.6 \pm 1.2$ |
| IBG-NN (no densification) | $49.6 \pm 0.1$ | $50.1 \pm 0.2$ | $52.1 \pm 0.1$ | $89.3 \pm 1.1$ | $90.9 \pm 0.6$ | $93.4 \pm 0.5$ |
| **IBG-NN** | $\mathbf{50.7} \pm 0.1$ | $\mathbf{51.2} \pm 0.2$ | $\mathbf{53.0} \pm 0.1$ | $\mathbf{92.3} \pm 1.1$ | $\mathbf{92.3} \pm 0.6$ | $\mathbf{94.1} \pm 0.5$ |

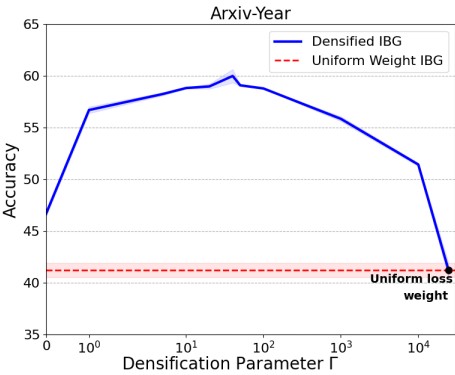

Figure 5: IBG-NN accuracy over the sparse dataset Arxiv-Year as a function of $\Gamma$. The dotted line is IBG-NN accuracy when using standard uniformly weighted loss for the IBG approximation. The rightmost point is the value of $\Gamma$ which results in a uniform cut norm under the densifying loss.

### M.3.2 DENSIFICATION FOR LARGE GRAPHS

**Setup.** We repeat the node-sampling subgraph SGD experiment from Section M.1.3. We compare IBG-NN with and without densification to ICG-NN, evaluating the importance of densification for IBG-NN on the large, undirected graphs Flickr and Reddit.

**Results.** Table 6 reveals that densification significantly impacts the performance of IBG-NN for large graphs. Here, IBG-NN without densification achieves results comparable to ICG-NN. When adding densification, IBG-NN significantly outperforms ICG-NN, further demonstrating the importance of our contribution.

### M.3.3 EFFECT OF DENSIFICATION ON APPROXIMATION QUALITY

**Setup.** We study the approximation error of ICG and IBG on their target metrics, cut metric and densifying cut similarity, when approximating *Erdős-Rényi* graphs with 1000 nodes on a different range of edge probabilities. We set the densification parameter $\Gamma = 1/2p$ when approximating $ER(1000, p)$.

**Results.** Figure 6 clearly demonstrates that the approximation quality of IBG remains consistent across different sparsity levels, while ICG's deteriorates as the graphs grow sparser. This greatly supports our claims that IBGs learn a densified version of the original graph while still sharing the same structure with the original graph.

## M.4 EFFICIENCY ANALYSIS

### M.4.1 MEMORY ANALYSIS

**Setup.** We compare the IBG approximation and IBG-NN forward pass memory complexity to that of GCN. We use *Erdős-Rényi* $ER(n, p = 0.5)$ graphs with up to $7k$ nodes and sample node features uniformly from $U[0, 1]$ with dimension 128. We test IBG-NN and GCN with 3 layers, and hidden and output dimensions of 128.

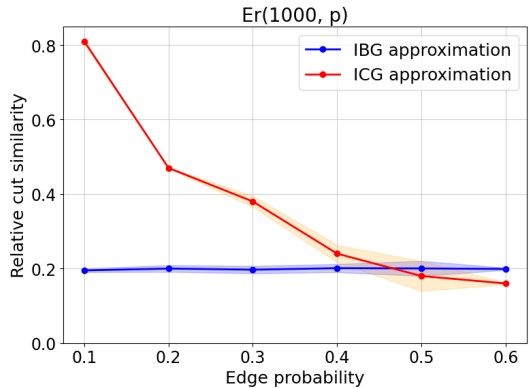

Figure 6: Cut norm and densifying cut similarity of ICG and IBG approximations on $ER(1000, p)$ for different values of $p$.

**Results.** Figure 7 clearly shows IBG's memory complexity scales linearly with GCN, while IBG-NN's memory complexity has a square root relationship with that of GCN. This result stands strongly in line with the results in Figure 2, as both the time and memory complexity of IBG approximation and IBG-NNs are $\mathcal{O}(E)$ and $\mathcal{O}(N)$ respectively.

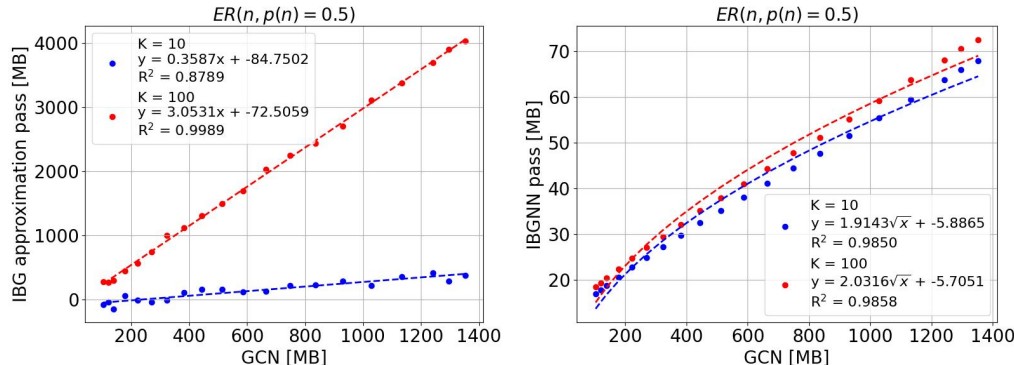

Figure 7: Memory complexity of K-IBG (left) and K-IBG-NN (right) as a function of GCN forward pass on $\mathrm{ER}(n, p = 0.5)$ graphs for K=10, 100.

### M.4.2 TIME UNTIL CONVERGENCE OF IBG APPROXIMATION

**Setup.** We test the time until convergence (in seconds) of the IBG approximation process when using our proposed SVD initialization method (G) vs. when using a random initialization. We consider the IBG representation to have converged if the loss in (8) has not improved by $0.5\%$ for over $500$ epochs. We perform our comparison on the directed graphs Squirrel, Chameleon, and Tolokers, reporting mean time until convergence and standard deviation over $5$ different seeds.

Table 7: Time until convergence in seconds on directed graphs.

|  | Squirrel | Chameleon | Tolokers |
| --- | --- | --- | --- |
| # nodes | 5201 | 2277 | 11758 |
| # edges | 217073 | 36101 | 519000 |
| avg. degree | 41.71 | 15.85 | 88.28 |
| random init. | $139.63 \pm 10.58$ | $101.69 \pm 6.56$ | $184.54 \pm 13.94$ |
| eigenvector init. | $107.20 \pm 2.73$ | $99.22 \pm 7.24$ | $65.89 \pm 0.39$ |

**Results.** Table 7 indicates that SVD initialization consistently converges faster than random initialization across all datasets. Notably, for Tolokers, using SVD initialization results in nearly $3\times$ faster convergence, highlighting the benefits of the method.

## M.5 ADDITIONAL EXPERIMENTS

### M.5.1 HYPERPARAMETER SENSITIVITY

**Setup.** We evaluate the sensitivity of IBG-NN to the number of communities $K$ and the densification parameter $\Gamma$ on three datasets: Squirrel, Chameleon (Pei et al., 2020), and Arxiv-Year (Lim et al., 2021). We train IBGs with varying values of $K$ and $\Gamma$, with $K \in \{25, 50, 100, 200, 250, 400\}$ and $e_{E,\Gamma} \in \{0, 0.01, 0.05, 0.1, 0.25, 1\}$, where $e_{E,\Gamma} = 1$ implies no densification was used. We report test accuracy averaged over the 10 of Pei et al. (2020) splits for Squirrel and Chameleon, and over 5 seeds for Arxiv-Year.

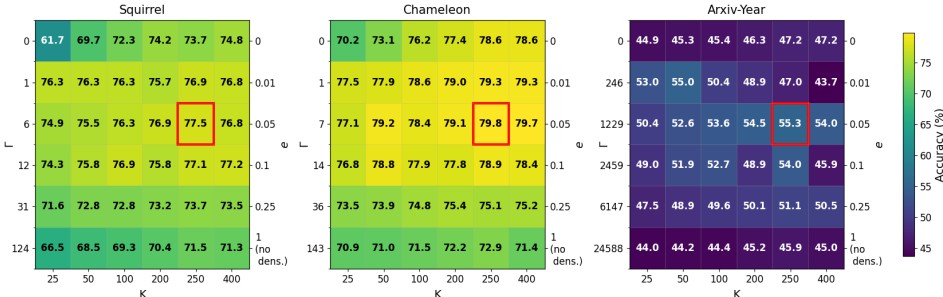

Figure 8: Accuracy of IBG-NN across different node classification benchmarks under varying values of $K$ and $e_{E,\Gamma}$. Top score is marked by a red boarder.

**Results.** Figure 8 shows test accuracy heatmaps for different $(K, \Gamma)$ combinations, with red boxes indicating optimal configurations. The results reveal that $K = 250$ and $e_{E,\Gamma} = 0.05$ consistently achieve top performance across all three datasets. Moreover, a general trend seems to be that $(K, e_{E,\Gamma})$ points that are further away from $(K = 250, e_{E,\Gamma} = 0.05)$ have worse performance than closer ones, suggesting $(K = 250, e_{E,\Gamma} = 0.05)$ as a promising starting point.

The table also showcases $(K = 25, e_{E,\Gamma} = 0.01)$ as a more efficient alternative which uses less communities, while still remaining close in accuracy to the top-performing choice.

### M.5.2 SVD INITIALIZATION

**Setup.** We test the IBG approximation quality and downstream IBG-NN performance when using random initalization compared to SVD initialization. We report results on the directed node classification benchmarks Squirrel, Chamelon, and Tolokers. To evaluate the IBG approximation we report the IBG loss (Equation (8)), with standard deviation over 5 seeds. We report average ROC AUC and standard deviation for Tolokers, and average accuracy and standard deviation for Squirrel and Chameleon, following the 10 splits of Platonov et al. (2023); Pei et al. (2020).

Table 8: Effect of initialization on IBG loss and downstream IBG-NN accuracy.

| | Squirrel | | Chameleon | | Tolokers | |
|---|---|---|---|---|---|---|
| | IBG loss | IBGNN acc. | IBG loss | IBGNN acc. | IBG loss | IBGNN acc. |
| random init. | $0.22 \pm 0.01$ | $77.36 \pm 1.62$ | $0.19 \pm 0.01$ | $79.59 \pm 1.02$ | $0.40 \pm 0.01$ | $83.41 \pm 0.87$ |
| eigenvector init. | $0.23 \pm 0.01$ | $77.41 \pm 1.79$ | $0.17 \pm 0.01$ | $79.65 \pm 1.13$ | $0.40 \pm 0.01$ | $83.40 \pm 0.75$ |

**Results.** As seen in Table 8, SVD initialization results in little to no change in IBG approximation quality and IBGNN performance. This further demonstrates the practicality of our initialization, as it offers improved approximation runtime efficiency with no performance drawback.

### M.5.3 CONVERGENCE OF IBG APPROXIMATION

**Setup.** To validate the stability and efficiency of our proposed method, we analyze the progression of the IBG approximation loss throughout the optimization process. We conduct experiments on three benchmark datasets: Squirrel, Chameleon, and Arxiv-Year. For each dataset, we track the IBG loss over 5,000 epochs, reporting average loss and standard deviation over 5 different seeds.

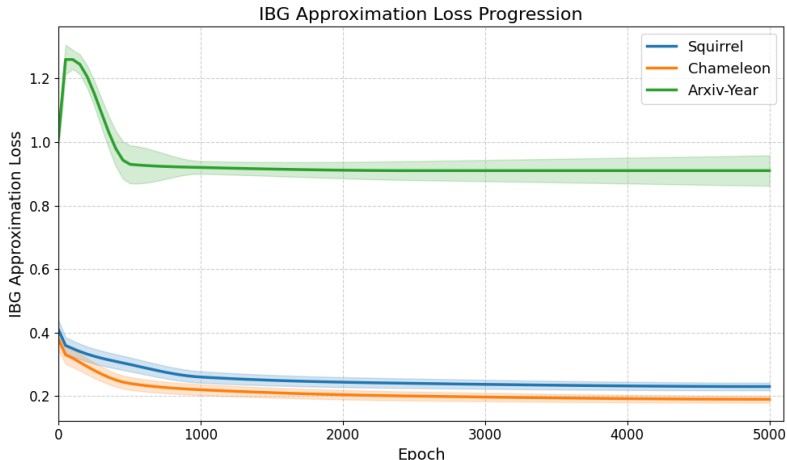

Figure 9: IBG approximation loss during IBG approximation process for Squirrel, Chameleon, and Arxiv-Year.

**Results.** As seen in Figure 9 across all datasets, the loss converges rapidly in the early stages of training, with minimal improvement observed after 1000 epochs. While the loss continues to decrease slowly with additional training, we observe that extending training beyond 10000 epochs rarely leads to additional improvement in downstream IBGNN performance.

### M.5.4 TOPOLOGICAL ANALYSIS

**Setup.** To evaluate how graph structure may affect IBG approximation quality and IBG-NN performance, we analyze several graph properties (Homophily level, Density, Cheeger constant, and Spectral gap) across six datasets, and examine their relationship to the IBG approximation quality and downstream IBG-NN performance.

Table 9: IBG approximation error and downstream performance over .

| | Homophilic | | | Heterophilic | | |
|---|---|---|---|---|---|---|
| Metric | Citeseer-full | Cora-ML | Ogbn-Arxiv | Chameleon | Squirrel | Arxiv-year |
| Homophily | 0.949 | 0.792 | 0.655 | 0.235 | 0.223 | 0.221 |
| Density | 0.0006 | 0.0009 | 0.0001 | 0.0121 | 0.0147 | 0.0001 |
| Cheeger Const. | $\sim 0$ | $\sim 0$ | 0.0004 | 0.0032 | 0.0218 | 0.0004 |
| Spectral Gap | 2.6 | 1.7 | 58.9 | 25.1 | 194.9 | 58.9 |
| IBG Approx. Error ↓ | 0.56 | 0.75 | 0.89 | 0.17 | 0.23 | 0.91 |
| IBG-NN Performance ↑ | 92.40 | 84.20 | 65.30 | 80.15 | 77.63 | 60.14 |

**Results.** Table 9 reveals that, despite the differences across the homophily level, density, Cheeger constant, and spectral gap, IBG-NN performance shows no clear correlation with these properties. While IBG approximation error is lower for denser graphs such as Chameleon and Squirrel, this does not directly translate to superior IBG-NN performance. These findings may suggest that graph structure alone does not determine IBG-NN performance, as factors like node feature quality and task-specific characteristics also play crucial roles.

## N CHOICE OF DATASETS

We present IBG-NNs, a method best suited for large graphs, capable of effectively approximating directional graphs. Consequently, our primary experiments in Section 6.3 follow the standard graph coarsening and condensation benchmarks Reddit (Hamilton et al., 2017a), Flickr (Zeng et al., 2019), Ogbn-Arxiv and products (Hu et al., 2020), where IBG-NNs demonstrates state-of-the-art

performance. We note that the large-graph benchmarks we use are undirected, as there are no commonly adopted large-scale directed graph datasets.

Additionally, IBG-NNs are used for handling directed graphs. To evaluate its performance in this domain, we conduct experiments on standard directed graph datasets Squirrel, Chameleon (Pei et al., 2020), and Tolokers (Platonov et al., 2023), where it also achieves state-of-the-art results. Finally, our graph approximation method proves particularly efficient for spatio-temporal datasets, where a fixed topology supports time-varying signals. We validate this by experimenting with IBG-NNs on the METR-LA and PEMS-BAY Li et al. (2018) datasets, where it matches the effectiveness of methods that are specifically designed for spatio-temporal data.

## O  DATASET STATISTICS

The dataset statistics of the real-world directed graphs, spatio-temporal, graph coarsening and knowledge graph benchmarks used are presented below in Tables 10 to 13.

Table 10: Non-sparse node classification dataset statistics.

|  | Squirrel | Chameleon | Tolokers |
|---|---|---|---|
| # nodes (N) | 5,201 | 2,277 | 11,758 |
| # edges (E) | 217,073 | 36,101 | 519,000 |
| Avg. degree ($\frac{E}{N}$) | 41.71 | 15.85 | 88.28 |
| # node features | 2089 | 2325 | 10 |
| # classes | 5 | 5 | 2 |
| Metrics | Accuracy | Accuracy | ROC AUC |

Table 11: Spatio-temporal dataset statistics.

|  | METR-LA | PEMS-BAY |
|---|---|---|
| # nodes (N) | 207 | 325 |
| # edges (E) | 1,515 | 2,369 |
| Avg. degree ($\frac{E}{N}$) | 7.32 | 7.29 |
| # node features | 34272 | 52128 |
| Metrics | MAE | MAE |

Table 12: Graph coarsening dataset statistics.

|  | Flickr | Reddit | Ogbn-Arxiv | Products |
|---|---|---|---|---|
| # nodes (N) | 89,250 | 232,965 | 169,343 | 2,449,029 |
| # edges (E) | 899,756 | 114,615,892 | 1,166,243 | 61,859,140 |
| Avg. degree ($E/N$) | 10.08 | 491.99 | 6.89 | 25.26 |
| # node features | 500 | 602 | 128 | 100 |
| # classes | 7 | 41 | 40 | 47 |
| Metrics | Accuracy | Micro-F1 | Accuracy | Accuracy |

Table 13: Knowledge graph completion dataset statistics.

|  | Kinship | UMLS |
|---|---|---|
| # entities | 104 | 135 |
| # relations | 25 | 46 |
| # training triples | 8,544 | 5,216 |
| # validation triples | 1,068 | 652 |
| # testing triples | 1,074 | 661 |
| Avg. train. degree | 82.15 | 38.64 |

## P  HYPERPARAMETERS

All experiments are conducted on a single NVIDIA L40 GPU, using the Adam optimizer.

In Tables 14 to 17, we report the hyper-parameters used in our real-world directed graphs, spatio-temporal, graph coarsening and knowledge graph completion benchmarks.

For our spatio-temporal experiments, we utilize the time of the day and the one-hot encoding of the day of the week as additional features, following Cini et al. (2024). Additionaly, for spatio-temporal graphs we usually ignore the signal when fitting the IBG to the graph, but one can also concatenate random training signals and reduce their dimension to obtain one signal with low dimension $D$ to be used as the target signal for the IBG optimization.

Table 14: Non-sparse node classification hyperparameters.

|  | Squirrel | Chameleon | Tolokers |
|---|---|---|---|
| # communities | 250 | 250 | 50 |
| Encoded dim | 128 | 128 | - |
| $\beta/\alpha$ | 1 | 1 | 1 |
| $\Gamma$ | 20 | 5 | 5 |
| Approx. lr | 0.03 | 0.03 | 0.03 |
| Approx. epochs | 10000 | 10000 | 10000 |
| # layers | 7 | 6 | 4 |
| Hidden dim | 128 | 128 | 128 |
| Dropout | 0.2 | 0.2 | 0.2 |
| Residual connection | - | ✓ | ✓ |
| Jumping knowledge | Cat | Cat | Max |
| Normalization | True | True | True |
| Fit lr | 0.003 | 0.003 | 0.003 |
| Fit epochs | 1500 | 1500 | 1500 |

Table 15: Spatio-temporal node regression hyperparameters.

|  | METR-LA | PEMS-BAY |
|---|---|---|
| # communities | 50 | 100 |
| Encoded dim | - | - |
| $\beta/\alpha$ | 0 | 0 |
| $\Gamma$ | 0 | 0 |
| Approx. lr | 0.01, 0.05 | 0.01, 0.05 |
| Approx. epochs | 10000 | 10000 |
| # layers | 6 | 6 |
| Hidden dim | 128 | 128 |
| Dropout | 0.0 | 0.0 |
| Residual connection | ✓ | ✓ |
| Jumping knowledge | Max | Cat |
| Normalization | True | False |
| Fit lr | 0.003 | 0.001 |
| Fit epochs | 300 | 300 |

Table 16: Graph coarsening node classification hyperparameters. On Flickr and Reddit we use 750 communities for a condensation ratio of 100%, and 50 communities for the other settings.

| | Reddit | Flickr | Ogbn-Arxiv | Products |
|---|---|---|---|---|
| # communities | 50, 750 | 50, 750 | 50 | 50 |
| Encoded dim | – | – | – | – |
| $\beta/\alpha$ | 0 | 0 | 0 | 0 |
| $\Gamma$ | 5 | 5 | 5 | 5 |
| Approx. lr | 0.05 | 0.05 | 0.05 | 0.05 |
| Approx. epochs | 1000 | 1000 | 2500 | 2500 |
| # layers | 4 | 4 | 3 | 4 |
| Hidden dim | 128 | 256 | 256 | 128 |
| Dropout | 0 | 0 | 0 | 0 |
| Residual connection | ✓ | – | ✓ | ✓ |
| Jumping knowledge | Max | Cat | Max | Max |
| Normalization | True | True | True | True |
| Fit lr | 0.003 | 0.003 | 0.003 | 0.003, 0.005 |
| Fit epochs | 1500 | 1500 | 1500 | 1500 |

Table 17: Knowledge graph completion hyperparameters.

| | Kinship | UMLS |
|---|---|---|
| # communities | 20 | 15 |
| Encoded dim | 24 | 48 |
| Approx. lr | 0.05 | 0.003 |
| Approx. epochs | 250 | 750 |
| # negative samples | 64 | 128 |

