# OpenReview forum: "Efficient Learning on Large Graphs using a Densifying Regularity Lemma"
_ICLR.cc/2026/Conference — ICLR 2026 Poster_

### Official Review · Reviewer_Besy · 2025-10-27

**Soundness:** 4
**Presentation:** 4
**Contribution:** 4
**Rating:** 8
**Confidence:** 3

**Summary:**

The paper works with graph approximations roughly based on the stochastic block model, i.e. approximating the adjacency matrix of the original graph as the sum of multiple blocks, i.e. bipartite subgraphs where all source nodes are connected to all target nodes.
In contrast to previous work where the number of blocks needed to achieve a given approximation quality depends on the sparsity of the graph, the paper guarantees this depending only on the approximation quality itself.
They also showed that good enough approximations can practically be found in reasonable time and thus a GNN based on the approximation is able to run in time depending only on the number of nodes (and blocks) instead of linear in the number of edges compared to message passing.
The implemented GNN achieves very good performance over a number of tasks on relatively large graphs.

**Strengths:**

- a much improved regularity lemma and corresponding metric that greatly improves over the predecessor architecture
 - a fast enough algorithm to compute the graph approximation
 - complemented by a GNN building on the graph approximation achieving SOTA performance on a variety of datasets
 - there is a vast and clean appendix explaining all the details of the paper and sufficient background.
 - The paper is easy to read for a theory paper

**Weaknesses:**

Overall, the paper is very easy to read (especially for a theory paper). I only have a few smaller remarks:

- It could be more clear whether the representation in terms of blocks for IBG and ICG is actually the same. If there is a discussion about it in the paper, I missed it and glancing over the ICG paper the difference is not immediately clear. (As, following this question is the exact role of $\textbf{b}$ in the definition of $\textbf P$, as this seems to be the main modification). The description in the appendix could also include pointers to how exactly the differences show themselves in the definitions.
- mapping the statement of Thm 4.1 to the claims about the contributions roughly works, but a 2-line explaination below would have been nice as well. But I guess due to space, this won't be possible.
- in 163 in the definition of the weighted cut: what does the i,j in the normalization factor range over? Is it all of Q or just U,V? (potentially use different indices within and outside of the max to make the difference more clear, given that its indeed ranging over all of Q)
- the workings of the IBG NN compared to a MPGNN is not clear (344ff). Is it doing something else than running a plain directed GNN on the approximated graph, taking into account the low-rank decomposition to make it fast? Or does the model behave differently? This is also relevant for the context in which the "simple and efficient operations" in 411 are to be understood.

Minor:
- would it be possible to add references to the sources of each line in table 1? This would also make it clear which of the computations are new. (I know that it is described in the text later on, but simple hyperlinks could help as well and in the text)

Typos:
248: soft indicator model -> soft affilitation model?
444: kipf and welling -> citep
461: performance

**Questions:**

see weaknesses

---

> ### Author Response · Authors · 2025-11-19
>
> We thank the Reviewer for acknowledging the novelty in our work, and the clarity of our mathematical presentation.
>
> >W1: It could be more clear whether the representation in terms of blocks for IBG and ICG is actually the same. If there is a discussion about it in the paper, I missed it and glancing over the ICG paper the difference is not immediately clear. (As, following this question is the exact role of **$b$** in the definition of **$P$**, as this seems to be the main modification). The description in the appendix could also include pointers to how exactly the differences show themselves in the definitions.
>
> Thank you for your suggestion. We agree that this point can be further clarified. Therefore, we have added to the updated manuscript the following description as part of our comparison between IBG and ICG (appendix J, lines 2340 - 2352):
>
> An ICG takes the form
>
> &nbsp;&nbsp;&nbsp;&nbsp;&nbsp;&nbsp; $C=Qdiag(r)Q^T, P=QF$,
>
> where $Q$ is the community affiliation matrix, and $F$ is the community feature matrix.
> Here, $Q$ is used as both the source and the target affiliation matrix, making the ICG a symmetric, undirected graph. The community affiliation matrix $Q$ also acts as a mapping from the community space to the node space, converting the community feature matrix $F$ to node features.
>
> An IBG takes the form
>
> &nbsp;&nbsp;&nbsp;&nbsp;&nbsp;&nbsp; $C=Udiag(r)V^T, P=UF+VB$.
>
> IBG generalizes the ICG by using a different affiliation matrix for source nodes $V$, and target nodes $U$, enabling approximation of directed graphs. The use of source and target community affiliation matrices naturally leads to the use of two community feature matrices. The source community feature matrix, $B$, and the target community feature matrix, $F$. Over undirected graphs, the IBG representation converges to the ICG representation.
>
>
> >W2: mapping the statement of Thm 4.1 to the claims about the contributions roughly works, but a 2-line explaination below would have been nice as well. But I guess due to space, this won't be possible.
>
> We have added in the updated manuscript the following explanation below Thm 4.1:
>
> "Specifically, the three main ways in which Theorem 4.1 generalizes ICG to IBG are depicted in: (1) The use of directional IBGs, (2) the deterministic certificate for the high probability event given by Item 1, and (3) the approximation bound in Item 2, which is independent of graph sparsity.
>
> >W3: in 163 in the definition of the weighted cut: what does the i,j in the normalization factor range over? Is it all of Q or just U,V? (potentially use different indices within and outside of the max to make the difference more clear, given that its indeed ranging over all of Q)
>
> The indices $i,j$ in the normalization factor $\sum_{i,j} q_{i,j}$ range over all of $Q$. Since $Q$ has already been defined earlier, this is standard mathematical notation where the summation implicitly runs over all valid indices of the matrix.
>
> >W4: the workings of the IBG NN compared to a MPGNN is not clear (344ff). Is it doing something else than running a plain directed GNN on the approximated graph, taking into account the low-rank decomposition to make it fast? Or does the model behave differently? This is also relevant for the context in which the "simple and efficient operations" in 411 are to be understood.
>
> IBGNN should not be interpreted as a message passing network. It does not perform message aggregation or propagation along the edges of an adjacency matrix. Instead, the network operates solely on the graph signal, and leverages the community affiliation matrices $U$ and $V$ to enrich the learned node representations with the structural information of the IBG.
>
> In practice, IBGNNs operate on the node and community signals using low-rank matrix multiplications and summations, avoiding explicit inefficient message passing operations on large adjacency matrices. This design justifies the “simple and efficient operations” claims referred to in line 411.
>
>
> >W5: would it be possible to add references to the sources of each line in table 1? This would also make it clear which of the computations are new. (I know that it is described in the text later on, but simple hyperlinks could help as well and in the text)
>
> We have added a reference to each source in the updated manuscript.
>
> Typos:
> > 248: soft indicator model -> soft affilitation model?
> > 444: kipf and welling -> citep
> > 461: performance
>
> We have corrected these typos in the updated manuscript (lines 260, 454, and 466 in the updated file).
>
> **We have added all the new suggestions provided by the Reviewer in the updated manuscript with the appropriate discussion.**
>
> We hope our responses address the Reviewer's concerns.

---

> > ### Author Response · Authors · 2025-11-25
> >
> > **Thank you for suggesting we clarify the foundamental paradigm shift of IBG-NNs compared to MPNNs.** This work can lead to a new set of algorithms designed specifically for large graphs. Clarifying the difference between our approach and standard MPNNs is essential to positioning our paradigm as the new go-to solution. This makes the paper much more valuable for both practitioners and researchers, reaching a wider audience.
> >
> > We believe we have addressed all of your concerns and **would highly appreciate your response** to ensure no outstanding issues remain.

---

### Official Review · Reviewer_jEYc · 2025-10-31

**Soundness:** 3
**Presentation:** 2
**Contribution:** 3
**Rating:** 6
**Confidence:** 2

**Summary:**

This paper proposes **Intersecting Block Graphs (IBGs),** low-rank directional graph representation allowing the efficient learning on large and even sparse graphs, by squeezing the adjacency matrix into a smaller matrix regarding communities. Additionally, the authors introduce a densifying cut similarity metric and prove a corresponding weak regularity lemma for their claim. The method is validated with various benchmarks with complexity analysis.

**Strengths:**

1. Theorems and lemmas introduced (quality, clarity)

Section 4 introduces the semi-constructive weak regularity lemma and its proof, developed upon prior work.

2. Competitive performance (significance)

For node classification, the spatio-temporal graph, and knowledge graph, show competitive performance against the baselines. Additionally, it is relevant for sparse and dense graphs.

3. Efficient architecture (significance)

The proposed method achieves a time complexity proportional to the number of nodes instead of edges, i.e., the complexity of MPNNs.

**Weaknesses:**

I do not find any critical weakness

Minor

- figure 1 is not mentioned
- line 218 - IBG, IGBS are both used

**Questions:**

1. SVD initialization (section M)

While the SBD initialization was used against random initialization for faster convergence, I don’t see any performance comparison. Does the performance of two initialization eventually end up in the same?

---

> ### Author Response · Authors · 2025-11-19
>
> We sincerely thank the reviewer for recognizing the quality of our theory and the significance of our results through performance and efficiency. We also thank the reviewer for stating that he found no critical weakness in our work.
>
> >W1: "figure 1 is not mentioned"
>
> We have added the following in the introduction section of the updated manuscript (line 77):
>
> "... where the different communities can overlap (see Figure 1 for a visualization of the approximating graph).".
>
> >W2: "line 218 - IBG, IGBS are both used"
>
> Line 218 states:
>
> "Next, we relax the $\{0,1\}$-valued hard indicator functions $1_U, 1_V$ to *soft affiliation functions* with values in $R$, as defined next, to allow continuously optimizing IBGs."
>
> "IGBS" does not seem to appear in our manuscript. We would appreciate some clarification regarding this weakness. We are committed to addressing all of your concerns.
>
>
> >Q1: "While the SVD initialization was used against random initialization for faster convergence, I don’t see any performance comparison. Does the performance of two initialization eventually end up in the same?"
>
>
> To address your question, we compare the IBG approximation convergence time, IBG approximation loss (Section 4.4, Eq. 8), and downstream IBGNN performance, for both random and SVD initialized IBGs.
>
> | Init. | Squirrel | | | Chameleon | | | Tolokers | | |
> |---|---|---|---|---|---|---|---|---|---|
> | | Time (s) | IBG loss | IBGNN acc. | Time (s) | IBG loss | IBGNN acc. | Time (s) | IBG loss | IBGNN acc. |
> | random | 139.63 ± 10.58 | 0.22 | 77.36 ± 1.62 | 101.69 ± 6.56 | 0.19 | 79.59 ± 1.02 | 184.54 ± 13.94 | 0.40 | 83.41 ± 0.87 |
> | eigenvector (SVD) | 107.20 ± 2.73 | 0.23 | 77.41 ± 1.79 | 99.22 ± 7.24 | 0.17 | 79.65 ± 1.13 | 65.89 ± 0.39 | 0.40 | 83.40 ± 0.75 |
>
> As seen in the table, SVD initialization results in little to no change in IBG approximation quality and IBGNN performance. This further demonstrates the practicality of our initialization, as it offers improved approximation runtime efficiency with no performance drawback.
>
> To further test the similarity between the resulting IBGs, we calculate the Frobenius norm between the IBG initialized randomly and the IBG initialized with SVD initialization.
>
>
> | Dataset| Frob. norm of difference |
> |-|-|
> | Squirrel  | 0.02|
> | Chameleon | 0.03|
> | Tolokers  | 0.01|
>
> As seen in the table, the learned IBGs under both initializations are very similar, indicating SVD initialization does not significantly change the final result of the IBG-approximation optimization process.
>
> **We have added all the experimental results and suggestions provided by the Reviewer in the updated manuscript with the appropriate discussion.**
>
> We hope our responses address your concerns and would greatly appreciate your reconsideration of the evaluation and increasing your score.

---

> > ### Author Response · Authors · 2025-11-25
> >
> > **Thank you for suggesting a performance comparison between SVD and random initializations, which validates that both yield the same IBG approximation and downstream performance.**
> >
> > We believe to have addressed all of your concerns and **would highly appreciate your response** as this would give us a chance to address any remaining issues. Thank you.

---

> > ### Comment · Reviewer_jEYc · 2025-11-27
> >
> > I thank the authors for the additional experiment results.
> >
> > For line 218, I mistaken seeing a typo for the wording IGB and IGBs being both used. Sorry for the confusion.
> >
> > For Q1, the results indeed show the SVD initialization shows faster convergence against the random initialization, with almost identical results. I have checked the updated manuscript, and have no more concerns.
> >
> > I have raise by score accordingly to 8.

---

> ### Author Response · Authors · 2025-11-28
>
> We are glad that our rebuttal addressed your concerns, and **we sincerely thank you for raising your score to 8.**

---

### Official Review · Reviewer_zVnQ · 2025-10-31

**Soundness:** 3
**Presentation:** 3
**Contribution:** 3
**Rating:** 6
**Confidence:** 2

**Summary:**

This paper presents a method for approximating a large graph that makes the GNN computation on the approximated graph linear with respect to the number of edges, representing a significant improvement over existing graph reduction methods. The approach compresses the original graph by extracting several communities that may overlap and have internal connections represented as bipartite graphs, referred to as blocks or directed communities. Specifically, it employs an extended version of WRL to solve the approximation with strong practical performance, although the theoretical guarantee may be limited.

**Strengths:**

1. The large computation required by MPNN or GNN on large graphs is indeed a problem, which hinders our understanding and analysis of large social networks, user item interactions in large recommendation networks, and similar systems.
2. The proposed method is not restricted to undirected graphs, unlike ICG. The objective function can be efficiently optimized using gradient descent.
3. The number of communities in the method is independent of any property of the graph, including the number of nodes and the sparsity level.

**Weaknesses:**

1. The semi constructive nature of the optimization is a problem. Then what guarantees do you have, or at least under what conditions of the graph structure, for instance, would you have confidence in the optimization results with guarantees? This would be essential, even though you claim in my comment strength 3)
2. Explicitly encouraging density is counterintuitive to me, since it is usually natural to think that in real world graphs, many edges are consequences of other edges. This makes the sparsification of graphs under certain principles appealing. Could you elaborate on your thoughts about this?
3. Does the proposed IBC happen to align with some graph assumptions in real world data? Since in various real world tasks, IBC performs well with its ability to compress the graph by extracting community wise bipartite graphs. Beyond the numerical improvements, it would be very interesting and necessary to include some real world examples in those datasets explaining why condensing the graph in this way would improve performance even beyond the original one, for both heterophilic (I think the performance is very good) and homophilic graphs.

**Questions:**

1. Typo of the number of the edges in line 38 (not it’s smaller than the number of nodes)?
2. Elaborate on `special interpretation' in line 72.
3. The intuition of the weight cut norm can be introduced in line 83 to improve the readability.

---

> ### Author Response · Authors · 2025-11-19
>
> We thank the reviewer for their comments and for acknowledging the theoretical significance of the number of communities being independent of any graph property.
>
> >W1: "The semi constructive nature of the optimization is a problem. Then what guarantees do you have, or at least under what conditions of the graph structure, for instance, would you have confidence in the optimization results with guarantees?"
>
> That is a good point. As in many other machine learning tasks, the convergence of the gradient descent optimization is not guaranteed, as the (IBG) loss is nonconvex.
>
> Gradient-based minimization on nonconvex losses is very standard in machine learning, and in practice, we see the method leads to IBGNNs with state of the art results across a wide range of datasets and tasks. To demonstrate the optimization behavior in practice, we present the IBG loss curve across multiple datasets in the table below.
>
> | Dataset \ Epoch | 1 | 100 | 500 | 1000 | 2500 | 5000
> |-|-|-|-|-|-|-|
> | Squirrel | 0.41|  0.35|  0.30|  0.26|  0.24|  0.23|
> | Chameleon | 0.38| 0.32 | 0.24|  0.22|  0.2| 0.19|
> | Arxiv-Year | 1.44| 0.96| 0.93|  0.92|  0.91| 0.91|
>
> As seen in the results across all datasets, the loss converges rapidly in the early stages of training, with minimal improvement observed after 1000 epochs. While the loss continues to decrease slowly with additional training, we observe that extending training beyond 10000 epochs rarely leads to additional improvement in downstream IBGNN performance.
>
> We have included these results in the appendix of the updated manuscript (Appendix M.5, p. 52).
>
> Finally, we note that algorithms for approximating IBGs (with no node features) with guarantees do exist, but they are computationally prohibitive, requiring exponential runtime in the number of communities [1]. Instead, our approach is both efficient, and achieves strong results across multiple domains (Node classification, spatiotemporal, knowledge graph link prediction).
>
> >W2: "Explicitly encouraging density is counterintuitive to me, since it is usually natural to think that in real world graphs, many edges are consequences of other edges. This makes the sparsification of graphs under certain principles appealing. Could you elaborate on your thoughts about this?"
>
> We agree that it is natural to think that in real-world graphs, many edges are derivative of others. As a result, message-passing neural networks (MPNNs) often aggregate redundant, correlated information from neighbors, making graph sparsification appealing.
>
> However, our method differs fundamentally from standard MPNNs. We do not use the input adjacency to propagate messages. Instead, we solely use the adjacency matrix in the IBG approximation stage. The learned IBG is then combined with linear layers and pointwise nonlinearities to form a neural network, IBG-NN (which does not follow the neighborhood message-passing scheme).
>
> The adjacency matrix influences the IBG approximation and, through it, the IBG-NN. The IBG approximation stage tries to “cover” the adjacency matrix using rank-1 blocks. We refer the Reviewer to Figure 1 for a visualization.
>
> When the raw adjacency is very sparse i.e., exhibits a “salt-and-pepper” pattern of isolated edges -- the optimal block cover fragments into many tiny blocks, making the approximation harder and less faithful. In contrast, our densification assigns small non-zero weights to non-edges, which reduces the high “salt-and-pepper” contrast, and yields larger, dense blocks better capturing the global patterns in the adjacency matrix.
>
> [1] Quick approximation to matrices and applications

---

> > ### Author Response · Authors · 2025-11-19
> >
> > >W3: "Does the proposed IBG happen to align with some graph assumptions in real world data? Since in various real world tasks, IBG performs well with its ability to compress the graph by extracting community wise bipartite graphs. Beyond the numerical improvements, it would be very interesting and necessary to include some real world examples in those datasets explaining why condensing the graph in this way would improve performance even beyond the original one, for both heterophilic (I think the performance is very good) and homophilic graphs."
> >
> > That is an interesting question. To address this, we analyzed several graph properties (e.g., homophily level, density, Cheeger constant, spectral gap) across six datasets, and examined their relationship to the IBG approximation quality and downstream IBGNN performance.
> >
> > |||Homophilic|||Heterophilic||
> > |-|-|-|-|-|-|-|
> > || **Citeseer-full** | **Cora-ML** | **Ogbn-arxiv** | **Chameleon** | **Squirrel** | **Arxiv-Year** |
> > |Homophily| 0.949 | 0.792 | 0.655| 0.235| 0.223| 0.221|
> > |Density| 0.0006 | 0.00094 | 0.000081 | 0.012107 | 0.014668 | 0.000081 |
> > |Cheeger const.| ~0| ~0| 0.000355| 0.0032| 0.0218| 0.000355|
> > |Spectral Gap| 2.6| 1.7| 58.9| 25.1| 194.9| 58.9|
> > |**IBG approximation error ↓** | 0.56| 0.75| 0.89| 0.17| 0.23| 0.91|
> > |**IBGNN downstream performance ↑** | 92.4| 84.2| 65.3| 80.15| 77.63| 60.14|
> >
> > Despite the differences across the homophily level, density, Cheeger constant, and spectral gap, downstream performance shows no clear correlation with these properties. While IBG approximation error is lower for denser graphs (Chameleon, Squirrel), this does not directly translate to superior IBGNN performance.
> > Similarly, while prior work [2] shows directionality benefits heterophilic graphs, our results in Table 1 of the paper reveal inconsistent gains, where Squirrel and Chameleon improve substantially ($\geq 5$%) when adding directionality, while the heterophilic Tolokers dataset shows no added improvement.
> >
> > **These findings may suggest that graph structure alone does not determine IBGNN performance, as factors like node feature quality and task-specific characteristics also play crucial roles.**
> >
> > We have included these results in the appendix of the updated manuscript (Appendix M.5, p. 52).
> >
> > >Q1: "Typo of the number of the edges in line 38 (not it’s smaller than the number of nodes)?"
> >
> > This is not a typo. We meant the number of edges is **$10^2 \sim 10^3$ times** the number of nodes ("$10^2 \sim 10^3$ as many edges." l. 38).
> >
> > >Q2: "Elaborate on 'special interpretation' in line 72."
> >
> > The special interpretation we referred to is described in lines 72-77 that follow after line 72:
> > "The approximating graph consists of a set of overlapping bipartite components. Namely, there is a set of $K\ll N$ pairs of node communities... where the different communities can overlap.".
> >
> > We have clarified this in the updated manuscript by explicitly signaling that the following lines elaborate on this interpretation, and added a reference in this paragraph (in line 77) to Figure 1, which visualizes this interpretation.
> >
> > >Q3: "The intuition of the weight cut norm can be introduced in line 83 to improve the readability."
> >
> > We have added the following paragraph in line 84 of the updated manuscript:
> >
> > "The cut norm is a well-established graph similarity measure that quantifies the maximum discrepancy in their connectivity structure. It enables graph approximations with rank independent of the number of nodes for dense graphs. However, for sparse graphs, the standard cut-metric is dominated by non-edges, which degrades approximation quality. This motivates the use of a weighted cut-norm that balances the contributions of edges and non-edges.".
> >
> > **We have added all the experimental results and suggestions provided by the Reviewer in the updated manuscript with the appropriate discussion.**
> >
> > We hope our responses address your concerns and would greatly appreciate your reconsideration of the evaluation and increasing your score.
> >
> > [2] Edge Directionality Improves Learning on Heterophilic Graphs.

---

> ### Author Response · Authors · 2025-11-25
>
> **Thank you for suggesting we validate that the non-convex IBG-approximation process converges. This is further indicated in our experiments, where the IBG approximation and the IBGNN performance shows no improvements with additional optimization of the IBG.**
>
> We believe to have addressed all of your concerns and **would highly appreciate your response** as this would give us a chance to address any remaining issues. Thank you.

---

> > ### Comment · Reviewer_zVnQ · 2025-11-25
> >
> > I genuinely thank the authors for their responses, and most of my cncerns have been addressed by their frank answers and responses. Therefore, I increase my score to 8.

---

> ### Author Response · Authors · 2025-11-28
>
> We are glad that our rebuttal addressed your concerns, and **we sincerely thank you for raising your score to 8.**

---

### Official Review · Reviewer_tVWW · 2025-11-04

**Soundness:** 3
**Presentation:** 3
**Contribution:** 3
**Rating:** 8
**Confidence:** 3

**Summary:**

This paper proposes a novel low-rank approximation method for general directed graphs, called Intersecting Block Graph (IBG), designed for efficient graph signal processing. IBG is a non-trivial extension of the Intersecting Community Graph (ICG), overcoming key limitations such as poor performance on sparse graphs and the restriction to undirected graphs. By introducing a densifying cut similarity and an efficient semi-constructive weak regularity lemma, IBG achieves accurate approximations with rank independent of graph size or sparsity. The paper also presents an efficient gradient-based algorithm for fitting IBGs to large directed graphs. Building on this representation, the proposed IBG-NN architecture delivers state-of-the-art results across multiple domains, including node classification, spatio-temporal graph analysis, and knowledge graph completion, while significantly reducing computational complexity.

**Strengths:**

Overall the work is well motivated and demonstrates strong empirical results. The paper introduces a densifying weak regularity lemma for directed graphs, which improves upon prior formulations and is supported by sound proofs. It demonstrates robustness to sparsity through the use of a weighted Frobenius norm and densifying cut similarity, enabling accurate approximation of sparse graphs as validated by experimental results. Empirically, the proposed IBG-NN architecture achieves state-of-the-art performance across diverse tasks, including node classification, spatio-temporal graph analysis, and knowledge graph completion. Furthermore, the work emphasizes reproducibility by providing a public codebase and detailed hyperparameter settings, ensuring that the results can be reliably replicated.

**Weaknesses:**

Performance seems depend on the hyperparameters such as $\Gamma$ and $K$, but tuning guidelines are minimal.

**Questions:**

How sensitive is the performance to the choice of hyperparameters such as $\Gamma$ and $K$ in practice? Any guidelines for setting these?

---

> ### Author Response · Authors · 2025-11-19
>
> We thank the Reviewer for their thoughtful feedback and for recognizing the novelty, versatility, and theoretical contributions of our approach.
>
> >W1: "Performance seems to depend on the hyperparameters such as Γ and K, but tuning guidelines are minimal."
>
> >Q1: "How sensitive is the performance to the choice of hyperparameters such as Γ and K in practice? Any guidelines for setting these?"
>
> This is a good point, and we agree that more explicit guidelines could help practitioners in the future. To address this concern, we conducted a sensitivity analysis across the Squirrel, Chameleon, and Arxiv-Year datasets (added in Appendix M.5.1, page 51 of the updated manuscript)
>
> We train IBGs with varying values of $K$ and $\Gamma$, with $K \in \{25, 50, 100, 200, 250, 400\}$ and $e_{E,\Gamma} \in \{0, 0.01, 0.05, 0.1, 0.25, 1\}$, where $e_{E,\Gamma}=1$ implies no densification was used. We report test accuracy averaged over 10 splits for Squirrel and Chameleon, and over 5 seeds for Arxiv-Year.
>
> ## Squirrel
>
> | $\Gamma$ | K=25 | K=50 | K=100 | K=200 | K=250 | K=400 | $e_{E,\Gamma}$ |
> |---|------|------|-------|-------|-------|-------|---------|
> | 0 | 61.7 | 69.7 | 72.3 | 74.2 | 73.7 | 74.8 | 0 |
> | 1 | 76.3 | 76.3 | 76.3 | 75.7 | 76.9 | 76.8 | 0.01 |
> | 6 | 74.9 | 75.5 | 76.3 | 76.9 | 77.5 | 76.8 | 0.05 |
> | 12 | 74.3 | 75.8 | 76.9 | 75.8 | 77.1 | 77.2 | 0.1 |
> | 31 | 71.6 | 72.8 | 72.8 | 73.2 | 73.7 | 73.5 | 0.25 |
> | 124 | 66.5 | 68.5 | 69.3 | 70.4 | 71.5 | 71.3 | 1 (no dens.) |
>
> ## Chameleon
>
> | $\Gamma$ | K=25 | K=50 | K=100 | K=200 | K=250 | K=400 | $e_{E,\Gamma}$ |
> |---|------|------|-------|-------|-------|-------|---------|
> | 0 | 70.2 | 73.1 | 76.2 | 77.4 | 78.6 | 78.6 | 0 |
> | 1 | 77.5 | 77.9 | 78.6 | 79.0 | 79.3 | 79.3 | 0.01 |
> | 7 | 77.1 | 79.2 | 78.4 | 79.1 | 79.8 | 79.7 | 0.05 |
> | 14 | 76.8 | 78.8 | 77.9 | 77.8 | 78.9 | 78.4 | 0.1 |
> | 36 | 73.5 | 73.9 | 74.8 | 75.4 | 75.1 | 75.2 | 0.25 |
> | 143 | 70.9 | 71.0 | 71.5 | 72.2 | 72.9 | 71.4 | 1 (no dens.) |
>
> ## Arxiv-Year
>
> | $\Gamma$ | K=25 | K=50 | K=100 | K=200 | K=250 | K=400 | $e_{E,\Gamma}$ |
> |-----|------|------|-------|-------|-------|-------|---------|
> | 0 | 44.9 | 45.3 | 45.4 | 46.3 | 47.2 | 47.2 | 0 |
> | 246 | 53.0 | 55.0 | 50.4 | 48.9 | 47.0 | 43.7 | 0.01 |
> | 1229 | 50.4 | 52.6 | 53.6 | 54.5 | 55.3 | 54.0 | 0.05 |
> | 2459 | 49.0 | 51.9 | 52.7 | 48.9 | 54.0 | 45.9 | 0.1 |
> | 6147 | 47.5 | 48.9 | 49.6 | 50.1 | 51.1 | 50.5 | 0.25 |
> | 24588 | 44.0 | 44.2 | 44.4 | 45.2 | 45.9 | 45.0 | 1 (no dens.) |
>
>
> The results reveal that $K=250$ and $e_{E,\Gamma}=0.05$ consistently achieve top performance across all three datasets. Moreover, a general trend seems to be that ($K$, $e_{E,\Gamma}$) points that are further away from ($K=250$, $e_{E,\Gamma}=0.05$) have worse performance than closer ones, suggesting ($K=250$, $e_{E,\Gamma}=0.05$) as a promising starting point.
>
> The table also showcases ($K=25$, $e_{E,\Gamma}=0.01$) as a more efficient alternative which uses less communities, while still remaining close in accuracy to the top-performing choice.
>
> **We have added all the new experimental results and suggestions provided by the Reviewer in the updated manuscript with the appropriate discussion.**
>
> We hope our responses address the Reviewer's concerns.

---

> > ### Author Response · Authors · 2025-11-25
> >
> > **Thank you for suggesting that we ablate over the choice of the $\Gamma$ and $K$ hyperparameters.**
> >
> > While researchers can build on the theoretical guarantees we provide, the ablation study you suggested increases the contribution towards practitioners as well, **making the work more complete and useful for a wider audience. This work can lead to a new set of graph machine learning algorithms** designed specifically for large graphs, valuable for both practitioners and researchers.
> >
> > We believe we have addressed all of your concerns and **would highly appreciate your response** to ensure no outstanding issues remain.

---

### Author Response · Authors · 2025-11-30
**Summary for AC**

We sincerely thank the new Area Chairs for taking on this additional workload. To assist you, we would like to provide a brief summary of the process so far.

The initial reviews for our work were very supportive, with an average score of 7. **In our rebuttal, we successfully addressed all reviewer concerns, resulting in a unanimous score of 8 from all reviewers.**

We believe our fundamentally different paradigm offers a scalable and effective contender to standard MPNNs for learning on large-scale graphs. **We sincerely hope our work will benefit researchers, practitioners, and the wider graph machine learning community alike.**

Thank you.

---

### Meta-Review · Area_Chair_FnY1 · 2026-01-06

**Summary:**

Reviewers agreed that the paper’s Intersecting Block Graph (IBG) representation and the associated densifying weak regularity lemma for directed (including sparse) graphs are novel, technically well-supported, and paired with strong empirical results and clear scalability advantages (node-linear time/memory rather than edge-linear). The main concerns raised during review centered on (i) practical sensitivity/guidance for the Γ/K hyperparameters, (ii) reliability of the nonconvex/semi-constructive IBG fitting optimization, and (iii) clarity/positioning (IBG vs. ICG, why densification is beneficial, and how IBG-NN differs from message passing), plus minor presentation issues. The rebuttal and revision directly addressed these points with additional ablations/diagnostics and clarifications, which led the initially borderline reviewers to raise their ratings. As AC, I carefully read the submission and appendix, checked that the key theoretical claims align with the stated contributions, and confirmed that the newly added experiments and explanations squarely target the reviewers’ methodological questions. On balance, I recommend accept for this submission.

**Reviewer Concerns:**

Addressed:
- Γ/K sensitivity and practitioner guidance were strengthened via a new sensitivity analysis and suggested defaults; the SVD-vs-random initialization question was answered with direct convergence/performance comparisons; and multiple clarity issues (IBG vs ICG, the densification intuition, and “IBG-NN is not an MPNN”) were incorporated into the revised writeup.
- concerns about the semi-constructive/nonconvex fitting procedure were mitigated by empirical convergence evidence and a clearer discussion of the efficiency–guarantee trade-off.

Still outstanding (non-blocking): the approach remains nonconvex without a formal global convergence guarantee, and additional qualitative/interpretability case studies could further improve the narrative.

**Reviewer Scores:**

Reviewer Besy: 8→8; Reviewer tVWW: 8→8; Reviewer zVnQ: 6→8; Reviewer jEYc: 6→8. Given the added experiments and clarifications already reflected in the discussion outcomes, I do not expect further score movement beyond these changes.

---

### Decision · Program_Chairs · 2026-01-26

Accept (Poster)